# The organizational principles of de-differentiated topographic maps in somatosensory cortex

Peng Liu[1,2†], Anastasia Chrysidou[1,2†], Juliane Doehler[1,2], Martin N Hebart[3], Thomas Wolbers[2,4†], Esther Kuehn[1,2,4*]

[1]Institute for Cognitive Neurology and Dementia Research (IKND), Otto-von-Guericke University Magdeburg, Magdeburg, Germany; [2]German Center for Neurodegenerative Diseases (DZNE), Magdeburg, Germany; [3]Vision and Computational Cognition Group, Max Planck Institute for Human Cognitive and Brain Sciences, Leipzig, Germany; [4]Center for Behavioral Brain Sciences (CBBS) Magdeburg, Magdeburg, Germany

**Abstract** Topographic maps are a fundamental feature of cortex architecture in the mammalian brain. One common theory is that the de-differentiation of topographic maps links to impairments in everyday behavior due to less precise functional map readouts. Here, we tested this theory by characterizing de-differentiated topographic maps in primary somatosensory cortex (SI) of younger and older adults by means of ultra-high resolution functional magnetic resonance imaging together with perceptual finger individuation and hand motor performance. Older adults' SI maps showed similar amplitude and size to younger adults' maps, but presented with less representational similarity between distant fingers. Larger population receptive field sizes in older adults' maps did not correlate with behavior, whereas reduced cortical distances between D2 and D3 related to worse finger individuation but better motor performance. Our data uncover the drawbacks of a simple de-differentiation model of topographic map function, and motivate the introduction of feature-based models of cortical reorganization.

**\*For correspondence:**
esther.kuehn@dzne.de

[†]These authors contributed equally to this work

**Competing interests:** The authors declare that no competing interests exist.

## Introduction

Topographic maps are a fundamental feature of cortex architecture and can be found in all sensory systems and in many motor systems of the mammalian brain. Topographic units organize subcortical brain structures such as the thalamus, the globus pallidus, and the striatum (*Crabtree, 1992*; *Hintiryan et al., 2016*; *Zeharia et al., 2015*), primary sensory input and output areas such as primary sensory and motor cortices (*Penfield and Boldrey, 1937*), and higher level integrative brain areas such as the medial and superior parietal cortices and the cingulate cortex (*Sereno and Huang, 2006*; *Zeharia et al., 2019*; *Zeharia et al., 2015*). Topographic maps and their malfunctions give rise to a multitude of sensory, motor, and cognitive functions and associated deficits (*Amedi et al., 2003*; *Kalisch et al., 2009*; *Kikkert et al., 2019*; *Kuehn et al., 2018*; *Makin et al., 2013a*; *Saadon-Grosman et al., 2015*). This warrants a precise understanding of their organizational features and their associated adaptive and maladaptive behavior.

One common theory posits that the 'de-differentiation' of topographic maps represents one mechanism of their malfunction. Cortical de-differentiation can be conceptualized as greater map activation (*Pleger et al., 2016*), a larger topographic map area (*Kalisch et al., 2009*), but also more noisy topographic units and/or less cortical inhibition between neighboring topographic units (*Lenz et al., 2012*; *Pleger et al., 2016*). Such changes are particularly observed in older adults' topographic maps, and one common model on cortical aging assumes 'overactivated' or more 'de-

differentiated' topographic maps in older compared to younger adults, which are assumed to explain reduced sensory, motor, and cognitive abilities of older adults in everyday life (*Cabeza, 2002*; *Cassady et al., 2020*; *Dennis and Cabeza, 2011*; *Heuninckx et al., 2008*; *Mattay et al., 2002*; *Reuter-Lorenz and Lustig, 2005*; *Riecker et al., 2006*). However, the precise topographic features that characterize a presumably 'de-differentiated' map are so far not clarified, neither are the precise behavioral phenotypes that relate to different aspects of topographic map change (*Cassady et al., 2020*).

Here, we used the hand area of the primary somatosensory cortex (SI) in younger and older adults as a model system to study the precise meso-scale features that characterize the presumably de-differentiated topographic maps of older adults, and their relation to behavior. Topographic maps in SI are a suitable model system to investigate basic aspects of cortical de-differentiation, because the tactile modality is not artificially corrected by glasses or hearing aids, and therefore offers access to the 'pure' architecture of the (altered) system. We assessed the functional architecture of topographic maps subject-wise at fine-grained detail using ultra-high-field functional magnetic resonance imaging at 7 Tesla (7T-fMRI), and investigated sensory readouts as well as everyday hand movement capabilities of our participants. 7T-fMRI is a valuable method for describing fine-grained features of topographic maps, because it allows mapping small-scale topographic units, such as individual fingers, subject-wise and with high levels of accuracy and reproducibility (*Kolasinski et al., 2016a*; *Kuehn et al., 2018*; *Kuehn and Sereno, 2018*; *O'Neill et al., 2020*). Recently, this allowed the precise description of features that characterize non-afferent maps in human SI (*Kuehn et al., 2018*), SI map changes after short-term plasticity (*Kolasinski et al., 2016b*), or movement-dependent maps in motor cortex (*Huber et al., 2020*).

To systematically characterize the meso-scale features of de-differentiated topographic maps and their relation to human behavior, we distinguished between global changes of the map that were present across topographic units (i.e. across finger representations), and local changes that only covered parts of the map (see *Figure 1*). This distinction is relevant due to the nonhomogeneous use of individual fingers in everyday life (*Belić and Faisal, 2015*), the non-uniform microstimulation-evoked muscle activity in motor cortex (*Overduin et al., 2012*), and for differentiating between age-dependent and use-dependent plasticity (*Makin et al., 2013a*). We also distinguished between topographic map features that link to functional separation, as here tested by perceptual finger individuation and by motor tasks that rely on precise spatial acuity of the fingertip (i.e. Pegboard test, *Kalisch et al., 2008*), and those that require functional integration, as here tested by perceptual finger confusion and a motor task that relies on haptic recognition involving multiple fingers (i.e. O'Connor Dexterity test). It is worth noting that local and global changes as well as integration and separation that are here introduced as different levels of the features 'spatial extent' and 'functional readout' (see *Figure 1*), may be interlinked and may share common variance. For example, less finger individuation in one task may relate to more finger integration in another task, and both may influence motor behavior. However, their distinct investigation allows a precise understanding of how specific map features link to behavioral phenotypes (i.e. features-based model of cortical reorganization, FMC, see *Figure 1*).

A cohort of healthy younger adults (21–29 years) and healthy older adults (65–78 years) was invited to several experimental sessions, where touch to their fingertips was applied in the 7T-MR scanner using an automated piezoelectric tactile stimulator (*Miller et al., 2018*; *Schmidt and Blankenburg, 2018*). They were also tested behaviorally in a finger mislocalization task (*Schweizer et al., 2000*), in the two-point discrimination task (*Timm and Kuehn, 2020*), and in three motor tests (*Kalisch et al., 2008*). By combining ultra-high-resolution functional magnetic resonance imaging with population receptive field mapping, Fourier-based functional analyses, representational similarity analysis, psychophysics, and measures of everyday behavior, we could compare precise map features that differed between younger and older adults' topographic maps, and link these to behavioral phenotypes relevant for everyday life. We could therefore test the basic assumption that de-differentiated cortical maps relate to impairments in everyday behavior. By targeting a mechanism that is assumed to be a hallmark feature of cortical aging, our data also help to uncover a fundamental principle of brain aging.

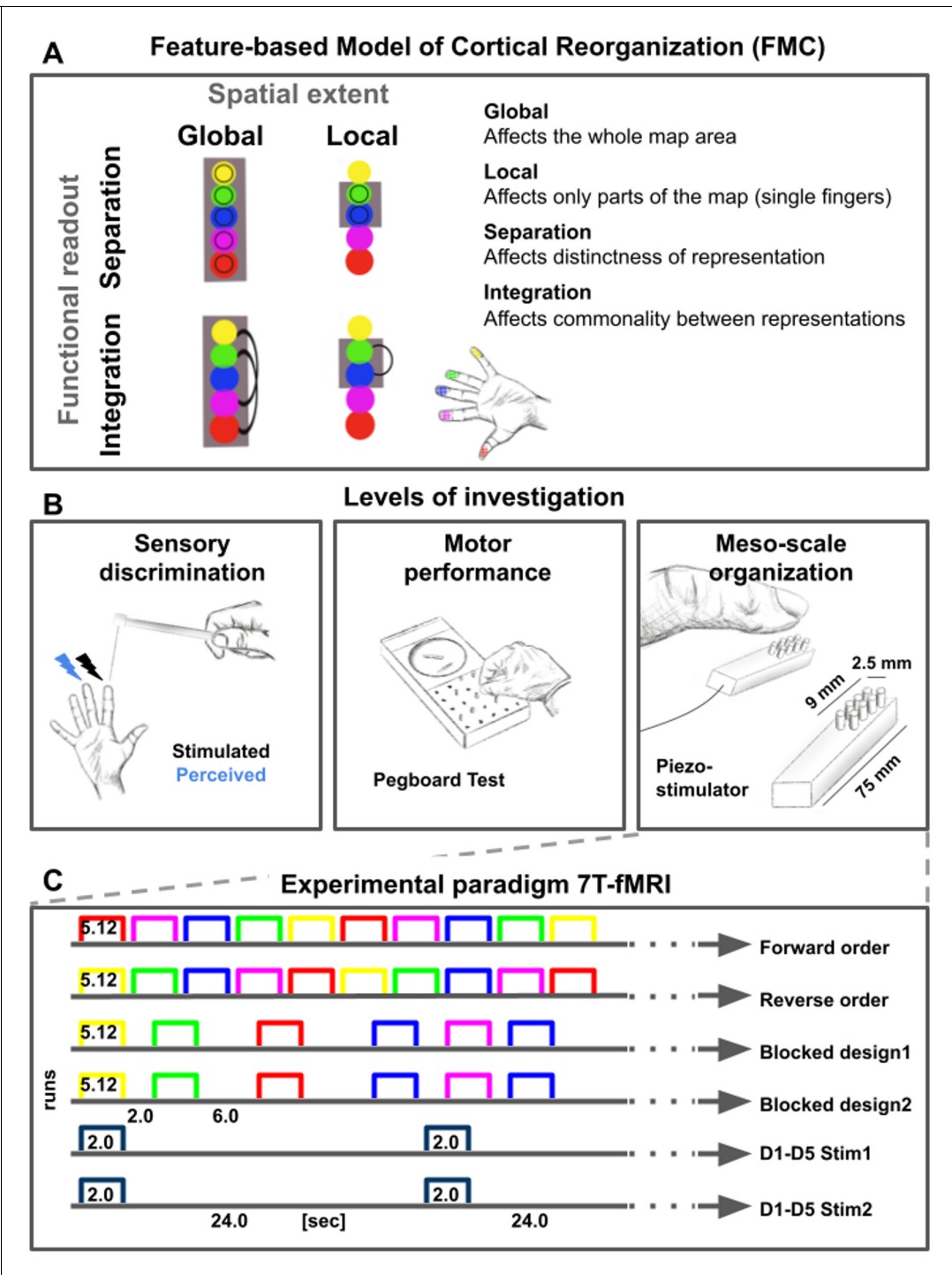

**Figure 1.** Feature-based characterization of meso-scale topographic maps in younger and older adults using 7T-fMRI and behavioral tests. (A) A feature-based model of cortical reorganization (FMC) requires a distinction between global and local map changes (factor of spatial extent), and between effects on functional integration versus effects on functional separation (factor of functional readout). (B) Younger and older adults' somatosensory thresholds were tested via tactile detection, spatial tactile acuity, and digit confusion (*left*). They were also characterized for individual differences in hand motor behavior using the Purdue Pegboard Test, the Grooved Pegboard Test, and the O'Connor Dexterity Test (*middle*). Participants underwent 7T-fMRI at a separate testing day, where tactile finger stimulation was applied using piezoelectric stimulators (one module per fingertip, five in total, *right*). (C) In the 7T-MRI scanner, different stimulation protocols were tested in separate runs (each row represents one run, note that one 5 min resting state run was acquired at the end). See *Figure 1—figure supplement 1* for an overview of analyses pipelines.

The online version of this article includes the following figure supplement(s) for figure 1:

*Figure 1 continued on next page*

*Figure 1 continued*

**Figure supplement 1.** Overview of fMRI analyses.

## Results

### Surface area and % signal change of area 3b topographic maps do not differ significantly between younger and older adults

We used 7T-fMRI data to compare the fine-grained architecture of topographic finger maps in SI between younger and older adults. Older adults were expected to present with more 'de-differentiated' cortical maps compared to younger adults, which is assumed to link to higher map amplitude and larger map size (*Kalisch et al., 2009*; *Pleger et al., 2016*). While undergoing fMRI scanning, younger and older adults were stimulated at the fingertips of their right hand using an automated piezoelectric stimulator, and different stimulation protocols (see *Figure 1*). Participants were stimulated at each finger at their 2.5-fold individual threshold to exclude topographic map changes that were due to peripheral (nerve or skin) differences between younger and older adults. We focused on topographic maps in area 3b of SI, because this area is the likely human homologue of the monkey SI cortex (*Kaas, 2012*).

Significant topographic finger maps in contralateral area 3b in response to finger stimulation were detected in younger and older adults, and across the group as a significant group effect (see *Figure 2A–C*, see *Figure 2—figure supplement 1* and *Figure 2—figure supplement 2* for zoomed-in individual maps). The topographic Fourier-based maps were, as expected, composed of the thumb [D1], the index finger [D2], the middle finger [D3], the ring finger [D4], and the small finger [D5] in all individuals of both age groups (see *Figure 2C*). The mean surface area that topographic maps covered in area 3b, % signal change within the map area, and mean f-values did not differ significantly between age groups (original surface area: $t(34)=0.04$, $p=0.97$, Cohen's $d = 0.10$; resampled surface area: $t(34)=-0.15$, $p=0.88$, $d = 0.05$; % signal change: $t(34)=1.17$, $p=0.25$, $d = 0.38$; f-value: $t(34)=0.84$, $p=0.41$, $d = 0.28$, see *Figure 2D–G*). This was confirmed for % signal change using the two one-sided t-test (TOST) for equivalence. TOST is a frequentist alternative for testing for the equivalence by defining a band around 0 that constitutes the minimally-relevant effect size ($\Delta L$ and $\Delta U$). TOST works (1) by running two t-tests, which test the null hypothesis that the effect is smaller than the maximum of the indifference area and larger than its minimum, and (2) by choosing the smaller of the two t-values. A significant result would reject this null hypothesis, indicating that the true value lies in the indifference area (*Lakens, 2017*). This was the case for % signal change ($t(34)=-2.11$, $p=0.000044$), which was significant, that is equivalent. Statistical equivalence using the TOST test was not confirmed for mean f-values ($t(34)=0.55$, $p=0.13$). Note that the TOST test for equivalence was only performed when effect sizes of non-significant differences were $d > 0.2$, that is, if there was a small effect based on *Cohen, 1988*.

To test for possible local differences, we compared % signal change, mean f-values, and mean surface areas finger-wise within the topographic map area between younger and older adults. We did not find a significant interaction between age and digit (% signal change: $F(136)=1.20$, $p=0.31$; f-value: $F(136)=1.17$, $p=0.33$; surface area: $F(136)=1.64$, $p=0.17$, see *Figure 2H–J*), which would be expected if finger-specific differences in any of these variables existed.

One further variable that may explain age-related differences in topographic maps is the variability of topographic map alignments *within* age groups that may be due to increased internal noise or distorted maps. One may expect the variability to be higher in older adults' compared to younger adults' topographic maps (*McGregor et al., 2012*). To inspect topographic map variability within each age group, we calculated the dispersion index $d$, which indicates map stability across the group ($d = 1$ indicates perfectly aligned vectors independent of vector amplitude, whereas lower $d$ indicates less stable topographic arrangements between individuals in one group, *Hagler et al., 2006*). Younger participants showed lower $d$ in the topographic map area compared to older adults ($d$ younger: $0.68 \pm 0.002$, $d$ older: $0.72 \pm 0.002$). The variability of topographic map alignments within each age group was therefore generally low, and slightly *higher* in younger compared to older adults (see *Figure 2B*).

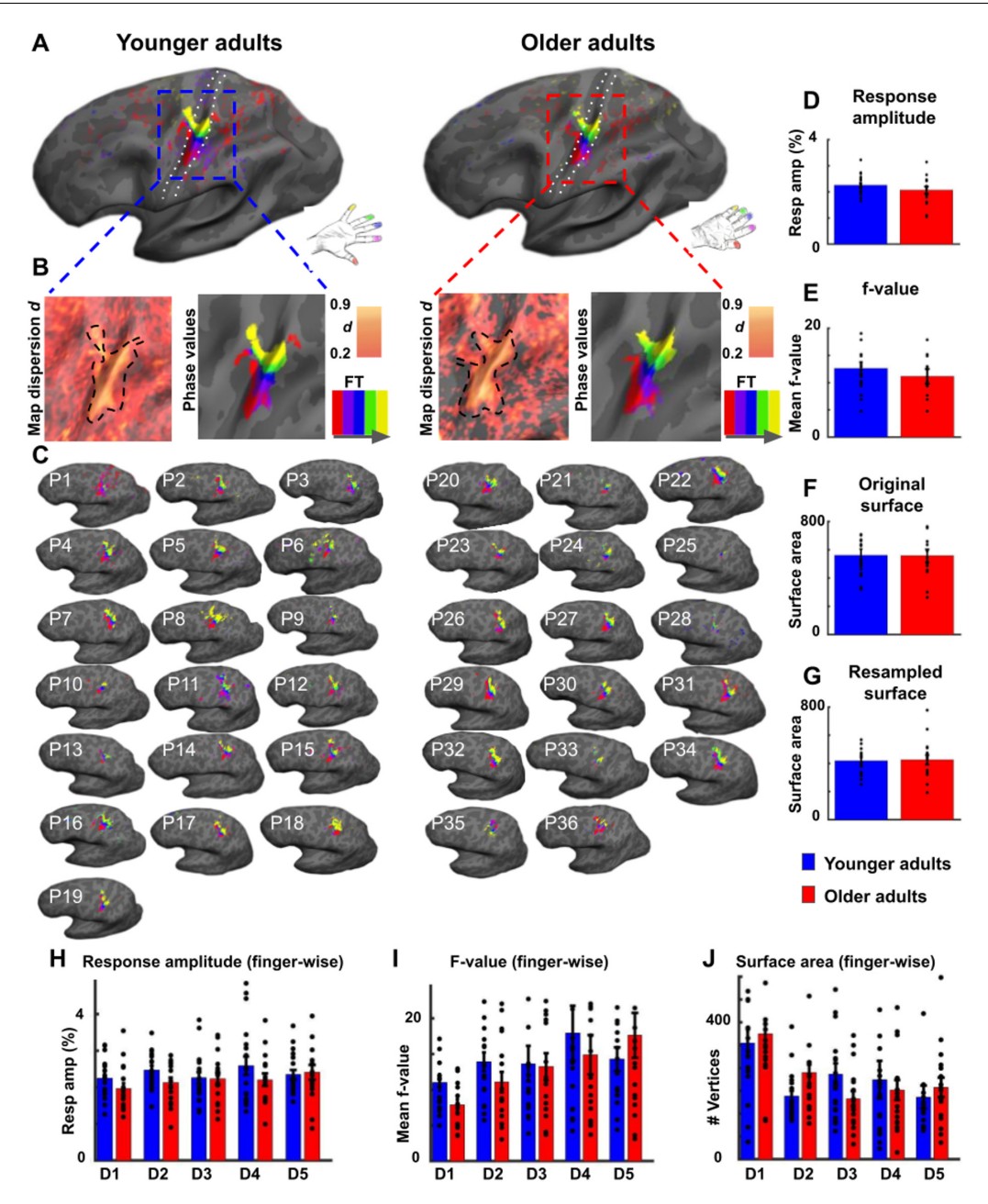

**Figure 2.** Surface area and % signal change of area 3b topographic maps do not differ significantly between younger and older adults. (A) Significant topographic finger maps of younger and older adults averaged separately over the group of younger adults and over the group of older adults. Data are visualized on average surfaces of the current set of subjects (younger/older).White dotted lines show the schematic outline of area 3b. (B) Map dispersion $d$ and Fourier transformed (FT, gray arrow indicates stimulation order) data of significant topographic group maps of younger and older adults ($d$ young: 0.68 ± 0.002, $d$ old: 0.72 ± 0.002). Lower $d$ indicates less stable topographic arrangements over the group. Black dotted lines indicate the area of the significant topographic map. (C) Significant topographic FT maps of each single participant (P1-P36). See *Figure 2—figure supplement 1* and *Figure 2—figure supplement 3* for zoomed-in views of individual maps. (D,H) Response amplitudes (in %) of topographic maps (D) and of individual fingers (H) in area 3b compared between younger and older adults (mean ± SE and individual data). (E,I) f-values of topographic maps (E) and of individual fingers (I) in area 3b compared between younger and older adults (mean ± SE and individual data). (F,G,J) Surface area of topographic maps of area 3b of younger and older adults; values extracted from original (F) and resampled (G) surfaces of the topographic maps, and of individual fingers (J) (mean ± SE and individual data). Shown are data of n = 19 younger adults and n = 17 older adults. See *Figure 2—figure supplement 3* for significant differences in response amplitudes between younger and older adults during fixed amplitude stimulation.

The online version of this article includes the following figure supplement(s) for figure 2:

**Figure supplement 1.** Zoomed-in topographic maps of younger adults.

*Figure 2 continued on next page*

*Figure 2 continued*
**Figure supplement 2.** Zoomed-in topographic maps of older adults.
**Figure supplement 3.** Tactile detection thresholds and response amplitudes during fixed amplitude stimulation in younger and older adults.

In the above reported analyses, time-series were used where participants were stimulated at their 2.5-fold individual threshold (calculated separately for each finger); an approach that was employed to prevent age-effects that were due to differences in the peripheral (nerve or skin) architecture (see above). To provide additional information on possible differences in cortical excitability between age groups, we used time-series that were acquired when all fingers were stimulated at once and at the same, fixed amplitude (see *Figure 1C*). We computed response amplitudes from these data, and found significantly higher response amplitudes in older adults compared to younger adults' SI maps (younger: 1.05 ± 0.01, older: 1.08 ± 0.008, $t(34)$=-2.10, p<0.05, Cohen's $d$ = 0.70 see *Figure 2—figure supplement 3*). Note that these data cannot be used to disentangle the effect of multiple finger stimulation from the effect of fixed amplitude stimulation. Mean response amplitude across fingers did not correlate with mean detection thresholds across fingers (see *Figure 2—figure supplement 3*).

## Reduced cortical distances between representations of D2 and D3 in older adults

Previous studies found *larger* cortical distances between the representations of D2 and D5 in older adults compared to younger adults, which was argued to evidence a global enlargement of topographic maps in older adults (*Kalisch et al., 2009*). At the same time, this effect could also reflect local changes in the topographic alignment between individual digit pairs. We used both absolute (Euclidean) and surface-based (geodesic) cortical distances measures to compare distances of digit representations in SI between younger and older adults to test for both global and local differences (see *Figure 3C*). We used peak representations and plotted each finger onto a three-dimensional grid (see *Figure 3E* for group averages and *Figure 3—figure supplement 1* for individual maps). An ANOVA with the factors finger-pair and age calculated on Euclidean distances revealed a significant main effect of finger-pair (F(3,102)=11.20, p<$10^{-5}$), no main effect of age (F(1,34)=1.69, p=0.20), but a significant interaction between finger-pair and age (F(3)=3.23, p<0.05). The main effect of finger-pair was due to increased Euclidean distances between D1 and D2 compared to D3 and D4 ($t$(35)=5.57, p<0.00001), reduced Euclidean distances between D1 and D2 compared to D4 and D5 ($t$(35)=4.87, p<0.0001), increased Euclidean distances between D2 and D3 compared to D3 and D4 ($t$(35)=3.24, p<0.01), and reduced Euclidean distances between D2 and D3 compared to D4 and D5 ($t$(25)=2.93, p<0.01) across age groups (see *Figure 3A*). The interaction between finger-pair and age was driven by significantly reduced Euclidean distances between D2 and D3 in older adults compared to younger adults (Euclidean distance D2-D3 younger: 7.67 ± 0.80, D2-D3, older: 4.98 ± 0.40, p<0.05, $d$ = 0.98, see *Figure 3A*). The latter effect was replicated for geodesic distances, where older adults showed significantly reduced geodesic distances compared to younger adults only between D2 and D3 (Geodesic distance D2-D3 younger: 7.80 mm ±0.72 mm, older: 5.72 mm ±0.64 mm, p<0.05, $d$ = 0.71, see *Figure 3B*). The latter effect also presents as a trend when using center estimates instead of peak vertices as a metric for estimating digit location (see *Figure 3—figure supplement 2* for full statistics and visualization). Effect size analyses using bootstrapping confirm large Hedge's $g$ for Euclidean and geodesic distances between D2 and D3, and low Hedge's $g$ for all other distances (D2-D3 Euclidean: $g$ = 0.73, LCI = 0.12, UCI = 1.45; D2-D3 Geodesic: $g$ = 0.69, LCI = 0.08, UCI = 1.45, all other distances $g$ < 0.3, see *Figure 3D*). We also tested whether we could replicate the enlargement of topographic maps with respect to the distance between D2 and D5 (*Kalisch et al., 2009*). There were no significant differences between the cortical distance of D2 and D5 neither for Euclidean distances ($t$(34)=-0.35, p=0.72, Cohen's $d$ = 0.12) nor for geodesic distances ($t$(34)=-0.11, p=0.91, $d$ = 0.04).

## Larger population receptive field (pRF) sizes in older adults

Previous studies on rats had indicated larger receptive field sizes in the SI hindpaw but not forepaw representation of older compared to younger rats (*Godde et al., 2002*). This left open the question

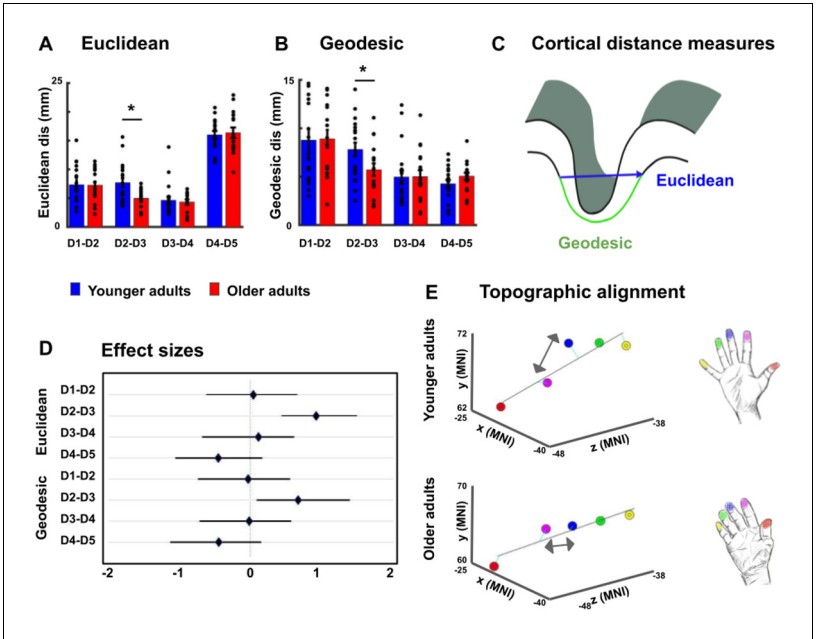

**Figure 3.** Reduced cortical distances between representations of D2 and D3 in older adults. (**A, B**) Cortical distances between digit representations in younger and older adults estimated as total (Euclidean) distance (**A**) and as surface-based (geodesic) distance (**B**) (mean ± SEM and individual data) (**C**) Schematic visualization of cortical distance measures. (**D**) Effect sizes (Hedge's *g* and 95% confidence intervals) for Euclidean and geodesic distances. (**E**) Spatial alignment of younger adults' (*top*) and older adults (*bottom*) digit representations in area 3b displayed in the MNI coordinate system. Line represents linear fit, arrows highlight significant differences in cortical distance between younger and older adults. Shown are data of n = 19 younger adults and n = 17 older adults. See *Figure 3—figure supplement 1* for individual plots, and *Figure 3—figure supplement 2* for cortical distances using center estimates.

The online version of this article includes the following figure supplement(s) for figure 3:

**Figure supplement 1.** Spatial alignment of individual maps.
**Figure supplement 2.** Cortical distances based on center estimates.

whether or not there are enlarged population receptive field (pRF) sizes in the human hand area in older adults compared to younger adults. Bayesian pRF modeling was employed to model pRFs in individual topographic maps, and to compare pRF sizes between younger and older adults. pRF centre locations were used to individuate the five fingers, and pRF sizes were extracted map- and finger-specific in each individual (*Puckett et al., 2020*) (see *Figure 4A,C*, see *Figure 4—figure supplement 1* for individual data and for a comparison between pRF-based and Fourier-based topographic maps). An ANOVA with the factors age and finger calculated on pRF sizes was used to test for both global and local differences in pRF size. The analysis revealed a significant main effect of finger ($F(4)=13.87$, $p<10^{-8}$), a significant main effect of age ($F(1)=4.15$, $p<0.05$), but no significant interaction between age and finger ($F(4)=1.31$, $p=0.27$). The main effect of finger was due to significantly smaller pRF sizes of D1 compared to D2 (D1: 4.40 ± 1.12, D2: 5.64 ± 1.81, $t(66)=-3.41$, $p<0.01$), D1 compared to D4 (D4: 6.92 ± 1.92, $t(64)=-6.61$, $p<10^{-8}$), D1 compared to D5 (D5: 5.89 ± 1.53, $t(66)=-4.58$, $p<10^{-4}$), D2 compared to D4 ($t(62)=-2.75$, $p<0.01$), D3 compared to D2 (D3: 4.44 ± 1.63, $t(63)=2.82$, $p<0.01$), D3 compared to D4 ($t(61)=-5.56$, $p<10^{-6}$), D3 compared to D5 ($t(63)=-3.71$, $p<10^{-3}$) and D5 compared to D4 ($t(62)=2.40$, $p<0.05$). The main effect of age was due to larger pRF sizes in older adults compared to younger adults (pRF size younger: 5.23 ± 1.09, older: 5.69 ± 1.14, d = 0.41, see *Figure 4D*). Note that there were finger representations 'missing' for 8 out of 36 subjects after Bayesian pRF mapping, as shown in *Figure 4—figure supplement 1*. This is equal for the group of younger and older adults (n = 4 in each group). The ANOVA with the factors age and finger calculated on pRF sizes was calculated with the missing cases excluded from the data. This is justified because overall, the missing values take up 8.9% of the data where biases are

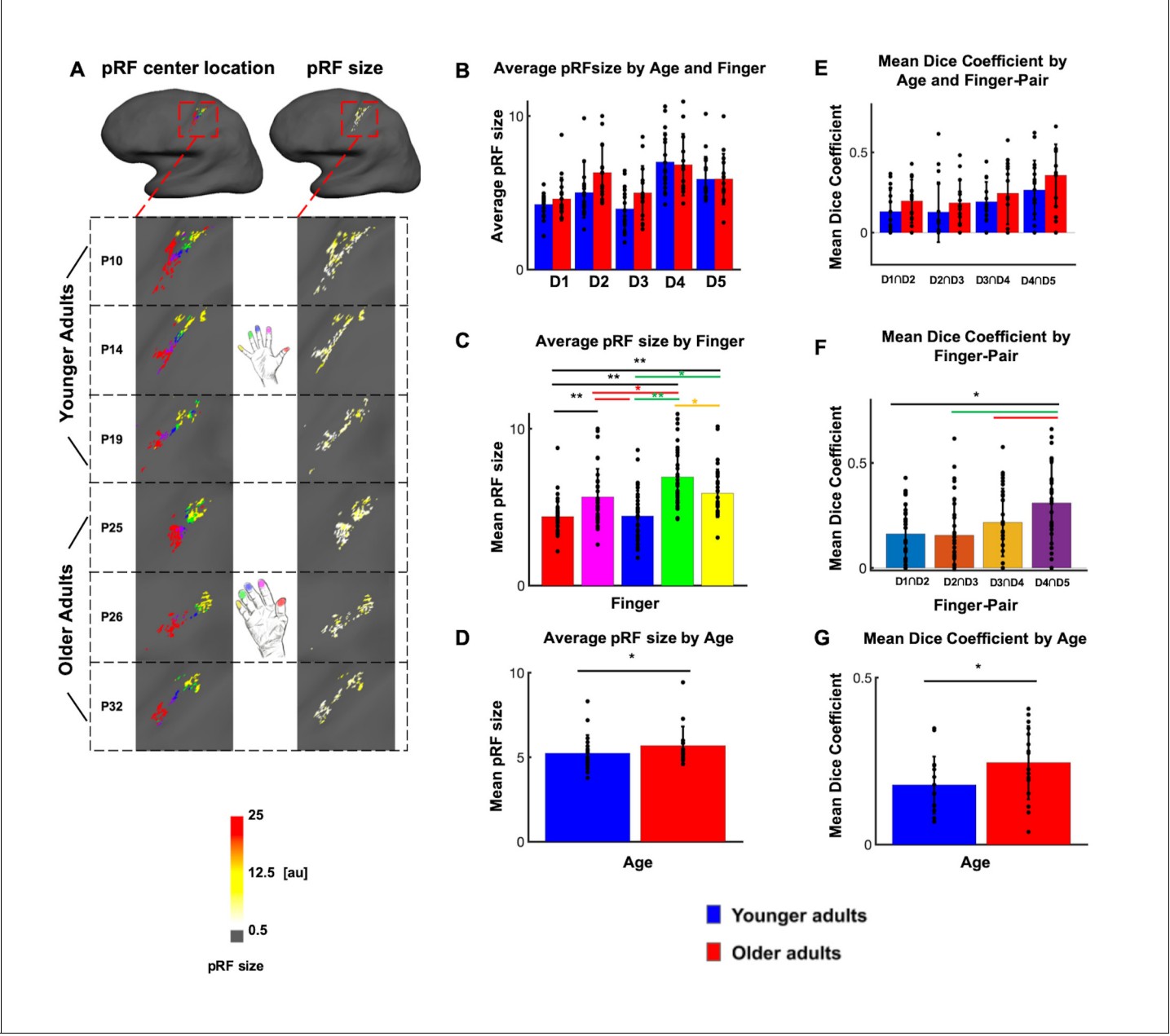

**Figure 4.** Larger population receptive field (pRF) sizes in older adults compared to younger adults. (**A**) pRF centre locations (which encode each individual finger, *left column*) and pRF sizes (which encode the estimated pRF size of each finger representation, *right column*) shown for six individual participants (randomly chosen, participant numbers same as in *Figure 2*, see *Figure 4—figure supplement 1* for all individual data and a comparison to the Fourier transformed maps shown in *Figure 2*). (**B**) Average pRF sizes for each finger for younger and older adults (mean ± SEM and individual data). (**C**) Visualization of significant main effect of finger for pRF sizes (mean ± SEM and individual data). (**D**) Visualization of significant main effect of age for pRF sizes (mean ± SEM and individual data). Correlations between pRF sizes and cortical distances, motor behavior and tactile discrimination performance are shown in *Figure 4—figure supplement 2*. Shown are data of n = 19 younger adults and n = 17 older adults. (**E**) Mean dice coefficients for each finger pair for younger and older adults (mean ± SEM and individual data). (**F**) Visualization of significant main effect of finger-pairs for dice coefficient (mean ± SEM and individual data). (**G**) Visualization of significant main effect of age for dice coefficient (mean ± SEM and individual data).

The online version of this article includes the following figure supplement(s) for figure 4:

**Figure supplement 1.** Comparison between Fourier-based topographic maps and pRF-based topographic maps for each younger and each older adult.

**Figure supplement 2.** Correlations between average pRF size and functional and behavioral variables.

**Figure supplement 3.** Dice coefficients at different statistical thresholds in younger and older adults.

expected when more than 10% of the data are missing (*Bennett, 2001*; *Dong and Peng, 2013*). Furthermore, there were no significant correlations between averaged pRF size and pegboard test results, and between D2 pRF size and 2PD thresholds (see *Figure 4—figure supplement 2*).

In addition, dice coefficients were used to compare the overlap of neighboring digit representations between younger and older adults. When computing an ANOVA with the factors finger-pair and age on dice-coefficients, the results show both a significant main effect of age (F(1)=5.49, p<0.05) and a significant main effect of finger-pair (F(3)=6.24, p<10$^{-3}$). There was no significant interaction between finger pair and age (F(3)=0.1, p=0.963). The main effect of finger-pair was due to significant smaller dice coefficients of D1-D2 compared to D4-D5 (D1-D2: 0.162, D4-D5: 0.309, t (64)=-3.58, p<10$^{-3}$), D2-D3 compared to D4-D5 (D2-D3: 0.155, t(64)=-3.48, p<10$^{-3}$) and D3-D4 compared to D4-D5 (D3-D4: 0.217, t(64)=-2.12, p<0.04). The main effect of age was due to higher mean dice coefficients in older adults compared to younger adults (dice coefficient younger: 0.18 ± 0.09, older: 0.25 ± 0.11, $d$ = 0.70, see *Figure 4G*). This effect was visible for different statistical thresholds (*Figure 4—figure supplement 3*).

## Lower representational similarity between distant finger representations in older adults

Another aspect of cortical de-differentiation is the assumed increased 'blurriness' of de-differentiated cortical maps. This was investigated here by using representational similarity analyses. We used across-run representational similarity analyses to compare the similarity of digit representations between different runs within area 3b (*Kuehn et al., 2018*). We computed an ANOVA with the factors neighbor and age on finger-specific representational similarity, which revealed a significant main effect of neighbor (F(4,136)=128.6, p<10$^{-44}$), no main effect of age (F(1,34)=1.7006, p=0.20, Cohen's $d$ = 0.02), but a significant interaction between age and neighbor (F(4)=3.63, p<0.05). The main effect of neighbor was due to higher representational similarity between 1 st neighbor fingers (N1) compared to 2nd, 3rd, and 4th neighbor fingers (N2-N4) across age groups. This was expected, because tactile finger stimulation is expected to excite neighboring fingers more than distant fingers. Critically, the interaction between age and neighbor was due to lower representational similarity in older compared to younger adults' SI maps for N3 (N3-similarity younger: 0.01 ± 0.03, older: −0.11 ± 0.04, p<0.05, Cohen's $d$ = −0.73, see *Figure 5A,B*). N3 representational similarity correlated significantly (and negatively) with age in older but not in younger adults (see *Figure 5—figure supplement 1* for complete statistics, see *Figure 5—figure supplement 2* for results after applying multivariate noise normalization to the data).

To test for local (finger-specific) differences, an ANOVA with the factors digit-pair and age was calculated, which revealed a significant main effect of digit-pair (F(3,102)=2.88, p<0.05), but no main effect of age (F(1,34)=0.0049, p=0.94, Cohen's $d$ = −0.19), and no interaction between age and digit-pair (F(3,102)=1.20, p=0.31). The significant effect of digit-pair was due to lower representational similarity between D2 and D3 compared to D4 and D5 (*t*(35)=2.18, p<0.05), and lower representational similarity between D3 and D4 compared to D4 and D5 (*t*(35)=2.96, p<0.05) across age groups.

In addition, resting state data were used to investigate whether the distance-mediated differences in representational similarity between younger and older adults (see above) are accompanied by differences in slow frequency fluctuations during rest (*Kuehn et al., 2017a*). This could be indicated via decreased functional connectivity between N3 fingers in older adults. Cross-correlation analyses revealed the highest correlations between time series using zero-lag correlations (tested were all possible lags between −130 and +130 TRs). This was true for all possible finger combinations. An ANOVA with the factors neighbor and age on zero-lag cross-correlation coefficients revealed no main effect of neighbor (F(4,88)=1.50, p=0.21), no main effect of age (F(1,22)=2.77, p=0.11), and no interaction between age and neighbor (F(4)=0.73, p=0.57). An ANOVA with the factors finger and age revealed no main effect of finger (F(4,88)=0.66, p=0.62), no main effect of age (F(1,22)=2.43, p=0.13), and no interaction between age and finger (F(4)=0.46, p=0.76, see *Figure 5C,D*). Note that due to technical problems during physiological data recording (see Materials and methods), only n = 12 younger and n = 12 older adults were included in the resting state analyses.

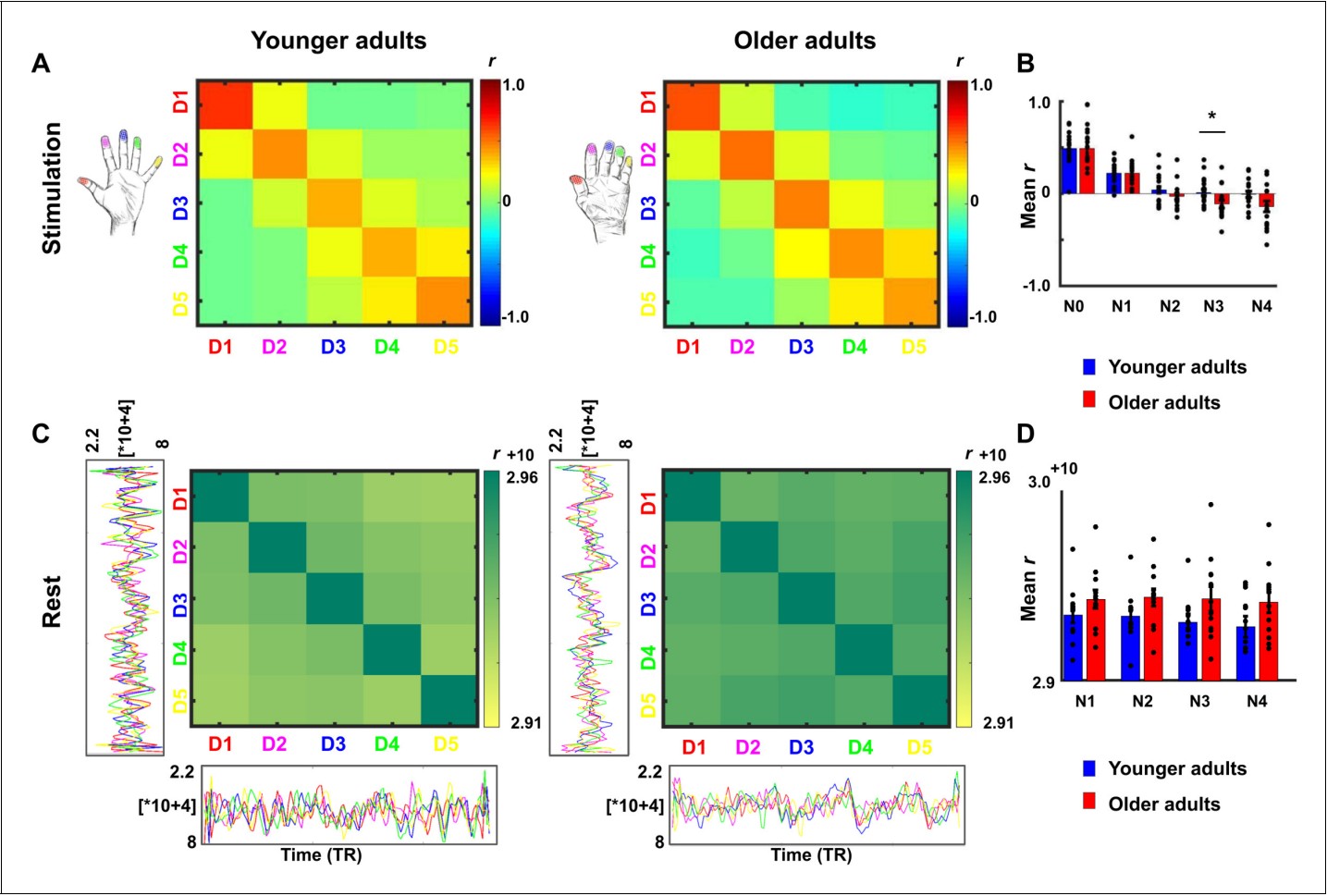

**Figure 5.** Lower representational similarity between distant finger representations in older adults. (**A**) Between-run representational similarity matrices of finger representations in younger and older adults. Higher values indicate higher representational similarity in area 3b. (**B**) Mean representational similarity between same fingers (N0), 1st neighbor fingers (N1), 2nd neighbor fingers (N2), 3rd neighbor fingers (N3), and 4th neighbor fingers (N4) (mean ± SE and individual data). For correlations between N3 representational similarity and individual age, see *Figure 5—figure supplement 1*. For results after applying multivariate noise normalization to the data, see *Figure 5—figure supplement 2*. (**C**) Cross-correlations between finger-specific time series of resting state data. On the x- and y-axes, exemplary finger-specific time series for one younger adult and one older adult are shown. TR = Repetition time, where each TR represents one volume. Note that the diagonal shows autocorrelations between resting state time series. (**D**) Mean cross-correlation coefficients of resting state data between N1-N4 in younger and older adults (mean ± SE and individual data). Shown are data of n = 19 younger adults and n = 17 older adults (**A,B**) and data of n = 12 younger adults and n = 12 older adults (**C,D**).

The online version of this article includes the following figure supplement(s) for figure 5:

**Figure supplement 1.** Correlations between fMRI variables and individual age.

**Figure supplement 2.** Comparison between representational similarity between distant finger representations in younger and older adults, before and after multivariate noise normalization.

## Mislocalizations reflect representational similarity of topographic fields

We used a behavioral finger mislocalization task to test whether the above described functional markers of cortical aging in area 3b have perceptual correlates. For this purpose, we used a perceptual task that is expected to reflect individual differences in topographic map architecture (*Schweizer et al., 2001*; *Schweizer et al., 2000*). During the task, participants were touched at the fingertips of their right hand at their individual 50%-threshold (see *Figure 2—figure supplement 1A* for individual tactile detection thresholds), and were asked to name the location of finger touch in a five-choice-forced-response paradigm (possible answers were 'thumb', 'index finger', 'middle finger', 'ring finger', or 'small finger'). Mislocalizations (i.e. errors where participants assigned touch to another finger than the one that was stimulated) are the variable of interest in this task, because

mislocalizations are assumed to be driven by overlapping and/or more similar representations in SI that cause perceptual confusion (*Pilz et al., 2004*). In total, the applied stimulation resulted in 41.10% of mislocalizations across all fingers and groups, which was expected due to the 50%-threshold that was applied during stimulation.

We first tested whether the distribution of mislocalizations followed the expected pattern of *higher than chance* mislocalizations to *adjacent* fingers and *lower than chance* mislocalizations to *distant* fingers. This pattern is expected if the task reflects the adjacency of cortical representations (*Schweizer et al., 2000*). For younger adults, more mislocalizations than expected by chance were detected at N1, N2, and N3, whereas less mislocalizations than expected by chance were detected at N4. For older adults, more mislocalizations than expected by chance were detected at N1 and N2, and less mislocalizations than expected by chance at N3 and N4 (see *Figure 6C* and *Figure 6—source data 1* for complete statistics). The comparison of the measured distribution of mislocalization with the proportional distribution as expected by chance showed a significant difference for both age groups (younger: G(3)=9.33, p<0.05; older: G(3)=43.59, p<0.001). There was a trend toward older adults showing in total more mislocalizations compared to younger adults (older: M = 0.45 ± 0.03, younger: M = 0.38 ± 0.03, t(48)=1.69, p=0.097, d = 0.48, see *Figure 6—figure supplement 1*).

We then tested whether the above identified age-related difference in global functional map architecture (i.e. less representational similarity between N3-fingers, see *Figure 5*) present with a perceptual correlate. For this aim, we computed an ANOVA with the factors neighbor and age on relative mislocalizations (in %). There was a main effect of neighbor (F(2.26,108.34) = 108.30, p<0.001), no significant main effect of age (p=1), and a trend toward a significant interaction between neighbor and age (F(2.26,108.34) = 2.50, p=0.08). The main effect of neighbor was due to significantly more mislocalizations to N1 compared to N2 (t(83.54) = 5.82, p<0.001), to N1 compared to N3 (t(98)=11.64, p<0.001), to N1 compared to N4 (t(79.97) = 17.54, p<0.001), to N2 compared to N3 (t(98)=7.66, p<0.001), to N2 compared to N4 (t(98)=15.40, p<0.001), and to N3 compared to N4 (t(98)=7.03, p<0.001) across age groups. This is expected based on the higher amount of mislocalizations to nearby compared to distant fingers, as outlined above. The trend towards a significant interaction between neighbor and age was due to older participants showing less mislocalizations to N3 compared to younger participants (older: 0.15 ± 0.01, younger: 0.21 ± 0.02, t(48)=-2.52, p<0.05, see *Figure 6D*). Less representational similarity between N3-fingers in older adults as identified using 7T-fMRI was therefore accompanied by less perceptual digit confusion between N3-fingers as tested behaviorally in the same participants, but on a separate testing day (note that the significant post hoc comparison was based on a trend towards an interaction between neighbor and age).

## Mislocalizations reflect adjacency of cortical representations

We then tested whether the above identified age-related differences in local finger-specific map architecture (i.e. less cortical distance between D2 and D3, see *Figure 3*) present with a perceptual correlate. For this aim, we computed a robust ANOVA with the factors digit and age for each of the five stimulated fingers on the relative distribution of mislocalizations (in %). We found a main effect of digit that was due to more mislocalizations to the respective neighboring digit/s, as outlined above. Importantly, we also found a significant interaction between age and digit for D2 only (F (3,22.02) = 4.84, p<0.05). Post hoc tests revealed that older adults showed higher percentages of mislocalizations from D2 to D3 compared to younger adults (older: M = 41.09 ± 5.75, younger: M = 17.56 ± 4.19, t(27.73) = 3.96, p<0.001, d = 0.94, see *Figure 6E*, see *Figure 6—source data 2* for complete statistics). Reduced local cortical distances between the representations of D2 and D3 in older adults, as identified using 7T-fMRI, was therefore accompanied with more perceptual confusion between D2 and D3 as tested behaviorally in the same participants, but on a separate testing day.

To investigate whether the above-described age-related local differences in perceptual finger confusion were due to finger-specific differences in sensitivity or bias (for example, lower sensitivity or higher bias in D2 or D3 in older compared to younger adults), we applied signal detection theory and determined d' and bias by calculating the amount of times a specific finger was touched but not detected (miss), was touched and detected (hit), was not touched but falsely detected (false alarm), or was not touched and not detected (correct rejection). We calculated an ANOVA with the factors

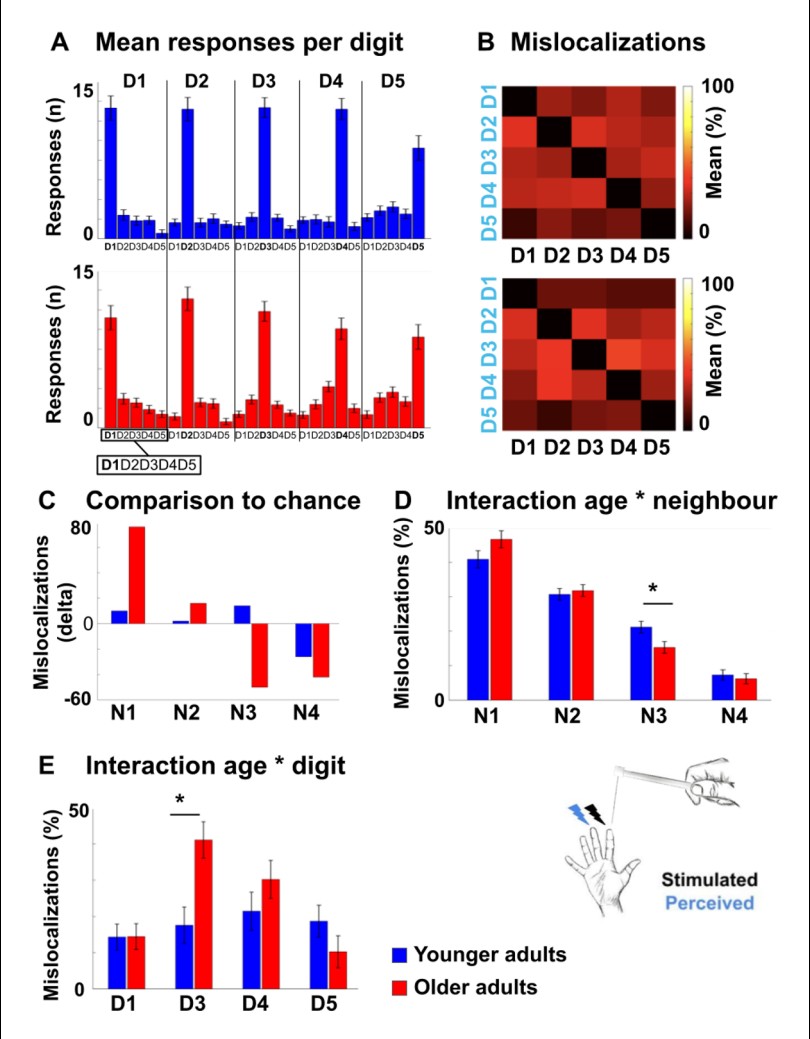

**Figure 6.** Mislocalizations reflect representational similarity of topographic fields and adjacency of cortical representations. (A,B) Finger-specific responses to digit stimulation in younger and older adults shown as numbers of responses per digit dependent on stimulated digit (stimulated digit shown at *top row*, **A**) and as relative distribution of mislocalizations (stimulated digit shown on x-axis, perceived digit shown on y-axis, **B**) (mean ± SE). See *Figure 6—figure supplement 1* for main effects, hit rates and response biases. (**C**) Difference values between the measured distribution of mislocalizations and the distribution as expected by chance for both age groups. The mislocalizations were summed for the fingers according to their distance to the stimulated finger. Raw values are shown in *Figure 6—figure supplement 1*. (**D**) Distribution of mislocalizations dependent on neighbor (N1-N4) and age group (mean ± SE and individual data). (**E**) Distribution of mislocalizations for D2 stimulations to each digit (mean ± SE and individual data). For complete statistics for each digit (D1-D5) see *Figure 6—source data 2*. * indicates significant difference at post hoc *t*-test at an alpha level of p<0.05. Shown are data of n = 25 younger adults and n = 25 older adults.

The online version of this article includes the following source data and figure supplement(s) for figure 6:

**Source data 1.** Tactile misclocalizations compared to chance level.
**Source data 2.** ANOVA results of finger-specific mislocalizations.
**Figure supplement 1.** Hit rates and biases in tactile mislocalization task.

age and digit on d' and a robust ANOVA with the same factors on bias. For d', there was no main effect of digit (F(4,192)=1.16, p=0.33), a trend towards a main effect of age (F(1,48)=3.37, p=0.073), with older adults showing lower d' compared to younger adults (older: M = 1.36 ± 0.12, younger: M = 1.67 ± 0.12, d = −0.52), but, critically, there was no interaction between age and digit (F(4,192) =0.45, p=0.77, see *Figure 6—figure supplement 1*).

For bias, there was a main effect of digit (F(4, 21.69)=3.54, p<0.05), which was due to lower bias for D2 compared to D1 (*t*(48.78) = 2.46, p<0.05), for D2 compared to D5 (*t*(39.41) = 3.68, p<0.001) and for D3 compared to D5 (*t*(35.06) = 2.93, p<0.01) across age groups, no main effect of age (F (1,24.66) = 0.67, p=0.42, *d* = 0.50), and, critically, no interaction between digit and age (F (4,21.69) = 0.91, p=0.47, see *Figure 6—figure supplement 1*). The specific local difference in perceptual confusion between D2 and D3 in older adults can therefore likely not be assigned to finger- and age-specific differences in d' or bias.

## Impaired motor behavior in older adults

Besides characterizing older and younger adults in local and global functional map features and perceptual digit confusion, we also tested them in three tasks that assessed individual differences in motor performance. These tests required coordinated finger movements via the precision grip (Pegboard Tests), and haptic object recognition via coordinated finger movements (O'Connor Dexterity Test). They were employed to answer the critical question of how the observed age-related differences in cortical map architecture and their perceptual correlates are relevant for different aspects of everyday hand use. The tests required participants to quickly move small round (Purdue Pegboard Test) or grooved (Grooved Pegboard Test) pins into corresponding holes, or to quickly identify and pick up three small needles at a time from a hole containing many needles, and to place them into a small hole (O'Connor Dexterity Test). These tests are successful in predicting skills relevant for everyday life, such as picking up and placing small parts, and are standard measures in clinical practice to detect deteriorated movement skills such as in increasing age or in neurodegenerative diseases (*Carment et al., 2018*; *Darweesh et al., 2017*; *Feys et al., 2017*). As expected, motor performance was significantly worse in older participants compared to younger participants in all three tests: Older adults were slower than younger adults to complete the Purdue Pegboard Test (older: 79.20s ± 3.00s, younger: 59.84s ± 1.49s, *t*(35.18) = 5.77, p<0.001, *d* = 1.51), older adults were slower than younger adults to complete the Grooved Pegboard Test (older: 86.80s ± 2.47s, younger: 63.84s ± 1.47s, *t*(39.07) = 7.98, p<0.001, *d* = 2.26), and older adults completed less holes within the given time interval compared to younger adults in the O'Connor Dexterity Test (older: 26.73 ± 1.55, younger: 36.17 ± 1.63, *t*(44)=-4.18, p<0.001, *d* = 1.23).

## Relation between functional map architectures and behavioral phenotypes

Individual differences in motor performance in the group of older adults allowed us to ask whether observed age-related changes in topographic map architecture and their perceptual correlates related to better or worse motor performance in everyday life. This question was evaluated using factor analyses. The aim of factor analysis is to explain the outcome of *n* variables in the data matrix *X* using fewer variables, the so-called *factors*. In order to understand common variances between the above explained differences in hand dexterity and functional as well as perceptual map features that were shown to differ between age groups, we fitted a model to the data matrix *X* consisting of the following variables: Motor performance (Purdue Pegboard Test, Grooved Pegboard Test, O'Connor Dexterity Test), perceptual digit confusion (D2-D3 confusion, N3 confusion), cortical distance (Euclidean distance D2-D3, geodesic distance D2-D3), representational similarity (N3-finger), pRF size (main effect across fingers), and dice coefficients (main effect across fingers). Note that we only included fMRI and perceptual features into the model that differed between age groups. Together with the iterative exclusion of variables (*Maskey et al., 2018*, see Materials and methods), this allowed us to control for the problem of dimensionality given the relatively low sample size.

The two-factor model (mean psi = 0.46) loaded performance in the O'Connor Dexterity Test, perceptual digit confusion (D2-D3), and cortical distance (D2-D3, both Euclidean and geodesic) onto *factor 1*, and performance in the Purdue Pegboard Test, performance in the Grooved Pegboard Test, perceptual digit confusion (N3), pRF size and dice coefficients onto *factor 2*. This overall picture remained the same when we fitted a three-factor model, with the difference that representational similarity was identified as a separate factor (see *Figure 7*). The models therefore separated perceptual and functional variables into local effects (D2-D3 cortical distance, D2-D3 perceptual confusion, *factor 1*) and into global effects (N3 representational similarity, N3 digit confusion, pRF size and dice coefficients, *factor 2* and *factor 3*, respectively). Whereas performance in the O'Connor Dexterity

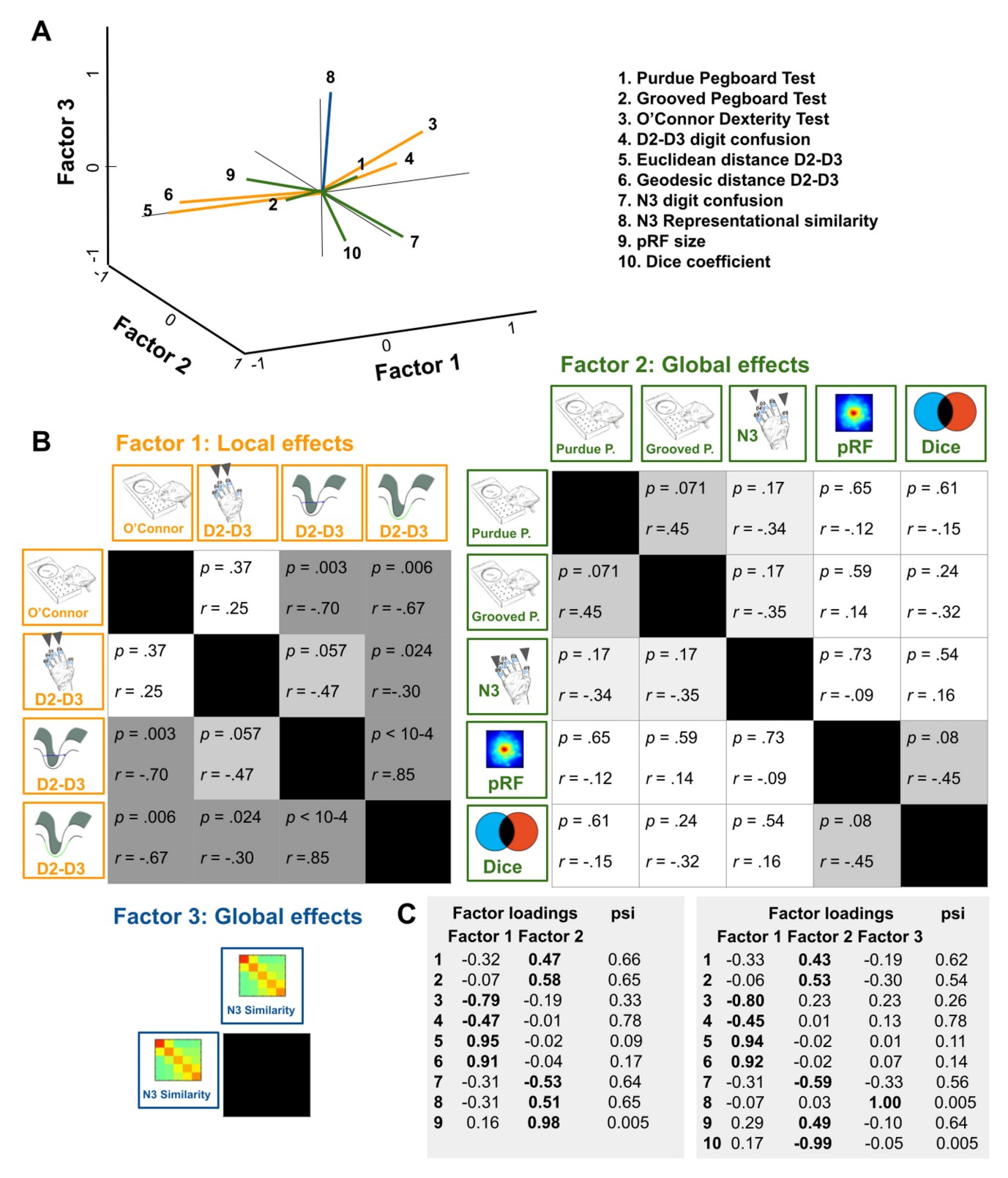

**Figure 7.** Relation between functional map architectures and behavioral phenotypes. (A,B) With a three-factorial model, the functional data were categorized into features that link to local effects, that is that cover parts of the map (perceptual confusion between D2 and D3, cortical distance between D2 and D3, *factor 1, orange lines*) and those that link to global effects, that is that affect the whole map area (representational similarity to N3, pRF size, dice coefficient, *factor 2, green lines,* and perceptual confusion to N3, *factor 3, blue line*). Whereas motor performance in the O'Connor

*Figure 7 continued on next page*

*Figure 7 continued*
dexterity Test loaded positively onto age-related local effects (higher values in O'Connor test, which reflect better performance, relate to higher perceptual confusion between D2 and D3, and lower cortical distance between D2 and D3), performance in the Pegboard Test loaded negatively onto global effects (higher values in Purdue and Grooved Pegboard Test, which reflect worse performance, relate to lower perceptual confusion between N3). (A) factors and visualization of three-factor model, (B) p-values and correlation coefficients (*r*) of Pearson correlations, (C) factor loadings and psi values of two- and three-factor models. Used were data of n = 17 older adults.

Test loaded *positively* on age-related *local effects* (more completed holes in O'Connor Dexterity Test for lower cortical distances between D2 and D3, and for higher perceptual confusion between D2 and D3 that both characterize older adults), performance in the Pegboard Test loaded *negatively* on age-related *global effects* (more time spent on Purdue Pegboard Test and more time spent on Grooved Pegboard Test for lower perceptual confusion to N3 that characterize older adults). Correlation analyses further show that reduced distances between the representations of D2 and D3 relate to better motor performance as revealed in the O'Connor dexterity test (see *Figure 7B*). In supplementary analyses, it was confirmed that pRF size did not correlate with cortical distance measures or motor performance (see *Figure 4—figure supplement 2*).

## Post hoc analyses: tactile spatial discrimination thresholds do not correlate with individual pRF size

In the course of the revision process, we computed additional analyses incorporating participants' spatial acuity performance of right D2 as assessed by a tactile two-point discrimination task (2PDT) to test whether larger pRF sizes (that do not correlate with behavior, see above) may relate to worse tactile spatial acuity (*Kalisch et al., 2009*; *Lenz et al., 2012*; *Peters et al., 2009*). The task required participants to spatially distinguish two rounded pins that were simultaneously applied to the skin surface of the fingertip. Participants were asked after each trial whether they perceived one or two pins. The two-point discrimination threshold was defined as the distance (in mm) between the two pins at which they were perceived 50% of the time two single stimuli rather than one. In clinical and scientific settings, the tactile 2PDT is broadly used to assess deteriorated tactile functioning such as in older age (e.g. *Bowden and McNulty, 2013*; *Desrosiers et al., 1999*; *Kalisch et al., 2009*; *Lenz et al., 2012*; *Ranganathan et al., 2001*; *Vieira et al., 2016*) or hand injuries and disorders (e.g. *Kus et al., 2017*; *Van Boven and Johnson, 1994*). As expected, older adults showed higher two-point discrimination thresholds than younger adults (older: 3.17 mm ±0.85 mm, younger: 1.78 ± 0.29 mm, $t(19.52) = 6.14$, $p<0.00001$, $d = 2.10$). However, there was no significant correlation between 2PDT thresholds of D2 and mean pRF sizes of D2 neither for younger adults nor for older adults (n = 16 younger: $r = -0.36$, $p=0.895$, n = 16 older: $r = 0.153$ $p=0.597$, see *Figure 4—figure supplement 2C*).

Finally, we addressed the question of whether impaired spatial acuity performance in tactile two-point discrimination is related to impaired motor performance. Based on previous research (*Kalisch et al., 2008*), we expected a relationship between spatial acuity and the Pegboard Task that would confirm its reliance on perceptual acuity required for the precision grip. There was a significant positive correlation between performance in the two-point discrimination task and performance in the Grooved Pegboard task in older (but not younger) adults (older: $r = 0.68$, $p<.005$, younger: $r = 0.01$, $p=0.983$), indicating that older adults who were slower in completing the Grooved Pegboard task performed worse (i.e. showed elevated thresholds) in the two-point discrimination task. There was no significant correlation between performance in the two-point discrimination task and performance in the other two motor tests (Purdue Pegboard Test: older: $r = 0.35$, $p=0.168$; younger: $r = -0.36$, $p=0.145$; O'Connor Finger Dexterity Test: older: $r = 0.01$, $p=0.982$; younger: $r = 0.26$, $p=0.312$).

## Discussion

Here, we used a combination of ultra-high-resolution functional magnetic resonance imaging, computational modeling, psychophysics, and everyday task assessments to detect and describe 'de-differentiated' cortical maps in primary somatosensory cortex (SI) and their association to functional readouts and everyday behavior. Older adults are an ideal population to study mechanisms of

cortical de-differentiation, because their topographic map architecture is assumed to become 'less precise' with increasing age, which has been related to maladaptive behavior (*Cabeza, 2002*; *Cassady et al., 2020*; *Dennis and Cabeza, 2011*; *Heuninckx et al., 2008*; *Mattay et al., 2002*; *Reuter-Lorenz and Lustig, 2005*; *Riecker et al., 2006*).

We did not detect significant differences in basic map statistics such as topographic map size and topographic map amplitude between younger and older adults' SI maps when stimulation amplitude was adapted for each finger of each individual. Rather, at the global level, we observed slightly more aligned topographic maps in the group of older adults (i.e. slightly lower dispersion when aligning the vectors of participants' Fourier maps within each age group), larger pRF sizes, more overlap between neighboring finger representations, and lower representational similarity between distant fingers in older adults. At the local level, we observed reduced cortical distances between the area 3b representations of the index finger and the middle finger in older adults. Some local and global functional map changes reflected the pattern of perceptual finger confusion, because older adults showed lower perceptual confusion between distant fingers (aligning with the lower representational similarity between them) and higher perceptual confusion between the index and the middle finger (aligning with the lower cortical distance between them) compared to younger adults. Because the latter result correlated with better performance in a motor task relying on haptic exploration, local cortical 'de-differentiation' (i.e. representations that are closer to each other) was here related to better motor performance in everyday life. These results are in three respects novel and even surprising.

Age-related differences in receptive field sizes in rats are restricted to the hindpaw representation and do not occur in the forepaw representation of SI, with more intensive use of the forepaw compared to the hindpaw assumed to be the underlying reason (*David-Jürgens et al., 2008*; *Godde et al., 2002*). In older rats, receptive fields in the hindpaw representation are less inhibitory and larger than receptive fields in younger rats (*David-Jürgens et al., 2008*; *Godde et al., 2002*; *Spengler et al., 1995*), which relates to worse walking behavior (*Godde et al., 2002*). This topographic pattern could not be replicated for humans in the present study, because we found larger pRF sizes in the hand area of older compared to younger participants, which does not corroborate the above described hindpaw-selective changes in rats. In contrast to rats, we also do not see a significant correlation between larger pRF sizes and worse motor control of the hand, neither for motor tasks that rely on precision grips nor for a motor task that relies on haptic exploration. We also do not see a significant correlation between pRF size and spatial tactile acuity at the fingertip. The behavioral relevance of increased pRF size in older humans therefore remains to be investigated.

We also do not see an increase in topographic map amplitude or topographic map size in older adults when the stimulation amplitude was adjusted to each finger of each individual. Similarly, we do not see an increased distance between the representations of the index finger and the small finger in older adults, as has been reported before. This does not corroborate the idea of a larger and 'overactivated' map in older adults. We did observe that older adults' maps show significantly higher response amplitudes compared to younger adults' SI maps when all fingers are stimulated together at a fixed amplitude. Other studies, however, have found amplitude differences even if stimulation intensity was adjusted to the individual threshold (e.g. 2.5 times above sensory threshold, *Pleger et al., 2016*). This difference may be due to our stimulation paradigm where only two pins stimulated the skin at the same time, with randomly changing pins every 62 ms, reducing age-related differences in neuronal adaptation to influence the results. We also observed that topographic maps within the group of older adults were more aligned to each other than the topographic maps in younger adults. This questions the view that older adults' topographic maps show higher degrees of stimulus-related noise, or are more disorganized. However, our data are in line with the assumption that changes in the inhibitory architecture in older adults' SI explain larger somatosensory representations in older adults (*Lenz et al., 2012*; *Pleger et al., 2016*). This view is supported by our finding that older adults have larger pRF sizes, greater overlap between neighboring finger representations, and show higher response amplitudes when all fingers are stimulated together; the latter could be due to decreased between-digit inhibition in older adults. Furthermore, decreased inhibition may also be one reason why the topographic maps within the group of older adults seem more consistent; this may be due to the less sharp representations of the fingers in SI.

The second finding that was not expected based on prior evidence is the reduced rather than enhanced representational similarity between distant finger representations in older compared to

younger participants. The presumably more de-differentiated cortical maps were in fact less de-differentiated with respect to distant finger representations. It was even more astonishing that this effect related to a behavioral phenotype, because the fingers that showed less representational similarity in older adults (i.e. third-neighbor fingers) were also mixed up less perceptually (note, however, that we did not find a significant correlation between these two measures). So far, research has stressed that older adults' topographic maps are characterized by less rather than more distinct map organization (*Cassady et al., 2020*; *Pleger et al., 2016*). However, subcortical U-fibers, which are located within the cortex or in the very outer parts of the subcortical white matter, particularly connect adjacent but not directly neighboring representations in the cortex. Short-association fibers are among the last parts of the brain to myelinate, and have very slow myelin turnover (*Reiser et al., 2007*). Myelination and protracted maturation of short-association fibers can continue until the age of fifty (*Wu et al., 2016*). Fully myelinated U-fibers in older adults and less myelinated U-fibers in younger adults may explain higher correlated short-distance representations in older adults, and good performance in discriminating non-neighboring signals. This finding hints toward potentially better 'abstract encoding' in older adults, here defined as the distinct extraction of information from adjacent but not neighboring topographic units.

Perhaps the most surprising finding of our study, however, is the reduced local cortical distances between the index and middle finger representations in older adults with preserved map size, where, previously, greater distances between fingers and larger map sizes were reported (*Kalisch et al., 2009*). Whereas the reduced cortical distance between neighboring fingers is in principle in line with a de-differentiation model of topographic map architecture, its relation to better rather than worse motor control is certainly not. In this respect, the distinction between functional readouts that capture integration versus separation seems relevant (see *Figure 1*). Specifically, we found that reduced cortical distances between D2 and D3 representations related to worse tactile discrimination between these two fingers (i.e. worse separation in a tactile task) but to better motor control in a task that required haptic object exploration and coordinated finger movements (i.e. improved integration in motor task). These findings highlight the importance of assessing different aspects of signal integration and signal separation to investigate the impact of functional map changes on everyday life. Because this relationship was only seen for the index and middle finger and not for the other fingers, also the factor of spatial extent (i.e. local versus global, see *Figure 1*) is relevant for comprehensive data analysis.

Interestingly, the reduced cortical distance between index and middle finger representations in older adults may explain the increased motor enslaving of the middle finger during index finger flexion, and the lower range of independent movements specifically of the index finger in older adults (*Van Beek et al., 2019*). This can be assumed because passive stimulation of the fingers also activates finger maps in the primary motor cortex (see *Figure 2*). Even though the interaction between tactile maps and motor maps warrants final clarification (e.g. *Kuehn and Pleger, 2020*), an interaction between both maps can be assumed in older adults (*Cassady et al., 2020*). Another interesting aspect is that the reduced cortical distances between index and middle finger representations in older adults may be explained by use-dependent plasticity (*Makin et al., 2013a*). During everyday hand movements, the index and the middle finger correlate less with each other than other neighboring fingers (*Belić and Faisal, 2015*), and the index finger is the most independent of the four fingers (*Ingram et al., 2008*). During tactile learning, however, plasticity transfers more from the middle finger to the index finger than from the middle finger to the ring finger (*Dempsey-Jones et al., 2016*), which indicate their interaction. The observed local map changes may be induced by the correlated input of the middle and ring finger or ring finger and small finger (*Kolasinski et al., 2016b*), or by age-related changes in the local myeloarchitecture that link to functional map topography (*Carey et al., 2018*; *Kuehn et al., 2017b*). Finally, because the reduced cortical distances between D2 and D3 seems to have a beneficial relationship to motor control, also the increased coupling of D2 and D3 during haptic exploration may explain the observed effect. In either way, a positive relationship between increased cortical de-differentiation and impairments in everyday hand use is not supported by our data.

Also the observed age-related differences in representational similarity between distant finger representations can be discussed in the light of use-dependent versus age-dependent plasticity. One way to explain less representational similarity between distant finger representations in older adults is to assume a greater clustering of D1, D2 and D3 and/or of D4 and D5 with increased hand

use. This could explain why the similarity between the representations of D1 and D4 and between D2 and D5 is lower in older adults. However, we do not see an interaction between finger and age in representational similarity, resting state correlations, or dice coefficients, which would be expected if a finger-specific clustering of D1 and D2 or D4 and D5 existed. Rather, we observe a specific shift of D2 and D3 toward each other, an effect that positively relates to hand dexterity. However, this shift does not correlate with the reduced representational similarity between distant fingers within the group of older adults. A direct relationship between these two measures can therefore not be established based on our data. On the other hand, we observed a significant correlation between N3 representational similarity and individual age in older adults, which hints towards a potential critical role of the factor age for the development of reduced similarity between distant finger pairs, and could be established via U-fiber maturation, as discussed above. Nevertheless, because we do not have information about hand use patterns of our participants, the effect of hand use on our data cannot be specifically investigated here.

Taken together, both in the case of cortical distance, where more local 'de-differentiation' related to better hand dexterity, and in the case of representational similarity, where presumably more 'de-differentiated' cortical maps showed less representations similarity at distant locations, a simple de-differentiation model of cortex function does not seem to appropriately reflect the empirical data. In our view, topographic maps should not be classified according to their 'de-differentiation-level' but according to specific map features that take into account spatial extent (global versus local map changes) and functional readout (integration versus separation, see *Figure 1*). Even though not exhaustive, this distinction facilitates the precise investigation of how specific map features relate to corresponding functional readouts such as cortical integration or cortical separation in sensory or motor tasks. For example, whereas reduced cortical distances may increase the local integration of cortical signals, this may benefit tasks that require coordinated finger movements but may worsen tasks that require finger individuation. The distinction between local and global map features is particularly relevant when distinguishing between use-dependent and age-dependent topographic map plasticity, and between adaptive versus maladaptive plasticity. In our data, the local shifts of the index and middle finger representations towards each other seem to have adaptive consequences, whereas maladaptive consequences for global changes in pRF sizes were not identified. However, future research should specifically investigate the effect of use-dependent plasticity on older adults SI maps (an aspect that was not investigated here) to dissociate between adaptive and maladaptive plasticity.

This approach sheds new light on future interventions and training paradigms that aim at speeding up, slowing down, or reversing neuroplastic processes in the cortex. Repeated sensory stimulation of the skin induces NMDA-dependent Hebbian plasticity at the corresponding cortical territory, a mechanism that improves local spatial discrimination thresholds (*Dinse et al., 2003*; *Kuehn et al., 2017a*). Synchronous stimulation of more than one finger, but also glueing of multiple fingers, has been used to induce neuroplastic processes of topographic map architectures (*Kalisch et al., 2008*; *Kolasinski et al., 2016b*). Synchronous stimulation of all five fingers causes less mislocalizations to nearby digits and more mislocalizations to distant digits (*Kalisch et al., 2008*; *Kalisch et al., 2007*), whereas temporal gluing of the index finger to the middle finger induces a shift of the ring finger toward the small finger, accompanied by less cortical overlap between the middle and the ring finger. This intervention also caused lower thresholds in temporal order judgments between the middle finger and the ring finger, and higher thresholds in temporal order judgments between the ring finger and the small finger (*Kolasinski et al., 2016b*). Integrating previous knowledge with our data leads to the assumption that concurrent stimulation of *distant* but not neighboring topographic units and/or correlated input to topographic units that neighbor the affected ones (here the *ring finger and small finger* instead of the index finger and the middle finger) may be particularly beneficial to induce adaptive neuroplasticity in aging topographic maps. This principle can be applied to other cases of distorted (*Saadon-Grosman et al., 2015*) or preserved (*Makin et al., 2013b*) map architectures in clinical cases. At a more abstract level, these data indicate that a precise characterization of local and global map changes and their relation to mechanisms of signal integration and separation is a prerequisite for the development of beneficial and individualized training strategies that aim at stopping or reversing maladaptive topographic map change.

One limitation of this study is the relatively low sample size, where MRI data of n = 17 older and n = 19 younger adults are presented. This limitation is due to current procedures and restrictions in

ultra-high field scanning, and their stringent exclusion criteria. It has the consequence that both type I and type II errors are more difficult to reduce (*Sullivan et al., 2016*). To reduce type I error, we performed corrections for multiple comparisons and reduced the number of performed statistical tests to the minimum number needed to test our hypotheses. To reduce type II errors, we report and follow-up statistical trends (i.e. $p>0.5$ and $<0.1$), for example for the interaction between age and neighbor on perceptual confusion.

In summary, we here provide a comprehensive description of 'de-differentiated' topographic maps in SI, and detail the topographic map features that relate to cortical aging. Our data may inspire future research on cortical plasticity, and may motivate the distinction between local and global changes of the map area in relation to functional readout that may either benefit integration or separation of neuronal representations. Future neuroimaging studies with larger cohorts will uncover global topographic map features and their relation to behaviorally relevant neuroplasticity.

## Materials and methods

### Participants

We tested n = 25 younger adults (mean age 25 ± 0.49, ranging from 21 to 29 years, 13 male and 12 female) and n = 25 older adults (mean age 72.2 ± 0.81, ranging from 65 to 78 years, 13 male and 12 female) for sensorimotor behavior at the right hand (sample size for touch thresholds, tactile mislocalization and pegboard test based on *Kalisch et al., 2008*, sample size for topographic shift based on *Kalisch et al., 2009*). According to the Edinburgh handedness questionnaire (*Oldfield, 1971*), all participants were right-handed (laterality index ranging from +40 to+100, M = 84.73 ± 18.18). Participants were recruited from the database of the DZNE Magdeburg. Due to the strict exclusion criteria for 7T-MR measurements (see below), participant recruitment and testing took 4 years in total (2016–2020). The Montreal Cognitive Assessment (MOCA) was used as a screening tool to assess the possibility of mild cognitive dysfunction amongst participants. Inclusion criteria were (i) no medication that influenced the central nervous system, (ii) intact hand function (sensory and motor), (iii) 7T-MRI compatibility (see below), and (iv) no sign of early dementia (note that n = 1 older adult had a MOCA score of 21; he showed good performance in all tests and was included in the analyses). The MOCA score of the other participants ranged between 25 and 30 (M = 28.44 ± 0.25).

We reinvited participants for one 3T-MRI session and one 7T-MRI session. Before the behavioral tests, participants were already screened for 7T-MRI exclusion criteria such as metallic implants and other foreign bodies, active implants (e.g. pacemaker, neurostimulators, cochlear implants, defibrillators, and pump system), permanent makeup, tinnitus, or hearing impairments. Due to changes in health conditions between the behavioral and MR-measurements, and/or due to stricter MR-regulations due to COVID-19 that were implemented in March 2020, we could reinvite n = 20 younger adults and n = 18 older adults of the original cohort for the MRI measurements. For n = 1 younger adult and n = 1 older adult, the 7T-MRI session could not be completed successfully. Therefore, MR analyses are presented for n = 19 younger adults (10 female, nine male, mean age: 24.89 years), and n = 17 older adults (eight female, nine male, mean age: 69.12 years). All participants were paid for their attendance and written informed consent was received from all participants before starting the experiment. The study was approved by the Ethics committee of the Otto-von-Guericke University Magdeburg.

### General procedure

Participants were invited to four appointments. There were two appointments for behavioral tests (one for digit confusion, detection thresholds, and hand dexterity, and one for the two-point-discrimination task), one appointment for a 7T-fMRI session where all fMRI data were acquired, and one appointment for a 3T-MRI session where a T1-based image was acquired used for cortex segmentation (see *Figure 1* for an overview of experimental design, see *Figure 1—figure supplement 1* for an overview of analyses pipelines).

### Digit confusion

The behavioral tests took place on the first testing day. To estimate perceptual digit confusion, a tactile finger mislocalization task was used (*Schweizer et al., 2001*; *Schweizer et al., 2000*) that is

assumed to reflect SI map topography (*Kalisch et al., 2008*; *Kalisch et al., 2007*; *Schweizer et al., 2001*). First, the detection threshold of each finger was estimated. During testing, participants sat on a chair with the hand positioned palm upwards on a foam cushion. The tested hand was occluded from view. Participants heard white noise through headphones during the task. Before the experiment started, five points were marked on the participant's fingertips via a felt-tip pen: one point at the center of the volar surface of the first segment of each digit (D1-D5). The detection threshold was estimated for each finger separately with a two-alternative forced choice task. For each finger, mechanical forces were applied to the marked area of the fingertip using Semmes Weinstein monofilaments (Semmes-Weinstein monofilaments; Baseline R, Fabrication Enterprises Inc, White Plains, NY, USA, applied weights: 0.008 g, 0.02 g, 0.04 g, 0.07 g, 0.16 g, 0.4 g, 0.6 g, 1.0 g, 1.4 g, 2.0 g, 4.0 g, 6.0 g). These calibrated filaments assert the same amount of pressure once the filament is bent. Stimulation duration was 1 s. At each trial, two intervals were presented with only one of them containing a stimulation. Participants were asked to detect the stimulation interval by pressing the respective key on the keyboard in a self-paced manner ('1' or '2'). For stimulus application, the experimenter followed auditory instructions via headphones. Neither the hand nor the experimenter were visible to the participant during testing. A randomized sequence (different for each participant) was used to determine which interval contained the stimulation. The adaptive thresholding procedure followed a 3-down/1-up staircase algorithm. Two such staircases were used in an alternating manner, one started at 0.4 g, the other at 0.02 g. The threshold was estimated if the standard deviation from the mean in stimulus intensity was equal or less than one step (*Gescheider et al., 1996*). This was repeated five times, once per finger, in a randomized sequence. The task took approximately 60 to 75 min.

After a short break, the finger mislocalization task was applied using Semmes Weinstein monofilaments. The stimulation sites were the same as for the tactile detection task (marked area at fingertip, see above). For each finger, the applied force matched the respective tactile detection threshold as assessed before. Therefore, both younger and older adults were stimulated at each finger at their individual tactile detection threshold, controlling for individual and finger-specific variability in tactile sensitivity. Each trial started with a 3 s stimulation interval, where stimulation was applied to one of five possible fingertips. Stimulation duration was 1 s. The beginning and end of this interval were marked by computer-generated tones. In this five-alternative-forced-choice test, participants were provided with 7 s time to verbally name the finger where they felt the touch. Previous studies showed similar tactile misattributions for verbal versus motor responses (*Badde et al., 2019*). This long response interval was chosen to prevent speed-accuracy trade-offs for older compared to younger adults. If participants did not feel touch at none of the fingers (note that touch was applied at individual thresholds and was therefore expected to be perceived in only around 50% of the cases), they were motivated to name their best guess. The next trial started once the experimenter had inserted the response into the computer. Each finger was stimulated 20 times, stimulation order was pseudo-randomized for each participant in a way that there was maximally one repetition in each sequence. All testing was done by one of the authors (A.C.). Because the results of this task are stable across multiple runs (*Schweizer et al., 2000*), all testing was done within one session. The task took approximately 20 min.

## Hand dexterity

Three standard tests were then used to test individual levels of hand motor function (similar to *Kalisch et al., 2008*). The *Purdue Pegboard Test* is composed of two rows of 25 small holes each, and one larger hole at the top that contains 25 small metal pins. The task was to pick one pin at a time with the right hand, and insert them into the holes on the right side of the board from top to bottom. If one of the metal pins dropped during the transfer, participants were instructed to continue with the next one. We measured the time to complete the test (in s), and the number of dropped pins (n). The *Grooved Pegboard Test* is composed of a 5 × 5 matrix of small (grooved) holes, and one larger hole at the top that contains 31 small metal pins. The task was to pick one pin at a time with the right hand, and insert them into the holes from left to right. Other than the Purdue Pegboard Test, this task requires changing the orientation of the pins such that they fit into the grooved holes (shown schematically in *Figure 1*). If one of the metal pins dropped during the transfer, participants were instructed to continue with the next one. We measured the time to complete the test (in s), and the number of dropped pins (n). The *O'Connor Finger Dexterity Test* is the most

difficult test of these three, and is composed of a 10 × 10 matrix of small holes, and one larger hole at the top that contains 315 small, thin metal sticks. Participants were asked to pick three sticks at a time with their right hand, and place all of them into a small hole, starting from left to right. This required orienting the three sticks within one hand in a way that they would fit into the small hole. If one of the sticks dropped during the transfer, they were instructed to again pick three sticks out of the hole. Because there are strong individual and age-related differences in this test, participants were here given 4 min time to fill as many holes as possible. We measured the number of holes that were successfully filled with three metal sticks (n) as well as the number of dropped sticks (n).

## Spatial acuity

A tactile two-point discrimination task was used to assess individual spatial acuity performance of the participant's right D2 (similar to *Kalisch et al., 2008*; *Pleger et al., 2016*). At a separate behavioral testing day, two rounded pins (diameter = 0.4 mm) were simultaneously applied to the skin surface of the fingertip. A fully automatic stimulation device controlled by the commercial software package Presentation (version 16.5, Neurobehavioral Systems, Inc, Albany, CA, USA) moved the pins up and down. Amplitude of pin movement was adjusted to individual detection thresholds of mechanical forces, but was at least set to 1.2 mm. Pin spacing ranged from 0.7 to 2.8 mm (in steps of 0.3 mm) for younger adults and 0.7 to 6.3 mm (in steps of 0.8 mm) for older adults. Additionally, a single pin (control condition) was included. Pin spacing was vertically adjusted by moving a rotatable disc containing all possible conditions (n = 9 altogether). Pin conditions were pseudo-randomly presented in a two-alternative forced-choice task. Participants were asked to judge whether they felt one or two pins touching their fingertip. Only if they were sure of feeling two points of contact they were instructed to answer 'two pins felt'. Decisions were indicated by mouse button press using the left hand. To prevent order effects, unique sequence lists of pin conditions were used per participant and task block. All participants performed two task blocks. Each task block included 90 trials (10 repetitions per pin condition). Intertrial intervals varied between 1 and 5 s (in steps of 1 s) and were pseudo-randomized to avoid fixed clock cycles gating reactions. All participants sat in front of a screen signaling the beginning and ending of a task block. The right D2 was fixated on the stimulator, and the hand was covered by a white box during the task to prevent effects caused by seeing the stimulated finger (*Cardini et al., 2012*; *Cardini et al., 2011*).

## MR sequences

Data was acquired at a whole body 7 Tesla MRI scanner (Siemens Healthcare, Erlangen, Germany) in Magdeburg using a 32 Channel Nova Medical head coil. First, a whole-brain MP2RAGE sequence with the following parameters was acquired: Voxel resolution: 0.7 mm isotropic, 240 slices, FoV read: 224 mm, TR = 4800 ms, TE = 2.01 ms, TI1/2 = 900/2750 ms, GRAPPA 2, sagittal positioning. Shimming was performed prior to collecting the functional data, and two EPIs with opposite phase-encoding (PE) polarity were acquired before the functional scan. The functional EPI sequence (gradient-echo) had the following parameters: Voxel resolution: 1 mm isotropic, FoV read: 192 mm, TR = 2000 ms, TE = 22 ms, GRAPPA 4, interleaved acquisition, 36 slices. The same sequence was used for all functional tasks (see below). 3T-MRI data were acquired at the Philips 3T Achieva dStream MRI scanner, where a standard structural 3D MPRAGE was acquired (resolution: 1.0 mm x 1.0 mm x 1.0 mm, TI = 650 ms, echo spacing = 6.6 ms, TE = 3.93 ms, $\alpha$ = 10°, bandwidth = 130 Hz/pixel, FOV = 256 mm×240 mm, slab thickness = 192 mm, 128 slices).

## Physiological data recording

A pulse oximeter (NONIN Pulse Oxymeter 8600-FO) clipped to the index finger of the participant's left hand was used to measure the pulse during functional scanning at the 7T-MRI scanner. Additionally, participants wore a breathing belt to capture respiration. An in-house developed setup was used to digitally record and analyze the physiological data (hardware employing National Instruments USB 6008 module with pressure sensor Honeywell 40PC001B1A). The sampling frequency was set to 200 Hz. Data acquisition started with the MR trigger of each functional run.

## fMRI task

Five independently-controlled MR-compatible piezoelectric stimulators (Quaerosys, http://www.quaerosys.com) were used to apply tactile stimulation to the five fingertips of the right hand of younger and older adults while lying in the 7T-MRI scanner (*Schweisfurth et al., 2015*; *Schweisfurth et al., 2014*; *Schweisfurth et al., 2011*). One stimulator was attached to the tip of each right finger using a custom-build, metal-free applicator that could be fitted to individual hand and finger sizes. Each stimulator had eight individually-controlled pins arranged in a 2 × 4 array, covering 2.5 × 9 mm$^2$ of skin (see *Figure 1*). Vibrotactile stimulation was applied to the fingertips at a frequency of 16 Hz (*Schweizer et al., 2008*). Stimulation intensity of each subject and each finger was adjusted to 2.5 times the individual tactile detection thresholds. To minimize adaptation-related differences in map activity between younger and older adults, two randomly chosen pins were raised once at a time, yielding 16 pin combinations per second (*Schweisfurth et al., 2015*; *Schweisfurth et al., 2014*; *Schweisfurth et al., 2011*).

Participants first underwent two phase-encoded protocols, and then continued with two blocked-design protocols. The phase-encoded protocols consisted of 2 runs of 20 cycles each. Each cycle lasted 25.6 s, stimulation was applied to each fingertip for 5.12 s, and for 20 times. Stimulation was delivered either in a forward order (D1- > D5) or in a reverse order (D5- > D1, see *Figure 1*). Half of the participants of each age group started with the forward-run, the other half started with the reverse-run. One run comprised 256 scans (512 s for a TR of 2 s), and lasted for 8 min and 31 s. Participants were instructed to covertly count short randomly distributed interrupts embedded in the tactile stimulation (duration 180 ms, slightly longer than in *Schweisfurth et al., 2015*, *Schweisfurth et al., 2014*, *Schweisfurth et al., 2011* to account for the effect of age). There were the same number of gaps in each run (15 gaps in total).

The blocked-design paradigm comprised six conditions: Stimulation to D1, D2, D3, D4, D5, and a rest condition with no stimulation. The same stimulation protocol as in the phase-encoded design was used (each finger was stimulated for 5.12 s, same frequency and stimulation duration). Fingers were here stimulated in a pseudo-random sequence, where there was never one finger stimulated more than two times in a row. In 70% of the trials, there was a 2 s pause between two subsequent stimulations, in 30% of the trials, there was a 6 s pause between two subsequent stimulations. This was counterbalanced across fingers. Each finger was stimulated 10 times. One run comprised 208 scans, and lasted for 6 min and 56 s. The same task was applied as in the phase-encoded paradigm. The blocked-design run was repeated twice. Subsequently, two runs were measured where a one-TR stimulation of all five fingers was followed by a 11-TR rest without any stimulation. This sequence was repeated 10 times for each run, with two runs in total. Finally, we acquired a 5 min resting state scan, where participants were asked to look at a centrally presented fixation cross, and to think about nothing in particular. All functional measurements together took around 40 min.

## Behavioral analyses: digit confusion

Using an adaptive staircase procedure, the detection threshold was estimated if the standard deviation from the mean in stimulus intensity was equal or less than one step (*Gescheider et al., 1996*). These values were transformed logarithmically ($\log_{10}0.1mg$), and were used as stimulation intensities for the mislocalization task. Mislocalizations were defined as responses where participants indicated another finger than the one that was stimulated, that is as false responses. These were analysed with respect to their distribution across the non-stimulated fingers. Mislocalizations were grouped according to their distance to the stimulated finger into 1st neighbor, 2nd neighbor, 3rd neighbor, and 4th neighbor (N1-N4) mislocalizations (e.g. if D2 was stimulated and the participant assigned the touch to D4, this was a N2 mislocalization, if the participant assigned the touch to D5 instead, this was a N3 mislocalization, and so forth, *Schweizer et al., 2000*). No errors for one specific finger were computed as zero values. The resulting distribution of the relative number of mislocalizations towards N1-N4 was compared to the expected equal distribution of mislocalizations for each finger using the *G*-test of goodness of fit (*Sokal and Rohlf, 1981*). An equal distribution is expected if the naming of the localization is at chance level and does not follow the principles of topographic arrangement, where more mislocalizations are expected to closer neighbors compared to distant neighbors (*Schweizer et al., 2001*; *Schweizer et al., 2000*). In this analysis, the different distributions of response options for the different neighbors were taken into account (i.e. the fact that

across all fingers, there are more response options for 1st compared to 2nd, 2nd compared to 3rd, and 3rd compared to 4th neighbor) (*Schweizer et al., 2000*). The *G*-tests of goodness of fit were Holm-Bonferroni corrected at a significance threshold of p<0.05.

To calculate hit rates, for each participant, the number of correctly perceived stimulus locations was accumulated for each finger and divided by the number of stimulations of that finger. This resulted in the proportion of correct responses (hit rates) in percent for each finger. False alarms were defined as the number of times that this same finger was falsely identified as the one being stimulated when actually not touched. This is irrespective of which finger was touched that time (i.e. it could be misclassified when N1, N2, N3 or N4 was touched). For the estimation of *d'*, hits and false alarms were first converted to *z*-scores. The false alarm *z*-scores were then subtracted from the hit rate *z*-scores, and the sensitivity index was obtained for each finger separately. The beta criterion (bias) was further calculated for each finger based on the *z*-scores by estimating the exponential of the difference between the false alarm *z*-scores and the hit rate *z*-scores, each raised to a power of two, and divided by 2. To overcome the problem of missing events, the loglinear transformation was applied to the analyses (*Stanislaw and Todorov, 1999*).

The distribution of mislocalizations (in %) was used for an ANOVA with the factors neighbor (N1-N4) and age (younger, older). To test for finger-specific effects, an ANOVA was calculated for each digit with the factors response digit and age (younger, older). For D1, response digit was specified as D2, D3, D4, D5; for D2, it was specified as D1, D3, D4, D5, and so forth. To estimate whether age-dependent changes in mislocalizations are due to age-related differences in sensitivity and/or bias, two ANOVAs with the factors digit (D1-D5) and age (younger, older) were also conducted for sensitivity and bias. In the case of sensitivity, the sensitivity index *d'* was used as a dependent variable and in the case of bias, the beta criterion was used as a dependent variable. Robust ANOVAs were calculated if the distribution of values in the non-normality distributed data was skewed, ANOVAs were calculated if the data were normally distributed, or if only sub-groups of the data were not normally distributed (*Glass et al., 1972*; *Harwell et al., 1992*; *Lix et al., 1996*). Robust ANOVAs based on trimmed means were computed in R (*R Development Core Team, 2019*) with the statistical package WRS2, developed by Wilcox, and the function 'bwtrim'. Trimmed means are formed after the removal of a specific percentage of scores from the lower and higher end of the score distribution, obtaining thus accurate results even for non symmetrical distributions, by removing outliers and skew (*Field, 2009*). As post hoc tests for the ANOVA, independent-sample *t*-tests were computed, as *post hoc* tests for the robust ANOVA, the Yuen-Welch method for comparing 20% trimmed means was applied, which is a robust alternative to independent samples *t*-test (*Mair and Wilcox, 2020*). The latter test was computed by using the function 'yuen' of the WRS2 package in R. An alpha level of p<0.05 was used to test for significant main effects and interactions.

## Behavioral analyses: hand dexterity

For the Purdue Pegboard Test and the Grooved Pegboard Test, the time (in s) taken to complete each test was compared between the two age groups using two independent sample *t*-tests. For the O'Connor Finger Dexterity Test, the number of successfully filled holes (n) was compared between the two age groups using one independent sample *t*-test. A Bonferroni-corrected alpha level of p<0.016 was used to test for significant group differences.

## Behavioral analyses: gap count

For the short randomly distributed interrupts counted covertly by all participants during fMRI scanning runs 1–4 (see *Figure 1*), the accuracies were compared between the two age groups. Non-parametric Mann-Whitney U tests were performed due to non-normal distribution of the data. For both phase-encoded sessions and block-design sessions, there was no significant difference between younger and older adults on accuracies (phase-encoded: U = 115.0, p=0.15, *d* = 0.25 and block-design: U = 214.0, p=0.10, *d* = 0.28).

## Behavioral analyses: spatial acuity

Two-point discrimination thresholds were calculated per participant and run. Answers 'two pins felt' were fitted as percentages across ascending pin distances (younger: 0.7–2.8 mm, older: 0.7–6.3 mm). A binary logistic regression was used to fit the data using the glmfit function (iterative

weighted least square algorithm to receive maximum-likelihood estimators) from the Statistics Tool-box implemented in MATLAB (R2017b, The MathWorks Inc, Natick, MA, 2017). The two-point discrimination threshold was taken from the pin distance where the 50 percent niveau crossed the fitted sigmoid curve (e.g. *Kalisch et al., 2009*; *Kalisch et al., 2008*; *Kuehn et al., 2017a*; *Pleger et al., 2001*). Individual two-point discrimination thresholds were averaged across runs and compared between the two age groups (younger and older adults) using Welch's two-independent samples t-test. Additionally, correlation analyses were performed using Pearson's correlation coefficient (two-point discrimination thresholds vs pRF sizes, two-point discrimination thresholds vs Purdue Pegboard Test performance, two-point discrimination thresholds vs Grooved Pegboard Test perfromance, two-point discrimination thresholds vs O'Conner Finger Dexterity Test performance, as requested by one reviewer). The significance level was set to $p<0.05$.

## MRI analyses
### Surface reconstruction
FSL 5.0 (*Smith et al., 2004*; *Woolrich et al., 2009*) and Freesurfer's recon-all (http://surfer.nmr.mgh.harvard.edu/) were used for brain segmentation and cortical surface reconstruction using the T1-weighted 3D MPRAGE.  Note that the spatial resolution of the T1-weighted MPRAGE that was used for brain segmentation and the functional EPI sequence was identical (1 mm isotropic). Recon-all is a fully automated image processing pipeline, which, among other steps, performs intensity correction, transformation to Talairach space, normalization, skull-stripping, subcortical and white-matter segmentation, surface tessellation, surface refinement, surface inflation, sulcus-based nonlinear morphing to a cross-subject spherical coordinate system, and cortical parcellation (*Dale et al., 1999*; *Fischl et al., 1999*). Skull stripping, construction of white and pial surfaces, and segmentation were manually checked for each individual participant.

## Preprocessing
Motion artefacts and compressed distortion can be a serious problem for functional MR data, particularly those acquired at 7T where field inhomogeneity is increased. To resolve these problems, two EPIs with opposite phase-encoding (PE) polarity were acquired before the functional scan. A point spread function (PSF) mapping method was applied to perform distortion correction of both EPIs with opposite PE polarity (*In et al., 2016*). PSF mapping allows reliable distortion mapping due to its robustness to noise and field inhomogeneity (*Robson et al., 1997*). Because the amount of spatial information differs between the opposite PE datasets, a weighted combination of the two distortion-corrected images was incorporated to maximize the spatial information content of the final, corrected image. The EPI-images of the functional blocks were motion corrected to time-point = 0, and the extended PSF method was applied to the acquired and motion-corrected images to perform geometrically accurate image reconstruction. Finally, after data acquisition, slice timing correction was applied to the functional data to correct for differences in image acquisition time between slices using SPM8 (Statistical Parametric Mapping, Wellcome Department of Imaging Neuroscience, University College London, London, UK).

Functional time series were then registered to the T1-weighted 3D MPRAGE used for recon-all using csurf tkregister (12 degrees of freedom, non-rigid registration). The resulting registration matrix was used to map the x,y,z location of each surface vertex into functional voxel coordinates. The floating point coordinates of points at varying distances along the surface normal to a vertex were used to perform nearest neighbor sampling of the functional volume voxels (i.e. the 3D functional data were associated with each vertex on the surface by finding which voxel that point lay within). Because time series of the different cycle directions (D1- > D5 and D5- > D1) were mirror-symmetric to each other, they were averaged time point by time point by reversing the direction of time on a scan-by-scan basis. The time-reversed cycle direction (D5- > D1 cycle) was time-shifted before averaging by 4 s (=2 TRs) to compensate for hemodynamic delay. Averaging was done in 3D without any additional registration. Note that data were neither normalized nor smoothed (beyond interpolation during registration) during this procedure. See *Figure 1—figure supplement 1* for an overview of the representational spaces in which data of the different analyses pipelines were analyzed.

Moreover, physiological fluctuations originating from cardiac pulsation and respiration are considered a primary source of noise in functional MR data sets, particularly for resting state data acquired at high-field strengths (*Krüger and Glover, 2001*). The resting-state functional data were therefore corrected for pulse- and respiration-induced noise. To prepare the physiological data for noise correction and to remove acquisition artifacts, we used the open-source Python-based software 'PhysioNoise' (*Kelley et al., 2008*). Resulting respiratory and cardiac phase data were then used to correct the resting-state time series for pulse- and respiration-induced noise by performing RETROspective Image CORrection (RETROICOR) (*Glover et al., 2000*) on a slice-by-slice basis (*Birn et al., 2006*). Residuals were taken as cleaned data to regress out motion-related noise parameters (extracted from the raw data) using the program vresiduals implemented in LIPSIA (freely available for download at: github.com/lipsia-fmri/lipsia; *Lohmann et al., 2001*). Finally, the data were high-pass filtered at 0.01 Hz (allowing frequencies faster than 0.01 Hz to pass) using the program vpreprocess implemented in LIPSIA. For n = 9 participants, physiological data could not successfully be acquired due to a loss of the pulse oximeter and/or loosening of the breathing belt during scanning, which interrupted successful physiological data sampling. For n = 4 participants, we observed severe motion artifacts for the resting state data. Therefore, resting state analyses are presented for a subset of participants only (n = 12 younger and n = 12 older adults).

## Phase-encoded analyses

The program Fourier implemented in csurf (http://www.cogsci.ucsd.edu/~sereno/.tmp/dist/csurf) was used to conduct statistical analyses on the averaged individual time series of the averaged forward- and reversed-order runs (*Kuehn et al., 2018*). Csurf was used to run discrete Fourier transformations on the time course at each 3D voxel, and then calculates phase and significance of the periodic activation. There were 20 stimulation cycles, which were used as input frequencies. No spatial smoothing was applied to the data before statistical analyses. Frequencies below 0.005 Hz were ignored for calculating signal-to-noise, which is the ratio of amplitude at the stimulus frequency to the amplitudes of other (noise) frequencies. Very low frequencies are dominated by movement artifacts, and this procedure is identical to linearly regressing out signals correlated with low frequency movements. High frequencies up to the Nyquist limit (1/2 the sampling rate) were allowed. This corresponds to no use of a low-pass filter. For display, a vector was generated whose amplitude is the square root of the F-ratio calculated by comparing the signal amplitude at the stimulus frequency to the signal at other noise frequencies and whose angle was the stimulus phase. The data were then sampled onto the individual freesurfer surface. To minimize the effect of superficial veins on BOLD signal change, superficial points along the surface normal to each vertex (upper 20% of the cortical thickness) were disregarded. The mean value of the other layers (20–100% cortical depth) were used to calculate individual maps. On the individual subject level, clusters that survived a surface-based correction for multiple comparisons of p<0.01 (correction was based on the cluster size exclusion method as implemented by surfclust and randsurfclust within the csurf FreeSurfer framework, *Hagler et al., 2006*), and a cluster-level correction of p<0.001, were defined as significant. On the group level, clusters that survived a cluster-filtered correction of the F-values were considered significant (pre-cluster statistical threshold of p<0.01, and minimum surface area of 14 mm$^2$, according to *Hagler et al., 2007*).

Complex-valued data from each individual subject's morphed sphere were also sampled to the canonical icosahedral sphere (7th icosahedral sub-tessellation) surface. For each target vertex, the closest vertex in the source surface was found. The components in this coordinate system were then averaged (separately for younger and older adults' brains), and the (scalar) cross-subject F-ratio was calculated. The cross-subject F-ratio is calculated based on the complex coefficients at the stimulus frequency from each subject. The F-ratio is described as follows:

$$\frac{[xav^2 + yav^2]}{[\text{Sum}(\_x - xav^{2/n}) + \text{Sum}(\_y - yav)^{2/n}]/[2*n-2]}$$

where _x and _y are the raw frequency, xav and yav are the average of them, and n is the number of subjects. This is described and demonstrated in *Hagler et al., 2007*. A cluster threshold of p<0.01 was defined as significant.

Surface-based masks of area 3b of each individual brain were used to define area 3b. Surface area measurements were extracted by calculating the sum of ⅓ of the area of each of the triangles surrounding that vertex. Surface area was defined as the summed vertex-wise area of the significant area 3b map. We report two sums: the surface area as of the individual surface (the. area file), and the surface area of the original surface (the lh.area file). F-values of the Fourier model from the significant tactile map area in area 3b were extracted subject-by-subject, and were then averaged. To estimate mean response amplitudes of the tactile maps (in %), we estimated the discrete Fourier transform response amplitude (hypotenuse given real and imaginary values) for each vertex within the significant map area. This value was multiplied by two to account for positive and negative frequencies, again multiplied by two to estimate peak-to-peak values, divided by the number of time points over which averaging was performed (to normalize the discrete Fourier transform amplitude), and divided by the average brightness of the functional data set (excluding air). Finally, the value was multiplied by 100 to estimate percentage response amplitude (*Kuehn et al., 2018*). Independent-sample *t*-tests with a Bonferroni-corrected alpha level of p<0.0125 were used to compare mean F-values, mean response amplitude, and mean surface area (current and original surface) between younger and older adults. The chi-square goodness-of-fit test was used to test for normality of the data.

In addition, a 'winner-takes-it-all' approach was employed to investigate finger-specific differences in mean F-values, mean response amplitude, and mean surface area between age groups. Within the significant topographic map area as specified above, for each vertex, it was estimated on the surface whether it could be assigned to D1, D2, D3, D4, or D5 using finger-specific *t*-values. F-values and response amplitudes could then be extracted and averaged in a finger-specific way. We computed an ANOVA with the factors age and digit on these three measures to investigate main effects and interactions. Main effects and interactions with a p-value below 0.05 were specified as significant.

For non-significant results that were interpreted as evidence for the absence of an effect, we conducted statistical tests of equivalence using the two one-sided t-test (TOST). TOST is a frequentist alternative for testing for the equivalence by defining a band around 0 that constitutes a minimally-relevant effect size ($\Delta_L$ and $\Delta_U$). TOST was employed (1) by running two t-tests, testing the null hypothesis that the effect is smaller than the maximum of the indifference area and larger than its minimum, and (2) by choosing the smaller of the two *t*-values. A significant result would reject this null hypothesis, indicating that the true value lies in the indifference area (*Lakens, 2017*). To determine ΔL and ΔU, we used an approach suggested by *Simonsohn, 2015* who argue that in order to determine boundaries for equivalence tests, the boundaries should be set to the effect size of the distribution where a difference could be detected with 33% power. In the TOST procedure, the null hypothesis is the presence of a true effect of ΔL or ΔU, and the alternative hypothesis is an effect that falls within the equivalence bounds or the absence of an effect that is worthwhile to examine. If a p-value is below 0.05, we assumed the absence of an effect that is worthwhile to examine.

## Consistency of map alignment

We estimated the vertex-wise consistency of the map gradient within each age group using the dispersion index *d*, which is described as follows:

$$d = \frac{amplitude\ of\ vector\ average\ (group\ level)}{average\ amplitude\ of\ individual\ vectors}$$

The individual vectors that are referred to here are the vectors generated during the first-level Fourier analyses as outlined above ('Phase-encoded analyses'). This index is 1.0 in the case of perfectly aligned vectors across the entire group (within younger and within older participants), independent of vector amplitude. *d* therefore distinguishes a vector average that was generated by a set of large but inconsistent signals (lower *d*) from a same-sized vector average that was generated by a set of smaller but more consistent signals (higher *d*). As such, *d* provides an indication about the consistency of the map alignment within each age group.

## GLM analyses

Fixed-effects models on the 1st level were calculated for each subject separately using the general linear model (GLM) as implemented in SPM8. The analyses were performed on the two blocked-design runs (run 3 and run 4 of the experiment, see *Figure 1*). Because we treated each finger individually and independently, BOLD activation elicited by each finger's tactile stimulation was treated as an independent measure in the quantification (*Kuehn et al., 2018*; *Ann Stringer et al., 2014*). We modeled one session with five regressors of interest each (stimulation to D1, D2, D3, D4, D5). We computed five linear contrast estimates: Touch to D1, D2, D3, D4, and D5 (e.g. the contrast [4 - 1 -1 -1 -1] for touch to D1). Frame-to-frame displacement of realignment parameters (first two shift regressors and rotation regressors) did not differ significantly between age groups (all p>0.5). Given that functional and anatomical data were not normalized, no group statistics were performed with SPM. Instead, on the individual subject level, voxels that survived a significance threshold of p<0.05 (uncorrected) and $k > 3$ were mapped onto the cortical surfaces using the procedure as described above. These thresholded contrast images were used for finger mapping analyses on the individual subject level within the FSL-framework, that is, for calculating the overlap between neighboring finger representations, and for defining regions-of-interest for resting state data extraction (see below).

We also used fixed-effects models on the 1st level using the GLM as implemented in SPM8 to compute response amplitudes in response to the stimulation of all fingers together at the same (maximal) stimulation amplitude. The analyses were performed on the two stimulation runs (runs 5 and run 6 of the experiment, see *Figure 1*). We modeled both sessions with two regressors of interest, 'on' and 'off' stimulation periods. The 'on' and 'off' stimulation maps were mapped onto the cortical surfaces using the procedure as described above. Signal amplitude was calculated vertex-wise within the region-of-interest as defined above (see 'Phase-encoding analyses') by calculating the difference value between 'on' and 'off' stimulation phases, divided by the baseline value. These data were averaged and compared between groups using a two-tailed independent sample t-test with a significance level of p<0.05.

Finally, we also used GLM to reduce the number of voxel time courses used for pRF analysis (see section below 'Bayesian pRF modeling' for details).

## Representational similarity analysis

For all participants, beta values (i.e. finger versus baseline) were extracted from each single finger receptive area within area 3b. Anterior-posterior boundaries were taken from Freesurfer surface labels. Significant tactile maps in each participant (see 'Phase-encoded analyses') were used to define the response regions. We characterized topographic similarity of finger representations for each digit pair by computing the similarity between all digit combinations between the first and second run within this area (*Kuehn et al., 2018*). Correlation coefficients of the vectors were calculated using Pearson correlations. The correlation coefficients were first computed at the individual subject level, Fisher z-transformed, and then averaged across subjects to calculate digit-specific group averaged correlation matrices. We computed an ANOVA with the factors neighbor (N0-N4) and age (young, old) to test for age-related differences in distant-dependent similarities between finger representations between runs. We used an alpha level of p<0.05 to test for significant main effects and interactions.

Preprocessed resting state data that were corrected for physiological noise (see above) were used to compute similarities between time series of finger representations in area 3b. The definition of finger representations was computed based on the GLM analyses (see above), but overlapping voxels were excluded from the analyses. The matlab-function 'xcorr' was used to compute cross-correlations between time series of all finger pairs. Cross-correlations measure the similarity between two vectors and their shifted (lagged) copies as a function of the lag, and take into account temporal shifts as a function of TR. For any given lag, cross-correlations estimate the correlation between two random sequences and estimate functional connectivity (*Hyde and Jesmanowicz, 2012*).

## Cortical distance and cortical overlap

Geodesic distances between finger representations (in mm) were computed using the Dijkstra algorithm. The path follows the edges of triangular faces on the surface between each peak vertex of

each individual receptive area. Distances between neighboring digit representations were calculated by extracting paths between peaks of neighboring digit representations. We also calculated Euclidean distances between each neighboring digit pair. As a further control analysis, we also computed the Euclidean distance using center estimates rather than peak vertices (*Vidyasagar and Parkes, 2011*). With A(x1,x2,z3) and B(x2,y2,z3) the Euclidean distance were computed as:

$$d(A,B) = \sqrt{((x2 - x1)2 + (y2 - x1)2 = (z2 - z2)2)}$$

Geodesic and Euclidean distances between each neighboring digit pair were compared between groups using a 4 × 2 ANOVA with the factors digit-pair (D1-D2, D2-D3, D3-D4, D4-D5) and age (younger, older). The chi-square goodness-of-fit test was used to test for normality of the data.

Cortical overlap between adjacent finger representations (D1-D2, D2-D3, D3-D4 and D4-D5) was calculated using the Dice coefficient (*Dice, 1945*; *Kikkert et al., 2016*; *Kolasinski et al., 2016b*). The definition of finger representations was computed based on the GLM analyses (see above). The dice coefficient varies from a value of 0, indicating no digit overlap, to a value of 1, indicating perfect digit overlap. Where A and B are the area of the two digit representations, the Dice Coefficient is expressed as:

$$2|A \cap B|/|A| + |B|$$

Vertices of digit representations were determined as the number of significant vertices within the tactile 3b map area. The chi-square goodness-of-fit test was used to test for normality of the data. For normally distributed data, an ANOVA was calculated with the factors digit-pair (D1-D2, D2-D3, D3-D4 and D4-D5) and age (younger, older). An alpha level of $p<0.05$ was used to identify significant main effects and interactions.

Cohen's *d* and Hedges' *g* were used to estimate effect sizes (*Hedges, 1981*). Hedges' *g* is similar to Cohen's *d* but outperforms Cohen's *d* when sample sizes are low. 95% confidence intervals and Hedges' *g* were computed via bootstrapping 10,000 times. Bootstrapping is a non-parametric statistical test that can be applied both when the data are normal and non-normal distributed. Bootstrapping is particularly suitable for data with small sample sizes. Forest plots were used to visualize effect sizes (https://www.mathworks.com/matlabcentral/fileexchange/71020-forest-plot-for-visualisation-of-multiple-odds-ratios). A forest plot is a graphical display that illustrates the relative strength of interventions, such as training effects, in different conditions (*Timm and Kuehn, 2020*), and is often used in clinical trials to compare the effectiveness of treatments (e.g. *Kang et al., 2016*).

## Bayesian pRF modeling

Population receptive field (pRF) modeling was performed using the SPM-based BayespRF Toolbox (freely available for download from https://github.com/pzeidman/BayespRF (copy archived at URL to added)) which is dependent on Matlab (SPM12 and Matlab 2018b). The BayespRF toolbox provides a generic framework for mapping pRFs associated with stimulus spaces of any dimension onto the brain. It was first used for mapping two-dimensional (2D) visual pRFs in human visual cortex (*Zeidman et al., 2018*), and it was recently applied to map somatosensory pRFs in human SI (*Puckett et al., 2020*). BOLD time-series were extracted for pRF modeling by reducing the number of voxel time courses. This was achieved by performing a two-stage analysis. The first stage was the GLM analyses stage, which was accomplished by performing a general 1st-level analysis with SPM to prepare data for pRF modelling procedure. At this stage, task regressors were first defined. Similar to *Puckett et al., 2020*, five regressors were constructed, corresponding with five fingers of the right hand. After performing an F-contrast, only time-series that passed a significance threshold of $p<0.05$ (uncorrected) were used for the pRF modeling (*Puckett et al., 2020*; *Zeidman et al., 2018*). This procedure allows reducing computing time considerably (note that one pRF modeling process takes around 2 days for the reduced input data). The resulting mask was combined with the freesurfer mask of area 3b. pRF modeling was then conducted on a voxel-wise basis, where the fit between an estimated waveform and the empirically measured BOLD-response was optimized. This was achieved by modifying the size and position of the pRF model. The posterior model probability was thresholded at >0.95 (*Puckett et al., 2020*; *Zeidman et al., 2018*). We defined the somatosensory space using the same 2D matrix that was used for visual pRF mapping, but with limiting the dimensions to ±12.5 in both dimensions. pRF modeling was performed on the x-dimension, that is the

inferior-superior dimension of topographic alignment. Note that similar to *Puckett et al., 2020*, these analyses model one dimension in the two-dimensional sensory space. We allowed the minimal pRF size to be not smaller than 1/10th of the sensory space occupied by a single fingertip, and the maximum size restricted to the equivalence of all five fingers (i.e. 25 units) (*Puckett et al., 2020*). A Gaussian pRF profile was chosen as a response function for pRF analysis (code available at https://gitlab.com/pengliu1120/bayesian-prf-modelling.git). This model was characterized as a normal, excitatory distribution with pRF center location (x) and width (i.e. σ, the standard deviation of the Gaussian profile) as estimated parameters (*Puckett et al., 2020*).

After processing, output volumes were extracted from the obtained results, including distance, angle, width, beta, decay, transit and epsilon. Distance and angle are the vectors of polar coordinates depending on stimuli space definition, width is the defined pRF size parameter, ranging from 0.5 to 25. Distance values were used to define locations of activated voxels for each finger, width values were used as pRF size estimates for activated voxels. Before pRF modeling, parameters for pRF centre locations were set between −12.5 (low x-values) and +12.5 (high x-values), and after pRF estimation, values were modeled accordingly. For display only, the values are shown with a range between 0 and 25, so that the boundaries between each finger could be shown in integral numbers. As the stimulus space defined before modelling was one-dimensional, only pRF centre location and size results were further analyzed by plotting them onto the surface using Freesurfer (csurf). Visualized results were saved as labels accordingly, with the coordination, and pRF centre location or pRF size specified for each voxel survived pRF modelling. After performing Bayesian pRF modelling and extracting result labels for every subject, individual and group average pRF sizes were calculated and used for statistical analysis. A two-way ANOVA was performed with age and finger as independent variables, and average pRF size of each finger for each individual as dependent variable.

## Factor analyses and correlations

We used the function 'factoran' in MATLAB_R2014b, which fits factor analysis models using maximum likelihood estimates. We searched for two and three common factors, that is factors that might affect several of the variables in common. In that way, we obtained maximum likelihood estimates of the factor loadings and the specific variances, representing independent random variability, for each variable. For the refitted models with more than one common factor, the loadings were initially estimated without rotation, but for better interpretability, we further used the 'promax' oblique rotation. Factor rotation aimed at obtaining for each variable only a small number of large loadings affected by the factors, preferably only one. The 'promax' rotation chosen here rotated the factor axes separately, allowing them to have an oblique angle between them, and computed new loadings based on this rotated coordinate system. We computed lambda as the coefficient, or loading, of the jth factor for the ith variable. Note that for the factor analysis with three common factors, some of the specific variances were equal to the value of the 'delta' parameter (close to zero), leading to a fit known as a Heywood case. We computed psi as the specific variance of the model. Psi = 1 would indicate that there is *no* common factor component in that variable, while psi = 0 would indicate that the variable is *entirely* determined by common factors.

Factor analyses were conducted only on the data of older adults, and only with variables that showed differences between the two age groups. Reduction of the dimensionality of the data only to these variables that were different was necessary to reveal variables that showed joint variations due to unobserved latent variables and not due to intertwined features. Our goal was to reduce the dimensionality in the data down to the number of variables needed to capture its maximum variability. The number of dimensions does not necessarily correspond to the number of features, as not all variables measured were independent from one another. Our goal was to obtain a reduced set of degrees of freedom in the data, which could be used to reproduce most of its variability, and thus redundant features were of no interest. This redundancy was evident when we included more variables in the model. By adding more variables, the presence of Heywood cases for some variables kept arising (meaning that these variables were entirely determined by common factors with specific variances-psi values close to zero) and in some cases, the model could no longer be computed. A factor loading of a variable can be considered important for a factor, if it is at least |0.4| (*Maskey et al., 2018*), explaining around 16% of the variance in the variable (*Field, 2009*). We set a cut-off criterion for the accepted loadings in our model to |0.4| and followed the procedure of *Maskey et al., 2018* regarding the exclusion of variables (i.e. deleting one variable at a time and

always rerun the model, based first on the lowest factor loadings and then, if required, on the existence of cross-loadings). The first variable to be removed from the two-factor model was N3 representational similarity (loading of $-0.19$). After rerunning the model, pRF size was excluded next (loading of 0.30) and the model obtained after this second variable exclusion was the final two-factor model reported here. No further exclusions were required. The same process was followed for the three-factor model. pRF size was the first variable that had to be removed from the model (loading of 0.30). After pRF size exclusion, the obtained model was accepted with no further variable exclusions (all factor loadings >|0.4|). Correlation coefficients between variables of each factor were then calculated within the group of older adults using Pearson correlations. Because pRF estimates were excluded by the model (see above), but there was interest about a potential behavioral correlate of the altered pRF size, we also performed correlation analyses between average pRF sizes and cortical distances (Euclidean distance and geodesic distance), recorded time of the Purdue Pegboard Test, recorded time of the Grooved Pegboard Test, and two-point discrimination thresholds.

## Acknowledgements

E.K. was funded by ESIF/EFRE 2014–2020; FKZ: ZS/2016/04/78113, Vorhaben: Center for Behavioral Brain Sciences (CBBS); P.L. and J.D. were funded by the German Research Foundation (Deutsche Forschungsgemeinschaft, DFG) (KU 3711/2–1, project number: 423633679). A.C. was funded by the Else Kröner Fresenius Stiftung (EKFS) (2019-A03). We thank Peter Zeidman for methodological support.

## Additional information

### Funding

| Funder | Grant reference number | Author |
|---|---|---|
| Center for Behavioral Brain Sciences | ESIF/EFRE 2014-2020 FKZ: ZS/2016/04/78113 | Esther Kuehn |
| German Research Foundation | KU 3711/2-1 | Peng Liu Juliane Doehler |
| German Research Foundation | project number: 423633679 | Peng Liu Juliane Doehler |
| Else Kröner-Fresenius-Stiftung | 2019-A03 | Anastasia Chrysidou |

The funders had no role in study design, data collection and interpretation, or the decision to submit the work for publication.

### Author contributions

Peng Liu, Formal analysis, Methodology, Writing - original draft; Anastasia Chrysidou, Formal analysis, Writing - original draft; Juliane Doehler, Formal analysis, Methodology, Writing - review and editing; Martin N Hebart, Formal analysis, Methodology; Thomas Wolbers, Conceptualization, Funding acquisition, Writing - review and editing; Esther Kuehn, Conceptualization, Formal analysis, Supervision, Funding acquisition, Investigation, Methodology, Writing - original draft

### Author ORCIDs

Martin N Hebart (iD) https://orcid.org/0000-0001-7257-428X
Esther Kuehn (iD) https://orcid.org/0000-0003-3169-1951

### Ethics

Human subjects: All participants were paid for their attendance and written informed consent for participation and data handling was received from all participants before starting the experiment. The study was approved by the Ethics committee of the Otto-von-Guericke University Magdeburg (68/16).

Decision letter and Author response
Decision letter https://doi.org/10.7554/eLife.60090.sa1
Author response https://doi.org/10.7554/eLife.60090.sa2

## Additional files

### Supplementary files

• Transparent reporting form

### Data availability

Behavioral and MRI data have been deposited in dryad (https://doi.org/10.50611/dryad.mgqnk98x8).

The following dataset was generated:

| Author(s) | Year | Dataset title | Dataset URL | Database and Identifier |
|---|---|---|---|---|
| Liu P, Chrysidou A, Doehler J, Wolbers T, Kuehn E | 2020 | The organizational principles of de-differentiated topographic maps - Data | http://dx.doi.org/10.5061/dryad.mgqnk98x8 | Dryad Digital Repository, 10.5061/dryad.mgqnk98x8 |

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
