## [Decision Letter]

**Acceptance summary:**

Liu and colleagues used high-resolution neuroimaging to study topographic maps in human somatosensory cortex in a far richer and more nuanced way than previously possible, simultaneously probing multiple features of finger representations such as size, distance and separation. Importantly, these neural map features and how they change with age are directly linked to multiple measures of somatotosensation and motor behaviour, providing insights in how these topographic maps determine behavioural performance.

**Decision letter after peer review:**

Thank you for submitting your article "The organizational principles of de-differentiated topographic maps" for consideration by *eLife*. Your article has been reviewed by 3 peer reviewers, and the evaluation has been overseen by a Reviewing Editor and Floris de Lange as the Senior Editor. The following individual involved in review of your submission has agreed to reveal their identity: Michel Akselrod (Reviewer #1).

The reviewers have discussed the reviews with one another and the Reviewing Editor has drafted this decision to help you prepare a revised submission.

We would like to draw your attention to changes in our revision policy that we have made in response to COVID-19 (https://elifesciences.org/articles/57162). First, because many researchers have temporarily lost access to the labs, we will give authors as much time as they need to submit revised manuscripts. We are also offering, if you choose, to post the manuscript to bioRxiv (if it is not already there) along with this decision letter and a formal designation that the manuscript is "in revision at *eLife*". Please let us know if you would like to pursue this option. (If your work is more suitable for medRxiv, you will need to post the preprint yourself, as the mechanisms for us to do so are still in development.)

The reviewers agreed that the question addressed is of great interest, and that the authors present an impressive and very valuable dataset and apply highly advanced fMRI methods. The refutation of a simple de-differentiation model is an interesting and novel contribution – though it should be said that the reviewers did not find that a clear alternative was demonstrated in the current paper.

Overall, they felt substantial revisions are required. While the fMRI analytical methods used by the authors are very advanced and appropriate, there is a general lack of precision in the description of these methods (detailed), which would prevent a thorough replication of the author's analytical pipeline. Additionally, the hypotheses require clarification and justification, the statistical approach should be improved, and interpretation of the results needs to be more stringent in places. The reviewers also noted that following these revisions, it is possible that the main results will no longer hold. The points to be addressed are detailed below.

Essential revisions:

1. The impact statement seems misleading and does not fit with the rest of the manuscript: "De-differentiated cortical maps in S1 are characterized by local shifts in topographic representations that benefit behaviour and reduced topographic similarity, which corroborates previous models on de-differentiation and cortical aging", as de-differentiation has been associated with behavioural impairment rather than behavioural benefit and should by definition be associated with increased similarity (i.e. less differentiation), this statement is very counter-intuitive at this stage of the manuscript. In addition, the presented results do not really corroborate results from existing literature or previous models of de-differentiation. It might be more helpful to emphasize that, in the current results study, cortical aging was not associated with classical hallmarks of de-differentiation.

2. The authors interpret these results as evidence that cortical 'de-differentiation' is related to higher hand performance. This is based on the negative correlation found between a specific haptic test (O'Connor) and changes in relative peak distances between D2 and D3. However, these changes in peak distances are also negatively correlated with the increased mislocalisation between these two fingers, which may not be beneficial in other contexts. While mislocalisation is likely to reflect map overlap that is likely to affect finger motor enslavement as mentioned in the discussion, precision grip and haptic tests as done here are likely to rely more on the ability to detect peg's edges and orientation with D1, D2 and D3, which points more towards the involvement of tactile spatial acuity, in a finger-specific way.

Thus, the concern is that the conclusion made here is likely to be task and feature specific, and that the opposite conclusion might be reached when considering other aspects of tactile processing involved in hand dexterity and haptic tasks that may relate to other cortical features. For instance, the authors did not test spatial acuity performances, which are known to be impaired in elderlies (e.g., Vega-Bermudez and Johnson, 2004; Stevens, 1992; Tremblay et al., 2003; Kalisch et al., 2008; Pleger et al., 2016), assumed to be dependent on RF size and the density of innervation (see Vallbo and Johansson, 1978; Peters et al., 2009) and were found to be highly related to performances in the pegboard task (see Tremblay et al., 2003; Kalisch et al., 2008; Kowalewski et al., 2012). Thus, this aspect of 'de-differentiation' (i.e., increased pRF size), could be related to impaired spatial acuity, which could together negatively impact performance in the pegboard task. This would suggest maladaptive consequences. So, the overall claim should be dampened since it depends on which parameters we look at.

3. The feature-based model of cortical reorganization (considering local vs global aspects and separation vs integration) seems like a promising approach; however, the authors do not convincingly use these concepts as a red-thread throughout the manuscript. It is briefly mentioned in the introduction, but lacking a thorough description (e.g. the graphical summary in Figure 1A is not easily understandable). In addition, it is difficult to understand which of the tested features map onto which aspects of the model (local vs global and separation vs integration). For example, it is not clear which behavioural task assessed "functional separation" and how it was tested. In addition, the local vs global aspects feel like a post-hoc classification of results to distinguish between statistical effects associated with only one specific finger/finger-pair (e.g. finger*age interaction for cortical distance) or with all fingers (e.g. main effect of age for pRF size) rather than a hypothesis driven approach. For example, a more hypothesis driven approach could distinguish between local features for single fingers (e.g. area size, pRF size, response amplitude, etc…) and global features across fingers (e.g. distance, overlap, similarity, etc…).

4. Even though mentioned in the introduction, the authors do not take into account or discuss how finger usage (and thus use-dependent plasticity) may change with age and play a role here, especially if finger-specific. For instance, the reduced similarity between distant fingers (N3, so D1-D4 and D2-D5) (if still there after cross-validation) could arise from a use-dependent 'clustering' of D5-D4 vs D3-D2-D1 that are maybe increasingly co-used with age. While such 'clustering' might not be detectable with peak distances, the reduced similarity observed may result from the combination of two small clusters of de-differentiation driven by use-dependent changes.

5. While the literature on age-related cortical changes referred to in the introduction reported volumes changes at the level of a single finger (D2), the present study reports only global metrics across all fingers' representations. Can the authors break down the results for each finger (i.e., response amplitude, f-value, surface area)? This would also help interpreting the finger-specific changes in relative peak distances.

6. Regarding cortical distances, why did the authors use the peak vertex rather than the centre of gravity (that can be weighted by betas and/or cluster size)? The peak is sensitive to artefacts or aberrant values and does not account for cluster size. Also, interpreting differences between groups by comparing relative distances computed at the individual level between pairs of adjacent fingers is difficult, since these distances are not directly comparable between participants if the data is not normalised. If the point is to look at finger-specific differences, it might be better to consider normalised distances relative to a single anchor point consistent across subjects, for instance the D5 centre of gravity, or even better, an anatomical landmark.

7. The use of cross-validated measures and especially cross-validated Mahalanobis distances provides a more reliable measure of similarity (see Walther et al., 2016 NeuroImage). In addition, by grouping the data based on neighbourhood, different amount of data is pooled for each class: 4 pairs averaged for N1 vs 1 pair for N4. This impacts variability, degrees of freedom and thus stats. The analysis should account for that (by pooling specific pairs of fingers in a balanced way and driven by clear hypotheses, or by just comparing the 10 cells across groups).

8. In the within-scanner passive task, the stimulus intensity has been adjusted according to the participants' 50% detection threshold, determined separately for each individual and finger. As shown, the 50% detection threshold is clearly higher for the older group, and the authors also found a main effect of digit and a digit x age interaction. Unmatched stimuli are a concern for the interpretability of the S1 results. For example, the authors report no differences in response amplitude, which, given more intense stimuli for the older group, suggests that the excitability of the older group's S1 is in fact lower. While the authors may wish to point at age-related degradation of tactile information in the periphery (e.g. Hanley et al., 2019), it is unlikely the age-related drop in detection is fully explained by the periphery (as opposed to S1 or higher-order decision areas, see Romo et al., 2013). Other measures analysed by the authors are likely also dependent on stimulus intensity, e.g. surface area, Dice overlap, and pRF size. A discussion of how these corrections have affected the findings seems essential.

This issue could potentially be partially addressed with the following analyses:

a) using GLM, one can include a nuisance regressor to account for different stimulation intensity between fingers. Thus, the model includes one regressor for each finger and a nuisance regressor for the overall effect of stimulation intensity. This approach can be directly applied for their block experiment.

b) For the phase encoding analysis, the effect of the nuisance regressor can be regressed out from the fMRI timecourses before applying the phase encoding analysis.

This should at least capture some of the variance associated with different stimulation intensities, but since each finger is stimulated at a single intensity, the confound finger*intensity cannot be completely resolved.

9. A very large number of statistical tests are performed, and most effects were not significant. This could suggest that either the authors successfully identified very specific correlates of aging (e.g. D2-D3 differences), or maybe that some effects were significant by chance (e.g. the authors computed 20 correlations between the different measures and age and found that 2 out of 20 are significant without applying correction for multiple comparisons). Statistical analyses should be strongly hypothesis-driven to limit the number of tests, especially when dealing with some many variables. For instance, the reason for testing cortical distances between peaks is not clear. Is it to test the interpretation of previous results arguing that the increased D2-D5 distance is evidence for 'an enlargement of topographic maps in older adults', or to see if the authors can replicate this D2-D5 result? If the former, then is this analysis necessary considering that enlargement of maps was already assessed above with the surface area measure (that can be done both globally and in a finger-specific way)? If the latter, then why are the authors testing relative distances between adjacent fingers instead of distances relative to D5? In the same line, correlations between data not showing significant main effect of age or interactions should be avoided (e.g., correlation between mean pRF sizes and dice coefficients that are not significant). Similarly, running stats on both the factors neighbour/age and digit-pair/age for RSA consists in running the test twice on the same data (circularity), just pooled differently (and in an unbalanced way when it comes to neighbourhood). The same applies to the mislocalisation analysis. If the purpose is to test global vs local effects, the authors should have independent measures to assess each. For instance, while local effects can easily be measured by maps overlaps/RSA and mislocalisations, global effects could be assessed by global changes in surface area/pRF sizes and hand dexterity? Finally, the hypothesis and justification for the resting state analysis is not clear.

10. While the correspondence between fMRI features and behaviour is quite convincing and suggests that these effects are not spurious, a more sensitive statistical approach might shed new light on the data. First, why did the authors decide to define two groups of age rather than using age as a covariate (e.g. ANCOVA or LMM)? This approach could be sensitive to both between-group differences and within-group differences, thus preventing to rely on a large number of correlations. Second, the comparison between fMRI features and behaviour mostly relies on the qualitative correspondence between significant effects (except for the correlations computed in the factor analysis). If the aim was to determine which local or global fMRI features are associated with behavioural changes and aging, it might be more optimal to directly compare fMRI features and behaviour by defining local fMRI features for individual fingers (e.g. area size, pRF size, response amplitude, etc…) and global fMRI features across fingers (e.g. distance, overlap, similarity, etc…). These fMRI features could then be used as co-variates to analyse behaviour (e.g. ANCOVA or LMM).

11. Since most of the effects tested here were non-significant, considering the Bayesian equivalents of these test (available in R or JASP) might provide useful information by assessing the evidence for the presence or absence of an effect (rather than rejecting a possible difference with frequentist approach).

12. The authors are particularly lenient with describing effects as 'trending towards significance'. P-values above 0.1 should certainly not be described this way. In some places, the (range of) p-values were not given directly in the main text, making it harder for the reader to evaluate the strength of the evidence.

13. Please provide more details regarding the Bayesian pRF modelling:

a. Voxels were preselected prior to the analysis, but only a significance threshold is given. What was the contrast used to exclude voxels?

b. Which prior(s) was/were used – if the prior was based on 'typical pRF estimates' did this bias your estimates away from finding group differences?

c. Why did the authors use Bayesian pRF modelling, rather than a frequentist approach, which would match the rest of the manuscript's statistics?

d. Please provide more information about the fit. Could the larger RF estimates in older participants be due to a poorer fit of the Gaussian?

e. Is there good correspondence between the pRF distance estimates and the topographic maps estimated using phase encoding? Please also provide a supplementary figure with all participants' pRF estimates.

f. Relatedly, P26 who appears to be an outlier with very high pRF size for D5 (in Figure 4A) has medial activity for D1 (Figure 2C). If 'digit space' is parameterised as a circle instead of a line (allowing shared D5-D1 RFs), are the group estimates affected?

g. "The main effect of finger was due to significantly larger pRF sizes of D1 compared to D2" – p. 8. Based on figure 4, I think this should be "smaller".

h. "For display, the dimensions of the pRF distance volumes were adjusted to 0/25." – p. 31. It is unclear what this means, please rephrase.

14. Representational similarity analysis (RSA):

a. The distinction between 'local' and 'global' is not very clear, particularly when applied to distant finger pairs. Are N3 correlations the within-subject average of D1-D4 and D2-D5? Do the authors propose a global mechanism that only affects these two finger pairs, but not, say, D2-D4 or D3-D5?

b. RSA values are typically plotted as dissimilarity (e.g. 1-corr.). The decision to plot the corr. directly while using the term RSA may cause some confusion. This is at the authors' discretion.

15. Dexterity and factor analysis:

a. The distinction between measures of hand dexterity that are 'global' and 'local' seems to be fully driven by the results from the factor analysis. As it stands, it is unclear why the O'Connor test is fundamentally different from the other two. The authors should attempt to explain why the O'Connor test was associated with a positive loading on local effects, while the Purdue and Grooved tests were associated with a negative loading on the global effects. These three tasks, while corresponding to different levels of difficulty, were described as assessing similar aspects of hand dexterity. It would also be helpful if the authors can provide information about potential different strategies used for the three tasks across the different age groups (e.g. which fingers are primarily used to handle the pins, did this differ across the tasks and across age groups).

b. In the Discussion section and abstract, then, the O'Connor test is reduced to "hand dexterity", e.g. in "more local 'de-differentiation' related to better hand dexterity" – p. 18. Given the other two tests of hand dexterity showed different effects, this should be phrased more carefully.

c. Both Euclidean and Geodesic distance are included as inputs to the factor analysis and are the inputs with the strongest loadings on factor 1. Does this influence the factors?

d. Why were the various non-significant digit pairs or neighbourhood relations (for confusion, distance, or repr. correlation) not included in the factor analysis? Perhaps one way to show how meaningful the distinction between the factors is would be to show that even non-significant group differences show an appropriate difference in factor loadings.

[Editors' note: further revisions were suggested prior to acceptance, as described below.]

Thank you for resubmitting your work entitled "The organizational principles of de-differentiated topographic maps" for further consideration by *eLife*. Your revised article has been evaluated by Floris de Lange (Senior Editor) and a Reviewing Editor.

The manuscript has been much improved but there are some remaining issues that need to be addressed, as outlined below:

Essential revisions:

1. The authors state in the results that the finger maps appeared in topographic alignment in all participants (page 6 and page 11). However, this is hardly appreciable when looking at the figures (Figure 2, 4 and 4S). There is also no description about how this topographic alignment was assessed. We recommend assessing statistically the topographic alignment between digits (different from the between subject dispersion analysis) to support these claims. Topographic alignment is visually depicted in Figure 3E, but this assessment is not described in the Results section.

To be specific, on page 6, "The expected topographic alignment of the fingers (thumb [D1], index finger [D2], middle finger [D3], ring finger [D4], and small finger [D5] arranged superior -> inferior in area 3b) was visible in all individuals of both age groups (see Figure 2C)", this seems like an overstatement after visual inspection of Figure 2C. Visual inspection of individual maps (Figure 2, 4 and 4S) suggest the presence of "missing" fingers in some individuals. This might be the case if no voxel responded preferably to a finger.

For example P1, P18, P25, P28, P33, P36 seem to have at least 1 missing color in the map. Please specify how the topographic alignment was assessed and how many participants had "missing" fingers in the maps (if any), and report the minimum number of voxels associated with a given finger in the topographic maps. In case a given subjects had a "missing finger", how was missing data treated for statistics?

Further, on page 11, "pRF centre locations revealed an organized finger map with D1-D5 arranged from superior to inferior, as expected", the authors probably meant "inferior to superior. As for phase encoding, please specify how the topographic alignment was assessed and whether some participants had "missing" fingers in the maps and how this was treated for statistics. Inspection of figure 4S suggests indeed that some subject had "missing" fingers in their maps.

2. In the GLM used to define finger-specific maps, which are then used for overlap and resting-state analyses, the statistical threshold used is too liberal (p<0.05 uncorrected) which might affect the results of subsequent analyses. Page 36, line 1155: "voxels that survived a significance threshold of p<.05 (uncorrected) and k>3", this is a very liberal threshold to define finger-specific areas, which will likely detect unspecific responses and result in larger overlap between digits. This is problematic as the result "increased overlap in elderly's maps" might be attributed to difference in unspecific responses. In addition, these maps were used for resting-state data extraction, which is also problematic, because an increased overlap would predict an increased resting-state functional connectivity. It may be better to use a more conservative threshold (e.g. 0.001 uncorrected) and to exclude any overlapping areas in the resting-state analysis.

3. Considering that the authors select the variables 'that were shown differ between age groups' as inputs to the factor analysis, and that they now report a significant main effect of age on dice coefficients (and not just a trend), these should be added to the factor analysis. Related to that, please consider adding the dice result in the summary of your results at the beginning of the discussion.

4. Response to previous comment 7. The scale is saturating the colours in the RDMs so it is difficult to inspect. Could the authors report the same stats as reported in the main text (ANOVA neighbour*age and ANOVA finger pairs*age) to see if results hold?

[Editors' note: further revisions were suggested prior to acceptance, as described below.]

Thank you for resubmitting your work entitled "The organizational principles of de-differentiated topographic maps" for further consideration by *eLife*. Your revised article has been evaluated by Floris de Lange (Senior Editor) and a Reviewing Editor.

The manuscript has been substantially improved but there are a few remaining issues that need to be addressed, as outlined below. We anticipate that these remaining issues can be assessed by the editors, without sending the manuscript out to the reviewers again.

1. From the authors' response to point 1:

"With respect to the missing fingers, there were fingers 'missing' in the pRF centre location maps for 8 out of 36 subjects, as shown in Figure 4S. This is equal for the group of younger and older adults (n=4 in each group). The ANOVA with the factors age and finger calculated on pRF sizes was calculated with the missing cases excluded from the data, as described on page 11. This is justified because overall, the missing values take up 8.9% of the data where biases are expected when more than 10% of the data are missing (Dong, Y., and Peng, C. Y. J. (2013). Principled missing data methods for researchers. SpringerPlus, 2(1), 1-17, Bennett, D. A. (2001). How can I deal with missing data in my study? Aust N Z J Public Health, 25(5), 464-469.)."

However, this information does not seem to be included in the manuscript. Page 10 describes the relevant ANOVA, but there is no mention of missing fingers or values. Please make sure to include this.

2. Please include the results of the analysis with multivariate noise normalization (response to reviewers point 4) in the manuscript. This could be either in the text or as a supplementary figure.

3. *eLife*'s policy is that article titles should clearly indicate the biological system under investigation; please consider changing your title accordingly. "Topographic maps" is ambiguous; you might consider "somatotopic maps" or "topographic maps in somatosensory cortex".

---

## [Author Response]

Essential revisions:1. The impact statement seems misleading and does not fit with the rest of the manuscript: "De-differentiated cortical maps in S1 are characterized by local shifts in topographic representations that benefit behaviour and reduced topographic similarity, which corroborates previous models on de-differentiation and cortical aging", as de-differentiation has been associated with behavioural impairment rather than behavioural benefit and should by definition be associated with increased similarity (i.e. less differentiation), this statement is very counter-intuitive at this stage of the manuscript. In addition, the presented results do not really corroborate results from existing literature or previous models of de-differentiation. It might be more helpful to emphasize that, in the current results study, cortical aging was not associated with classical hallmarks of de-differentiation.

We agree with the reviewers’ objections. We now changed the impact statement to:

“SI Topographic finger maps of older adults show signs of cortical aging but do not show classical hallmarks of cortical de-differentiation.”

2. The authors interpret these results as evidence that cortical 'de-differentiation' is related to higher hand performance. This is based on the negative correlation found between a specific haptic test (O'Connor) and changes in relative peak distances between D2 and D3. However, these changes in peak distances are also negatively correlated with the increased mislocalisation between these two fingers, which may not be beneficial in other contexts. While mislocalisation is likely to reflect map overlap that is likely to affect finger motor enslavement as mentioned in the discussion, precision grip and haptic tests as done here are likely to rely more on the ability to detect peg's edges and orientation with D1, D2 and D3, which points more towards the involvement of tactile spatial acuity, in a finger-specific way.Thus, the concern is that the conclusion made here is likely to be task and feature specific, and that the opposite conclusion might be reached when considering other aspects of tactile processing involved in hand dexterity and haptic tasks that may relate to other cortical features. For instance, the authors did not test spatial acuity performances, which are known to be impaired in elderlies (e.g., Vega-Bermudez and Johnson, 2004; Stevens, 1992; Tremblay et al., 2003; Kalisch et al., 2008; Pleger et al., 2016), assumed to be dependent on RF size and the density of innervation (see Vallbo and Johansson, 1978; Peters et al., 2009) and were found to be highly related to performances in the pegboard task (see Tremblay et al., 2003; Kalisch et al., 2008; Kowalewski et al., 2012). Thus, this aspect of 'de-differentiation' (i.e., increased pRF size), could be related to impaired spatial acuity, which could together negatively impact performance in the pegboard task. This would suggest maladaptive consequences. So, the overall claim should be dampened since it depends on which parameters we look at.

First, we thank the reviewers for highlighting the importance of spatial acuity for characterizing behavioral and cortical aging. As part of another study where the same participants took part (both older and younger), we performed a spatial acuity task (two-point discrimination task) at D2 where the ability to discriminate two pins at the fingertip was measured. This task has also been used by some of the studies the reviewers cite above (for example Kalisch et al., 2008; Pleger et al., 2016). We agree with the reviewers that these data provide potentially important information to the present study with respect to the relationship between cortical and behavioral aging. We therefore now included these data into the manuscript (see Methods part for more information on experimental procedure) to specifically test whether

1. Older adults show impaired tactile spatial acuity at the fingertip compared to younger adults.

2. Individual tactile spatial acuity shows the expected relation to performances in the Pegboard Test as mentioned by the reviewers (see Tremblay et al., 2003; Kalisch et al., 2008; Kowalewski et al., 2012).

3. Individual pRF size correlates with individual 2PDT thresholds, which would, as the reviewers correctly noted, weaken our conclusion that increased pRF size does not have a behavioral correlate in our study.

The results of these analyses are now also reported in the manuscript including the used methodology. The results revealed:

1. As expected, we found significantly higher tactile spatial discrimination thresholds (reflecting worse performance) in older compared to younger adults (older: 3.17mm±0.85mm, younger: 1.78±0.29mm, *t*(19.52)=6.14, *p*<.00001). This replicates prior studies that evidenced worse spatial acuity at the fingertip in older compared to younger adults.

2. As expected and as also suggested by the reviewers, individual tactile spatial acuity thresholds correlated with performance in the Grooved Pegboard Test in older, but not in younger adults (n=17 older: r=0.68, p<.005, n=18 younger: r=0.01, p=0.983); the positive correlation revealed that higher spatial acuity relates to better performance in the Pegboard Test.

3. 2PDT thresholds did not correlate with individual pRF sizes neither in younger nor in older adults (n=16 younger: r=-0.07, p=0.789, n=16 older: r=0.35 p=.179).

The spatial acuity data therefore showed the expected patterns (worse spatial tactile acuity in older compared to younger adults, and correlation with Pegboard Test performance in older adults). However, there was no significant correlation between the 2PDT and pRF size neither of D2 nor of the full map area. Our previous notion that in the present study, pRF sizes are not related to a behavioral readout is therefore still valid. The potential behavioral relevance of enlarged pRF sizes still remains to be determined by future studies.

We also agree with the reviewers that in order to understand the behavioral relevance of functional map differences, the interpretation depends on the specific features that are investigated (see also Figure 1, feature-based model of cortical reorganization). It is in particular relevant, as the reviewers also point out, to distinguish between the Pegboard Test (which relies on finger acuity) and performance of the precision grip, and the O’Connor Dexterity Test, (which relies on haptic object recognition) and haptic exploration. It is also relevant to distinguish between map features that link to functional integration and those that link to functional separation, as both differently affect performance in different tasks (i.e., tactile separation versus haptic object recognition). This is now outlined in the Introduction, and also discussed in the Discussion in the following way:

“We also distinguished between topographic map features that link to functional separation, as here tested by perceptual finger individuation and by motor tasks that rely on precise spatial acuity of the fingertip (i.e., Pegboard test, Kalisch et al., 2008), and those that require functional integration, as here tested by perceptual finger confusion and a motor task that relies on haptic recognition involving multiple fingers (i.e., O’Connor Dexterity test). It is worth noting that local and global changes as well as integration and separation that are here introduced as different levels of the features “spatial extent” and “functional readout”, respectively (see Figure 1) are in fact interlinked and may share common variance. For example, less finger individuation in one task may relate to more finger integration in another task, and both may influence motor behavior. However, their distinct investigation allows a precise understanding of how functional map features link to behavioral phenotypes (i.e., feature-based model of cortical reorganization, FMC, see Figure 1).” (page 3)

“Perhaps the most surprising finding of our study, however, is the reduced local cortical distance between the index and middle finger representations in older adults with preserved map size, where, previously, greater distances between fingers and larger map sizes were reported (Kalisch et al., 2009). Whereas the reduced cortical distance between neighbouring fingers is in principle in line with a de-differentiation model of topographic map architecture, its relation to better rather than worse hand dexterity is certainly not. In this respect, the distinction between functional readouts that capture integration versus separation seems relevant (see Figure 1). Specifically, we found that reduced cortical distances between D2 and D3 representations related to worse tactile discrimination between these two fingers (i.e., worse separation in a tactile task) but to better motor control in a task that required haptic object exploration and coordinated finger movements (i.e., improved integration in motor task). These findings highlight the importance of assessing different aspects of signal integration and signal separation to investigate the impact of functional map changes on everyday life. Because this relationship was only seen for the index and middle finger and not for the other fingers, also the factor of spatial extent (i.e., local versus global, see Figure 1) is relevant for comprehensive data analysis.” (page 24/25)

3. The feature-based model of cortical reorganization (considering local vs global aspects and separation vs integration) seems like a promising approach; however, the authors do not convincingly use these concepts as a red-thread throughout the manuscript. It is briefly mentioned in the introduction, but lacking a thorough description (e.g. the graphical summary in Figure 1A is not easily understandable). In addition, it is difficult to understand which of the tested features map onto which aspects of the model (local vs global and separation vs integration). For example, it is not clear which behavioural task assessed "functional separation" and how it was tested. In addition, the local vs global aspects feel like a post-hoc classification of results to distinguish between statistical effects associated with only one specific finger/finger-pair (e.g. finger*age interaction for cortical distance) or with all fingers (e.g. main effect of age for pRF size) rather than a hypothesis driven approach. For example, a more hypothesis driven approach could distinguish between local features for single fingers (e.g. area size, pRF size, response amplitude, etc…) and global features across fingers (e.g. distance, overlap, similarity, etc…).

We agree with the reviewers that our model is a potentially promising approach to investigate the relationship between topographic map features and everyday behavior. We apologize if this approach and the associated experimental design was not clearly explained in the previous version of the manuscript. We now added changes to the manuscript (Introduction, Results, Discussion) to better explain our experimental approach and the underlying theoretical concept. We also modified Figure 1 and added a definition of the terms “global”, “local”, “separation”, and “integration” into the figure. In order to clarify which of the tested features map onto which aspect of the model, we delineated which of the tested features were classified as local (finger specific measures of map amplitude, size, model fit, cortical distance, representational similarity, etc.) and which as global (features that characterize the full map area). We explain this when results on these variables are reported and interpreted. We also explain which of the tested features can be used to investigate separation, and which can be used to investigate integration. In fact, functional separation was tested by perceptual finger individuation and by motor tasks that rely on precise spatial acuity of the fingertip (i.e., Pegboard test, Kalisch et al., 2008), whereas functional integration was tested by perceptual finger confusion and a motor task that relies on haptic recognition involving multiple fingers (i.e., O’Connor Dexterity test, see also response to Essential Revision 2). Therefore, as the reviewers suggested, different experimental variables link to the different levels and factors of the model, which is now clarified in different sections of the manuscript.

4. Even though mentioned in the introduction, the authors do not take into account or discuss how finger usage (and thus use-dependent plasticity) may change with age and play a role here, especially if finger-specific. For instance, the reduced similarity between distant fingers (N3, so D1-D4 and D2-D5) (if still there after cross-validation) could arise from a use-dependent 'clustering' of D5-D4 vs D3-D2-D1 that are maybe increasingly co-used with age. While such 'clustering' might not be detectable with peak distances, the reduced similarity observed may result from the combination of two small clusters of de-differentiation driven by use-dependent changes.

We agree that a different (and longer) history of finger usage is an important component of aging that should be taken into account when interpreting our data. When considering the suggestion that there may be a clustering of D4 and D5 and/or of D1, D2, and D3 that may explain our representational similarity results (reduced N3 similarity in older adults), the dice coefficient results, the resting state results, and the cortical distance results do not support such a clustering. The representational similarity results show no interaction between digit-pair and age (statistics now reported in the manuscript: F(3,102)=1.20, *p*=.31) that would be expected if we assumed a finger-specific clustering. Rather, the data show an interaction between neighbour and age, which indicates a similarity effect that is independent of the specific finger. The dice coefficient results do not show an interaction between finger-pair and age (F(3)=0.1 *p*=0.963), which again does not support the clustering hypothesis. Finally, the resting state results also do not show an interaction between finger-pair and age (F(4)=0.73, *p*=.57).

Our data indicate a clustering of D2 and D3 in older adults, which seems to be beneficial for motor control. Following the reviewers’ argument, one could now assume that the more frequent co-use of D2 and D3 in older adults (perhaps due to its positive relation to manual dexterity) causes D2 and D3 to shift towards each other, and causes D1 and D4 as well as D2 and D5 to be less similar to each other. To test this assumption, we computed a correlation between individual D2-D3 distances and N3 representational similarity in older adults. These two correlations (one with Euclidean distance and one with geodesic distance) were not significant, and had low correlation coefficients (Euclidean distance-N3: *r*=-.084, *p*=.75, geodesic distance-N3: *r*=-.056, *p*=.83, see Author response image 1).

**Author response image 1. sa2fig1:** 

In addition, we investigated another hypothesis of how the increased N3 similarity in older adults could be explained. Even though individual age and individual hand usage are related (where we do not have data on individual hand usage patterns over the participants’ lifespan), we would expect shared variance between N3 representational similarity and individual age if both measures are related to one another. For this purpose, we computed a correlation between N3 representational similarity and individual age in older adults, which was significant (*p*=.047, *r*=.49). The correlations between cortical distance and age, however, were not significant (*p*=.721, r=0.094 and *p*=.483, r=.138). This indicates that individual age is one factor that relates to individual levels of N3 representational similarity but not to individual cortical distances. Even though speculative, one may interpret these results by assuming that the reduced cortical distances are use-dependent whereas the representational similarity changes are age-dependent. These analyses are now included in Figure 5—figure supplement 1.The aspect of age-related versus use-dependent changes is now discussed in the Discussion in the following way:

“Another interesting aspect is that the reduced cortical distances between index and middle finger representations in older adults may be explained by use-dependent plasticity (Makin et al., 2013a). During everyday hand movements, the index and the middle finger correlate less with each other than other neighbouring fingers (Belić and Faisal, 2015), and the index finger is the most independent of the four fingers (Ingram et al., 2008). During tactile learning, however, plasticity transfers more from the middle finger to the index finger than from the middle finger to the ring finger (Dempsey-Jones et al., 2016), which indicate their interaction. The observed local map changes may therefore be induced by the correlated input of the middle and ring finger or ring finger and small finger (Kolasinski et al., 2016b), or by age-related changes in the local myeloarchitecture that link to functional map topography (Carey et al., 2018; Kuehn et al., 2017a). Finally, because the reduced cortical distances between D2 and D3 seems to have a beneficial relationship to motor control, also the increased coupling of D2 and D3 during haptic exploration may explain the observed effect.” (page 25)

“Also the observed age-related differences in representational similarity between distant finger representations can be discussed in the light of use-dependent versus age-dependent plasticity. One way to explain less representational similarity between distant finger representations in older adults is to assume a greater clustering of D1, D2 and D3 and/or of D4 and D5 with increased hand use. This could explain why the similarity between the representations of D1 and D4 and between D2 and D5 is lower in older adults. However, we do not see an interaction between finger and age in representational similarity, resting state correlations, or dice coefficients, which would be expected if a finger-specific clustering of D1 and D2 or D4 and D5 existed. Rather, we observe a specific shift of D2 and D3 towards each other, an effect that positively related to hand dexterity. However, this shift does not correlate with the reduced representational similarity between distant fingers within the group of older adults. A direct relationship between these two measures can therefore not be established based on our data. On the other hand, we observed a significant correlation between N3 representational similarity and individual age in older adults, which hints towards a potential critical role of the factor age for the development of reduced similarity between distant finger pairs, and could be established via U-fiber maturation, as discussed above. Nevertheless, because we do not have information about hand use patterns of our participants, the effect of hand use on our data cannot be specifically investigated here.” (page 25/26)

5. While the literature on age-related cortical changes referred to in the introduction reported volumes changes at the level of a single finger (D2), the present study reports only global metrics across all fingers' representations. Can the authors break down the results for each finger (i.e., response amplitude, f-value, surface area)? This would also help interpreting the finger-specific changes in relative peak distances.

We thank the reviewers for this point. We computed finger-specific f-values, finger-specific % response amplitudes, and finger-specific surface areas using a “winner-takes-it-all” approach to extract these variables in a finger-specific way. We computed an ANOVA with the factor age and digit for these three metrics. Our results revealed no main effect of age (as expected because there was also no main effect of age for the global metrics), but also no interaction between age and digit (% response amplitude: main effect of age: F(34)=1.46, *p*=.23, interaction age and digit: F(136)=1.20, *p*=.31; f-value: main effect of age: F(34)=0.30, *p*=.59, interaction age and digit: F(136)=1.17, *p*=.33, surface area: main effect of age: F(34)=0.02, *p*=.88, interaction age and digit: F(136)=1.64, *p*=.17). These results are now also reported in the main text and in the novel figure panels of Figure 2 H-J.

6. Regarding cortical distances, why did the authors use the peak vertex rather than the centre of gravity (that can be weighted by betas and/or cluster size)? The peak is sensitive to artefacts or aberrant values and does not account for cluster size. Also, interpreting differences between groups by comparing relative distances computed at the individual level between pairs of adjacent fingers is difficult, since these distances are not directly comparable between participants if the data is not normalised. If the point is to look at finger-specific differences, it might be better to consider normalised distances relative to a single anchor point consistent across subjects, for instance the D5 centre of gravity, or even better, an anatomical landmark.

Thank you for these important comments. Functional values were first projected onto the cortical surface before peak vertices were extracted and cortical distances were estimated. Data were extracted between 100-20% cortical depth, which reduced the contribution of superficial veins that are a major contributor for aberrant values and image artifacts particularly at 7T fMRI. Data points between 100-20% were then averaged (rather than taking the maximum value), which also reduced the influence of outliers on the final estimates. Finally, we applied a surface-based smoothing kernel of 1mm to the cortical surface to account for small variations in surface sampling or small artifacts in the reconstructed surface. This procedure reduced the number of aberrant values and artifacts significantly and provides us with reliable topographic mapping data (see Figure 2 and Figure 4—figure supplement 1).

We agree with the reviewers, however, that different methods can be employed to compute the localization of fMRI activations, the most prominent being peak voxel, center coordinate, and center of gravity (COG). Vidyasagar and Parkes, (2011, JMRI) used the primary somatosensory finger map area to investigate which of these three measures is the most reliable to define finger locations and to compare finger distances between different runs. Similar to our design, the authors applied vibrotactile stimulation either to D2 or to D4, and performed reproducibility analyses to estimate which measure (under which statistical threshold) was most reliable to estimate finger localization and finger distance. The authors used an EPI voxel size of 2 mm isotropic and applied prospective motion correction to ensure that differences between sessions would not be due to head motion. The authors concluded: “This study shows a high level of reproducibility for fMRI localization in the somatosensory system.”, which applied to all three measures. This study therefore provides empirical evidence that the estimation of finger distances based on peak voxel estimates that we use is a suitable and reliable method to investigate cortical distances in S1.

Vidyasagar and Parkes, (2011, JMRI) also found that even though all three analyses showed high reliability (as explained above), center coordinates at an activation size of 200 voxels showed the highest reliability of the three introduced methods. Therefore, to investigate whether the observed age-related differences in D2-D3 cortical distances can be replicated with this method, we computed center coordinates for each finger using a threshold that allowed a mean activation size of 200 voxels, as employed in Vidyasagar and Parkes, (2011, JMRI). The results of this analysis repeated the patterns of our previous results: there was a trend towards reduced distances in older compared to younger adults only for the D2-D3 distance, but not for all other distances (D2-D3 young: 5.34 mm±0.83 mm, old: 2.62 mm±0.67 mm, *p*=.099, D1-D2 young: 3.38 mm±0.69 mm, old: 3.36 mm±0.70 mm, *p*=.98; D3-D4 young: 3.09 mm±0.41 mm, old: 3.59 mm±0.45 mm, *p*=.41; D4-D5 young: 9.09 mm±0.64 mm, old: 10.36 mm±0.92 mm, *p*=.25). Power analyses revealed that a significant effect for the D2-D3 distance would be obtained for center co-ordinates with slightly higher sample sizes (n=20 for each group to reach *p*<0.05 and power of 80% according to Dhand, N. K., and Khatkar, M. S. (2014), whereas we performed the analyses on n=19 younger and n=17 older adults). This additional analysis is now reported in Figure 3—figure supplement 1.

It is also worth noting that the pattern of results (reduced distance between D2 and D3) does not only reflect the pattern of our behavioral data that were assessed at a separate day (more tactile confusion between D2 and D3), but that the pattern also reflects previously reported behavioral data (increased motor enslaving specifically of the middle finger during index finger flexion, and lower range of independent movement specifically of the index finger in older adults, Van Beek et al., 2019).

Different anatomical brain changes are associated with cortical aging, such as cortical thinning, cortical flattening, and decreased cortical volume. Given these changes, defining an exactly corresponding anatomical location in younger and older adults’ brains at the single subject basis is challenging. In addition, using probabilistic mapping, a recent study has shown that there is high individual variability in finger locations in SI that does not allow to define a distinct anatomical location for each individual digit (O’Neill et al., 2020 Neuroimage). Therefore, functional rather than anatomical localizers are the preferred and more precise option to localize individual fingers in SI, and to compare finger maps between older and younger adults.

It is worth noting, however, that we expect the functional changes we report here to have an anatomical correlate. Identifying these correlates, however, is beyond the scope of the present article. We are currently preparing a manuscript on structural cortical differences of the same individuals, where we focus on layer-specific differences in cortical myelin and cortical iron. First results can be inspected already (https://www.researchgate.net/publication/328134104_On_the_Influence_of_Age_on_Intracortical_Myelin_in_Human_Primary_Somatosensory_Cortex_and_its_Relation_to_Tactile_Behavior#fullTextFileContent) and two poster presentations are planned for the OHBM 2021 where Juliane Döhler and Alicia Northall will present first results on age-related differences in the cortical meso-scale architecture. These results indicate that age-related changes in superficial myelin content may relate to the topographic map changes, but the analyses are not yet finalized as also differences in iron content may be influential. However, as stated above, including these results into the present manuscript is beyond the scope of the article.

7. The use of cross-validated measures and especially cross-validated Mahalanobis distances provides a more reliable measure of similarity (see Walther et al., 2016 NeuroImage). In addition, by grouping the data based on neighbourhood, different amount of data is pooled for each class: 4 pairs averaged for N1 vs 1 pair for N4. This impacts variability, degrees of freedom and thus stats. The analysis should account for that (by pooling specific pairs of fingers in a balanced way and driven by clear hypotheses, or by just comparing the 10 cells across groups).

We thank the reviewer for their suggestion. Indeed, Walther et al., (2016, Neuroimage) reported that the use of multivariate noise normalization can contribute to strongly improved pattern reliability, whereas the exact distance metric used after noise normalization (Euclidean or correlation) had a smaller effect, and where cross-validation was useful specifically for getting unbiased estimates of mean pattern similarity. However, the improvements through using multivariate noise normalization have not been unequivocal. Charest, Kriegeskorte and Kay, (2018, Neuroimage, Figure 4a) and Ritchie and Op de Beeck, (2020 Journal of Vision abstract) showed reduced reliability through multivariate noise normalization. In response to the reviewer's request, we carried out multivariate noise normalization, using the residuals of the general linear model for a better estimate of the noise covariance matrix, which we regularized using the same criteria as in Walther et al., (2016). The change in pattern reliability is shown in Author response image 2 (before / after), indeed demonstrating that noise normalization harmed estimated effects in this case. Since correlation coefficients are based on independent data (separate runs), the estimated mean correlation coefficients should not be positively biased. Since we carried out classical random-effects analyses, the different number of comparisons at the subject level should only minimally affect the estimated statistics at the group level.

8. In the within-scanner passive task, the stimulus intensity has been adjusted according to the participants' 50% detection threshold, determined separately for each individual and finger. As shown, the 50% detection threshold is clearly higher for the older group, and the authors also found a main effect of digit and a digit x age interaction. Unmatched stimuli are a concern for the interpretability of the S1 results. For example, the authors report no differences in response amplitude, which, given more intense stimuli for the older group, suggests that the excitability of the older group's S1 is in fact lower. While the authors may wish to point at age-related degradation of tactile information in the periphery (e.g. Hanley et al., 2019), it is unlikely the age-related drop in detection is fully explained by the periphery (as opposed to S1 or higher-order decision areas, see Romo et al., 2013). Other measures analysed by the authors are likely also dependent on stimulus intensity, e.g. surface area, Dice overlap, and pRF size. A discussion of how these corrections have affected the findings seems essential.This issue could potentially be partially addressed with the following analyses:a) using GLM, one can include a nuisance regressor to account for different stimulation intensity between fingers. Thus, the model includes one regressor for each finger and a nuisance regressor for the overall effect of stimulation intensity. This approach can be directly applied for their block experiment.b) For the phase encoding analysis, the effect of the nuisance regressor can be regressed out from the fMRI timecourses before applying the phase encoding analysis.This should at least capture some of the variance associated with different stimulation intensities, but since each finger is stimulated at a single intensity, the confound finger*intensity cannot be completely resolved.

We controlled tactile stimulation intensity for each finger and each individual to investigate age-related differences in the topographic architecture of the SI maps that are independent of age-related differences in peripheral nerve integrity and skin thickness. For this reason, we assessed tactile detection thresholds for each finger and each individual prior to fMRI scanning to adjust stimulation intensity accordingly. This procedure is a standard procedure used in studies that investigate age-related changes in SI (see for example Pleger et al., 2016 Sci Rep where stimulation intensity was adjusted to the 2.5-fold detection threshold). In addition, by using a stimulation method where only two out of eight (random) pins were risen at any one time point (and changed every 62.3 ms), we also minimized the effect of age-related differences in neuronal adaptation on our results. Controlling for these aspects is, in our view, a strength rather than a weakness of our experimental design. Interactions between peripheral factors, neuronal adaptation, lateral inhibition and cortical excitability can usually hardly be disentangled using fMRI (see e.g., Kuehn et al., 2014 Brain Struct Funct for a discussion on this point). On the other hand, controlling these variables allows us to more precisely understand observed (or missing) age-related differences.

We agree, however, that age-related differences in cortical response amplitudes (where both peripheral and central aspects are taken into account) may provide complementary information. We therefore performed novel analyses to investigate this aspect. We used the two short runs that were scanned at the end of the MR session, in which all participants were briefly stimulated at all of their fingers at once using the same (maximal) stimulation amplitude (see Materials and methods, fMRI Task). We computed response amplitudes for the same region-of-interest as used before (see Figure 2), and using the same extraction method as before (i.e., 100-20% of cortical depth, average, surface-based sampling). We computed a two-tailed *t*-test to compare the response amplitudes between younger and older adults, which revealed significantly higher response amplitudes in older adults compared to younger adults (younger: 1.05±0.01, older: 1.08±0.008, *t*(34)=-2.10, *p*<.05, see Figure 2—figure supplement 1). This confirms prior findings on greater excitability / less inhibition of older adults’ SI maps when the same stimulation amplitude is used and/or when more than one finger is stimulated at a time. Note, however, that other studies also showed differences in cortical excitability when they adjusted stimulation amplitude to individual tactile thresholds (e.g., Pleger et al., 2016 Sci Rep) whereas we only see such a difference when the same amplitude was applied, and when all fingers were stimulated together. This may support the view that age-related differences in lateral inhibition and adaptation are a particularly critical factor to explain the larger cortical excitability in previous studies. Given in our design, only two small pins stimulated the fingertip at a time (see explanation above), lateral inhibition and adaptation differences in SI between younger and older adults may have influenced our major analyses to a lesser extent compared to other previously published datasets. We now report these results in the main text, and in Figure 2—figure supplement 1.

To address the reviewers’ concern whether the difference in excitability during same amplitude stimulation relates to individual tactile detection thresholds, we computed a correlation between mean tactile detection thresholds across fingers and mean response amplitudes across the topographic map area as reported above. These correlations were not significant, and are also reported in Figure 2—figure supplement 1.

To make this distinction clear to the reader, we now point out at different parts of the manuscript, including the Discussion, that the reported differences are based on an adjusted threshold stimulation.

9. A very large number of statistical tests are performed, and most effects were not significant. This could suggest that either the authors successfully identified very specific correlates of aging (e.g. D2-D3 differences), or maybe that some effects were significant by chance (e.g. the authors computed 20 correlations between the different measures and age and found that 2 out of 20 are significant without applying correction for multiple comparisons). Statistical analyses should be strongly hypothesis-driven to limit the number of tests, especially when dealing with some many variables. For instance, the reason for testing cortical distances between peaks is not clear. Is it to test the interpretation of previous results arguing that the increased D2-D5 distance is evidence for 'an enlargement of topographic maps in older adults', or to see if the authors can replicate this D2-D5 result? If the former, then is this analysis necessary considering that enlargement of maps was already assessed above with the surface area measure (that can be done both globally and in a finger-specific way)? If the latter, then why are the authors testing relative distances between adjacent fingers instead of distances relative to D5? In the same line, correlations between data not showing significant main effect of age or interactions should be avoided (e.g., correlation between mean pRF sizes and dice coefficients that are not significant). Similarly, running stats on both the factors neighbour/age and digit-pair/age for RSA consists in running the test twice on the same data (circularity), just pooled differently (and in an unbalanced way when it comes to neighbourhood). The same applies to the mislocalisation analysis. If the purpose is to test global vs local effects, the authors should have independent measures to assess each. For instance, while local effects can easily be measured by maps overlaps/RSA and mislocalisations, global effects could be assessed by global changes in surface area/pRF sizes and hand dexterity? Finally, the hypothesis and justification for the resting state analysis is not clear.

We are grateful that the reviewers raised these various concerns. We tested for cortical distances between peaks because previous studies have reported that older adults show greater distances between D2 and D5 compared to younger adults (Kalisch et al., 2009). The authors related this to global effects (i.e., a larger map), but it could also be explained by local effects (e.g., shift of D5 away from D4 or shift of D2 away from D3 and so forth). The latter result would support local map distortion that could also relate to worse performance, as argued by Saadon-Grosman et al., (Saadon-Grosman et al., 2015 *PNAS*). In fact, testing for local map distortions was the major rationale for the distinction between local and global effects in the first place. When computing the ANOVA between finger-pair and age, our expectation was that we would find a shift of a specific finger-pair (or of a cluster of fingers, see Essential Revision Nr. 4) away from each other. Our idea was that in case this change is non-organized (i.e., different between each of the older adults) and relates to greater map dispersion in older adults (see Figure 2), this may indicate local map distortion. Alternatively, we expected to see a larger map (i.e., as shown by replicating the increased D2-D5 distance, and by showing a larger topographic map area). Both of these assumptions were not confirmed, but we found a selective shift of D2 and D3 towards each other. Given this pattern was systematic and related to lower map dispersion in older adults, the hypothesis of greater local map distortion in older adults was not confirmed. This experimental approach is now explained in the Results section in the following way:

“Previous studies found larger cortical distances between D2 and D5 in older adults compared to younger adults, which was argued to evidence a global enlargement of topographic maps in older adults (Kalisch et al., 2009). At the same time, this effect could also reflect changes in the topographic alignment between individual digit pairs (local effect). We used both absolute (Euclidean) and surface-based (geodesic) cortical distances measures to compare distances of digit representations in SI between younger and older adults to test for both global and local differences (see Figure 3C).” (page 9)

We followed the reviewers’ suggestion and removed correlations from the manuscript that did not directly address a scientific question. We removed the correlations between average pRF size and dice coefficients, and we also removed the correlations between average pRF size and cortical distance from the manuscript as we had no specific hypotheses on the outcome of these correlations. Further, we removed the correlations between behavioral and fMRI measures and individual age (except for addressing a specific reviewers’ comment as was the case for the data now reported in Figure 5—figure supplement 1). Because those were not discussed in the main text and were also not part of our main hypotheses, we agree that the better approach is to remove these correlations to reduce the number of tests performed.

With respect to the comparison of different neighbours, when performing the mislocalization analyses where we compared the different neighbours to each other to estimate whether or not the distribution was non-random (*G*-test of goodness-of-fit), we accounted for the different amounts of data in the different levels by comparing the measured distribution of mislocalizations with the proportional distribution as expected by chance (see Figure 6—figure supplement 1). This statistical approach was introduced and explained in detail by Schweizer et al., (2000) for the given set of data, and we followed the same approach.

With respect to comparing neighbours as post hoc comparisons between younger and older participants (for example in RSA), this approach is justified because both age groups have the same variance and number of variables in each variable (because all have 5 fingers, and all fingers were touched 20 times). This analysis is therefore valid as long as normality and equal variance can be ensured, which we tested. Also, as pointed out above, since correlation coefficients are based on independent data (separate runs), the estimated mean correlation coefficients should not be positively biased. Since we carried out classical random-effects analyses, the different number of comparisons at the subject level should only minimally affect the estimated statistics at the group level.

The rationale for performing the resting state analyses was to investigate whether the age-related difference in N3 representational similarity would be reflected in differences in low-frequency fluctuations during rest. This would indicate that task-independent differences in functional connectivity between the fingers would drive the observed changes. Our data, however, do not support this hypothesis. This is now clarified in the following way:

“In addition, resting state data were used to investigate whether the distance-mediated differences in representational similarity between younger and older adults, as reported above, are accompanied by differences in low frequency fluctuations during rest (Kuehn et al., 2017). This could be indicated via decreased functional connectivity between N3 fingers in older adults.” (page 14)

10. While the correspondence between fMRI features and behaviour is quite convincing and suggests that these effects are not spurious, a more sensitive statistical approach might shed new light on the data. First, why did the authors decide to define two groups of age rather than using age as a covariate (e.g. ANCOVA or LMM)? This approach could be sensitive to both between-group differences and within-group differences, thus preventing to rely on a large number of correlations. Second, the comparison between fMRI features and behaviour mostly relies on the qualitative correspondence between significant effects (except for the correlations computed in the factor analysis). If the aim was to determine which local or global fMRI features are associated with behavioural changes and aging, it might be more optimal to directly compare fMRI features and behaviour by defining local fMRI features for individual fingers (e.g. area size, pRF size, response amplitude, etc…) and global fMRI features across fingers (e.g. distance, overlap, similarity, etc…). These fMRI features could then be used as co-variates to analyse behaviour (e.g. ANCOVA or LMM).

We agree with the reviewers that our analyses suggest a relationship between the functional fMRI measures and phenotypic behavioral measures that needs to be explored. It is first important to clarify that we did not define two groups in the factor model but we only tested the relationship between fMRI features and behavior within the group of older adults. The factor model was therefore only computed on older adults’ data and not on younger adults’ data, because our aim was to link the extent of specific features of cortical “de-differentiation” that should be present in older adults’ maps but not in younger adults’ maps to behavioral phenotypes. To follow the reviewers’ previous suggestions, we removed the correlations between functional variables, behavioral variables and age from the manuscript unless they were used to test specific hypotheses, or they were requested by the reviewers. Therefore, we do not report large numbers of correlations anymore.

We would also like to explain our concept of global versus local changes and how it relates to the reported analyses. The critical difference between global and local features is in how far the map area is affected (global = affects the whole map area, local = affects only parts of the map, see Figure 1). Therefore, each variable that we tested can in principle be classified as global and local dependent on whether the full map area shows a difference in this variable, or whether only parts of the map show a difference in this variable. This is what we tested by computing interaction effects between finger and age, and between neighbour and age as well as computing main effects of age. We find some differences at the global level (such as changes in pRF size and N3 differences that spread across the map at a specific distance) and some differences at the local level (such as D2-D3 distance and D2-D3 confusion). The difference between local and global does therefore not refer to the variable itself, but rather to the spatial extent of change. This is the reason why we cannot include both factors within one model, because this would violate the assumption of independence.

Instead, we followed the approach to first detect those global and local differences that differentiate younger from older adults’ maps (i.e., that define the presumably “de-differentiated map”), and to then use a factorial model to investigate relationships between these variables and behavior. This is clarified in the introduction in the following way:

“By combining ultra-high resolution functional imaging with population receptive field mapping, Fourier-based functional analyses, representational similarity analysis, psychophysics, and measures of everyday behavior, we could compare precise map features that differed between younger and older adults’ topographic maps, and link these to behavioral phenotypes relevant for everyday life.” (page 4)

In addition to adhering to the assumption of independence (by only including each variable once), we also have to address the problem of dimensionality reduction. This was done in the following way (now also described in the Materials and methods section):

“Factor analyses were conducted only on the data of older adults, and only with variables that showed differences between the two age groups. Reduction of the dimensionality of the data only to these variables that were different was necessary to reveal variables that showed joint variations due to unobserved latent variables and not due to intertwined features. Our goal was to reduce the dimensionality in the data down to the number of variables needed to capture its maximum variability. The number of dimensions does not necessarily correspond to the number of features, as not all variables measured were independent from one another. Our goal was to obtain a reduced set of degrees of freedom in the data, which could be used to reproduce most of its variability, and thus redundant features were of no interest. This redundancy was evident when we tried to include more variables in the model. By adding more variables, the presence of Heywood cases for some variables kept arising (meaning that these variables were entirely determined by common factors with specific variances-psi values close to zero) and in some cases, the model could no longer be computed. However, not all factor loadings provide meaningful information and their interpretation should be treated with caution, particularly when the sample size for this analysis is small. In general, there are different opinions regarding whether a factor loading is high or low (Maskey, Fei and Nguyen, 2018). A factor loading of a variable can be considered important for a factor, if it is at least |0.4| (Maskey, Fei and Nguyen, 2018), explaining around 16% of the variance in the variable (Field, 2009). We set a cut-off criterion for the accepted loadings in our model to |0.4| and followed the procedure of Maskey, Fei and Nguyen, (2018) regarding the exclusion of variables (i.e., deleting one variable at a time and always rerun the model, based first on the lowest factor loadings and then, if required, on the existence of cross-loadings). The first variable to be removed from the two-factor model was N3 representational similarity (loading of -0.19). After rerunning the model, pRF size was excluded next (loading of 0.30) and the model obtained after this second variable exclusion was the final two-factor model reported here. No further exclusions were required. The same process was followed for the three-factor model. pRF size was the first variable that was removed in this model (loading of 0.30). After pRF size exclusion, the obtained model was accepted with no further variable exclusions (all factor loadings >|0.4|). Correlation coefficients between variables of each factor were then calculated within the group of older adults using Pearson correlations.“ (page 42)

11. Since most of the effects tested here were non-significant, considering the Bayesian equivalents of these test (available in R or JASP) might provide useful information by assessing the evidence for the presence or absence of an effect (rather than rejecting a possible difference with frequentist approach).

We agree with the reviewers on this point. We took two approaches to provide additional information on non-significant results that were interpreted as evidence for the absence of an effect. First, we calculated Cohen’s *d* as a measure of effect size and now provide the results in the text. We adhered to the classification of *d* > 0 and *d* < 0.2 as no effect, of *d* > 0.2 and *d* < 0.5 as small effect, and of *d* > 0.5 and *d* < 0.8 as intermediate effect (as suggested by Cohen, 1988). With respect to the non-significant test results reported before, we found that both measures of surface area revealed no effect (*d* = 0.048 and *d* = 0.10), that the geodesic distance between D2 and D5 revealed no effect (*d* = 0.04), that the Euclidean distance between D2 and D5 revealed no effect (*d* = 0.12), and that percent signal change and the f-value revealed a small effect (*d* = 0.38 and *d* = 0.30). This is now reported in the text.

For non-significant results that were interpreted as evidence for the absence of an effect that presented with Cohen’s *d* > 0.2 (that is, with a small effect), we also computed statistical tests of equivalence. We used the two one-sided t-test (TOST), which is a frequentist alternative for testing for the equivalence by defining a band around 0 that constitutes a minimally-relevant effect size (Δ*_L_* and Δ*_U_*). (A similar specification is required for Bayesian alternatives by specifying the width, and possibly shape, of the prior). TOST works (1) by running two t-tests, testing the null hypothesis that the effect is smaller than the maximum of the indifference area and larger than its minimum, and (2) by choosing the smaller of the two t-values. A significant result would reject this null hypothesis, indicating that the true value lies in the indifference area. For more detail, please see Lakens, (2017) Equivalence Tests: A Practical Primer for *t* Tests, Correlations, and Meta-Analyses.

Using this approach, we tested non-significant results using the TOST equivalence test on results which were critical for the interpretation of our data. To determine Δ*_L_* and Δ*_U_*, we used an approach suggested by Simonsohn, (2015) who argue that in order to determine boundaries for equivalence tests, the boundaries should be set to the effect size of the distribution where a difference could be detected with 33% power (Simonsohn, (2015). Small telescopes detectability and the evaluation of replication results. Psychological Science, 26, 559–569). In the TOST procedure, the null hypothesis is the *presence* of a true effect of Δ*_L_* or Δ*_U_*, and the alternative hypothesis is an effect that falls within the equivalence bounds or the *absence* of an effect that is worthwhile to examine. If the *p*-value is below 0.05, we can therefore assume the *absence* of an effect that is worthwhile to examine. This was the case for % signal change (*t*(34)=-2.11, *p*=.000044) and mean f-values (*t*(34)=0.55, *p*=.13). We now report these statistics in the manuscript and conclude that whereas there is no difference in most variables reported, there is a small difference in mean f-values between groups that is not significant but that is also large enough not to be considered indifferent.

12. The authors are particularly lenient with describing effects as 'trending towards significance'. P-values above 0.1 should certainly not be described this way. In some places, the (range of) p-values were not given directly in the main text, making it harder for the reader to evaluate the strength of the evidence.

We apologize if p-values above 0.1 were described as trends in the previous version of the manuscript. We removed these statements. We now also provide information on exact p-statistics when we report non-significant effects. In addition, as noted in response to comment No. 11, we now performed additional tests on the non-significant results in order to reject a tested hypothesis.

13. Please provide more details regarding the Bayesian pRF modelling:a. Voxels were preselected prior to the analysis, but only a significance threshold is given. What was the contrast used to exclude voxels?

Bayesian pRF modelly analysis in the present study was performed by closely following the procedure of Zeidman et al., (2018) and Puckett et al., (2020). They divided the analysis into two stages: The first stage was to conduct a GLM analysis, which was performed by applying a general first-level analysis to prepare data for the pRF modelling procedure using SPM12. Similar to Puckett et al., (2020), 5 regressors were defined, corresponding to the five fingers of the right hand.

1 0 0 0 0

0 1 0 0 0

0 0 1 0 0

0 0 0 1 0

0 0 0 0 1

An F-contrast and a threshold of p<.05 (uncorrected) was employed to reduce the number of voxel time-courses to model, and to reduce the number of BOLD time-series required for the modelling computations (Puckett et al., 2020). Note that this procedure is recommended to restrict the computationally heavy modeling (i.e., 36-48 hours per subject) to those voxels that show responsivity to the stimulus (Zeidman et al., 2018). Then, the extracted BOLD time-series were sent into the second stage, the actual pRF modelling analysis (Zeidman et al., 2018). During pRF modelling, there was no contrast employed, except that within the area of regions of interest, voxels were excluded if the pRF parameters failed to deviate from their priors (p > 0.95). This is now clarified in Materials and methods in the following way:

“BOLD time-series were extracted for pRF modeling by reducing the number of voxel time courses. This was achieved by performing a two-stage analysis. The first stage was the GLM analyses stage, which was accomplished by performing a general 1st-level analysis with SPM to prepare data for pRF modelling procedure. At this stage, task regressors were first defined. Similar to Puckett et al., (2020), 5 regressors were constructed, corresponding with five fingers of the right hand. After performing an F-contrast, only time-series that passed a significance threshold of p<.05 (uncorrected) were used for the pRF modeling (Zeidman et al., 2018, Puckett et al., 2020). This procedure allows reducing computing time considerably (note that one pRF modeling process takes around 2 days for the given input data). The resulting mask was combined with the freesurfer mask of area 3b. pRF modeling was then conducted on a voxel-wise basis, where the fit between an estimated waveform and the empirically measured BOLD-response was optimized. This was achieved by modifying the size and position of the pRF model. The posterior model probability was thresholded at >0.95 (Puckett et al., 2020; Zeidman et al., 2018).” (page 40)

b. Which prior(s) was/were used – if the prior was based on 'typical pRF estimates' did this bias your estimates away from finding group differences?

The priors used in this study were chosen based on the recommendations of the developer of the Bayesian pRF toolbox (Zeidman et al., 2018), and based on their use in a later study on SI pRF mapping using the same method (Puckett et al., 2020): To constrain the centre of the pRF to fall within a certain area, latent variables were used. Two latent variables, namely l_ρ_ and l_θ_ (which control the polar coordinates, i.e., distance and angle), were introduced to constrain the pRF location, while latent variable l_β_ was introduced to constrain the neuronal response to be positive. To complete the neuronal model specification, each latent variable was assigned a prior (normal) distribution, as the estimated parameters provided by the algorithm employed in the toolbox were computed under the Laplace assumption. According to Puckett et al., (2020), using voxel-wise priors would only have little impact on the final results compared with using the same prior for all voxels. Hence, the same priors for all voxels were also employed in the present study, which is not expected to bias pRF estimates away from finding a group difference.

c. Why did the authors use Bayesian pRF modelling, rather than a frequentist approach, which would match the rest of the manuscript's statistics?

The Bayesian approach to pRF modeling has advantages compared to the traditional pRF technique (Dumoulin and Wandell, 2008). The reasons are that estimates of the uncertainty associated with the pRF parameter estimates are provided, and that variability in the hemodynamic response across the brain is accounted for. This is particularly important for pRF modeling of older adults where a precise match between the BOLD response and the model is required. This method has already been applied in the tactile domain by Puckett et al., (2020), which allowed us to use an established framework and a similar experimental approach to answer questions on age-related differences in SI. This is an advantage compared to introducing a novel model to the tactile domain where the relation to previously reported data is unknown.

In addition, it is worth noting that the use of a Bayesian statistical model is not the same as using Bayesian inferential statistics. Unless one applies Bayesian inferential statistics to it, one is not mixing two philosophies. The alternative to Bayesian modeling is not frequentist statistics but maximum likelihood estimation.

d. Please provide more information about the fit. Could the larger RF estimates in older participants be due to a poorer fit of the Gaussian?

Regarding this question, we have sought advice from the developer of the Bayesian pRF modelling toolbox, Dr. Peter Zeidman. We would like to refer to his view on this question, which is the following:

“The model ‘fit’ mentioned here would be signal-to-noise ratio, which was scored by the % explained variance. One may ask what role signal-to-noise ratio plays in the modelling we performed here. For instance, what would be the consequence if older people have noisier data? Would it lead to worse model fits? The generative model we employed partitions variance in the BOLD signal into three parts: the pRF model, the haemodynamic model, and observation noise. Each of these parts are parameterized, and Bayesian methods are used to estimate a (multivariate normal) probability density over the parameters. As a result, we obtained both the expected values of the parameters, as well as their (co)variance or uncertainty. Decreased SNR would therefore be expected to increase the contribution of the noise part of the model, and potentially decrease the estimated precision of the pRF and haemodynamic parameters. In the extreme case of uninformative data, all parameters would revert to their prior expectations.”

It was not the case in our data that nosier data lead to worse model fits. In our data, the parameters encoding pRF size confidently moved away from their prior expectation in almost all subjects. This means that the larger pRF estimates we obtained from older participants was not due to poorer fit of Gaussian during modelling.

e. Is there good correspondence between the pRF distance estimates and the topographic maps estimated using phase encoding? Please also provide a supplementary figure with all participants' pRF estimates.

We thank the reviewers for this point. We now added a figure where we show topographic maps based on phase estimates and topographic maps based on pRF estimates of all individual participants. As can be inspected in Figure 4—figure supplement 1 in the manuscript, there is a clear correspondence in the arrangements of both topographic maps within each individual. As can be inspected, the Fourier maps are larger than the pRF maps, which is due to the sensitive statistical approach used for frequency-based mapping that usually offers higher *t*-values compared to other statistical techniques. For the pRF mapping, a biological model of expected tissue responses is used to identify high probability values that are used for thresholding, which is more conservative compared to taking into account each voxel that responds to the respective stimulation frequency and phase. In this respect, both analyses offer complementary information on topographic map features.

f. Relatedly, P26 who appears to be an outlier with very high pRF size for D5 (in Figure 4A) has medial activity for D1 (Figure 2C). If 'digit space' is parameterised as a circle instead of a line (allowing shared D5-D1 RFs), are the group estimates affected?

P26 is indeed an outlier who shows relatively high pRF size for D5 (as shown in Figure 4A) and medial activity for D1 (shown in Figure 2C). Outliers are generally expected when investigating older adults due to the high heterogeneity of the individual aging process (in the range between “superagers” to “normal agers” and “fast agers”, Depp and Jeste, 2006; Borelli et al., 2018 and Sun et al., 2016). To further address this point, we performed an ANOVA with the factors age and finger calculated on pRF sizes with P26 excluded to investigate if the outlier has an impact on the results. Our results revealed that also without the outlier included in the model, there was a significant effect on age (F(1)=0.58, *p*<.05) as well as a significant effect on finger (F(4)=13.26, *p*<.0001), whereas no significant interaction between age and finger was shown, as before (F(4)=1.7, *p*=.15). This result replicates the previous ANOVA result where P26 was included. These additional analyses are now also reported in the manuscript.

It is worth noting that although the D5 pRF size of P26 appears to be relatively high, it still follows the expected pattern that pRF centre locations of D1-D5 are arranged from superior to inferior. In addition, the calculation of pRF sizes was performed based on the defined regressors, which would not be influenced by the locations of pRF shown on the surface. That is, even if the ‘digit-space’ was defined into a circle instead of a line, the result would remain the same, as the pRF for five fingers would be calculated separately.

g. "The main effect of finger was due to significantly larger pRF sizes of D1 compared to D2" – p. 8. Based on figure 4, I think this should be "smaller".

Thank you for this comment, the corresponding context has been modified in the manuscript.

h. "For display, the dimensions of the pRF distance volumes were adjusted to 0/25." – p. 31. It is unclear what this means, please rephrase.

This sentence has now been rephrased to make this clearer. It now reads: “Before pRF modeling, parameters for pRF centre locations were set between -12.5 (low x-values) to +12.5 (high x-values), and after pRF estimation, values were modeled accordingly. For display, the values are shown with a range between 0 and 25 to show the boundaries between each finger in integral numbers.”

14. Representational similarity analysis (RSA):a. The distinction between 'local' and 'global' is not very clear, particularly when applied to distant finger pairs. Are N3 correlations the within-subject average of D1-D4 and D2-D5? Do the authors propose a global mechanism that only affects these two finger pairs, but not, say, D2-D4 or D3-D5?

The distinction between local and global effects has now been clarified in response to concern no. 3. The reviewer is right that we interpret the N3 effect as a global effect given the effect seems to be distance-mediated rather than location-mediated.

b. RSA values are typically plotted as dissimilarity (e.g. 1-corr.). The decision to plot the corr. directly while using the term RSA may cause some confusion. This is at the authors' discretion.

RSA values can in principle be plotted both as similarity and dissimilarity. We chose the former to align with the other matrix plot in the same figure that shows similarity (i.e., resting state correlations). We chose different colors for both matrices to show that the measures are different. We think that the interpretation of similarity is more intuitive in the present context of the figure than the interpretation of dissimilarity.

15. Dexterity and factor analysis:a. The distinction between measures of hand dexterity that are 'global' and 'local' seems to be fully driven by the results from the factor analysis. As it stands, it is unclear why the O'Connor test is fundamentally different from the other two. The authors should attempt to explain why the O'Connor test was associated with a positive loading on local effects, while the Purdue and Grooved tests were associated with a negative loading on the global effects. These three tasks, while corresponding to different levels of difficulty, were described as assessing similar aspects of hand dexterity. It would also be helpful if the authors can provide information about potential different strategies used for the three tasks across the different age groups (e.g. which fingers are primarily used to handle the pins, did this differ across the tasks and across age groups).

We agree with the reviewers that the Purdue Pegboard and the Grooved Pegboard tasks are more similar to each other than the O’Connor task is similar to the other two. This is evident in our data when inspecting the results of the factor analysis, and by inspecting correlation coefficients between these measures, but this is also evident when inspecting prior research. Previous studies have found that particularly demanding versions of the Pegboard task relate to measures of tactile spatial acuity at the fingertip in older adults (Kalisch et al., 2008, Trembley et al., 2003). We replicated this effect by now incorporating 2PDT data into our manuscript (see comment No. 2) that show higher correlation coefficients (and a significant correlation) between the 2PDT and the more difficult Grooved Pegboard Task compared to the 2PDT and the easier Purdue Pegboard Task (r = 0.68 versus r = 0.35). This confirms the view that Pegboard tests rely on an accurate precision grip that relies on good spatial acuity at the fingertip (Kalisch et al., 2008).

Previous research has also shown that the relationship between tactile acuity at the hand and haptic object recognition is weaker (Kalisch et al., 2008). This is similar to our data, where we do not find a significant correlation between the O’Connor Dexterity test and the 2PDT. In fact, the O’Connor Dexterity test relies more on haptic object recognition than the Pegboard tasks by the requirement of the former to grasp exactly three needles out of a box with 20-30 needles in it. Usually, participants take the longest time of the task to grasp the three needles and align them to each other (both of what they have to master using one hand only) rather than placing the needles into the holes. In the other two tasks, however, the major requirement is to put the object correctly into the holes. Therefore, there is evidence both in prior research and in our data that the O’Connor task relies on haptic object recognition and thus interaction between the fingers, whereas the Pegboard tests rely on tactile spatial acuity at the finger tip, where within-digit spatial acuity is more relevant than between-digit interaction. This difference between the tasks is now also explained in the introduction and discussion.

b. In the Discussion section and abstract, then, the O'Connor test is reduced to "hand dexterity", e.g. in "more local 'de-differentiation' related to better hand dexterity" – p. 18. Given the other two tests of hand dexterity showed different effects, this should be phrased more carefully.

We agree with the reviewers. In the abstract, we now replaced the term hand dexterity with motor performance. In the discussion, this is now explained in more detail:

“In contrast to rats, we also do not see a significant correlation between larger pRF sizes and worse motor control of the hand, neither for motor tasks that rely on precision grips nor for a motor task that relies on haptic exploration. We also do not see a significant correlation between pRF size and spatial tactile acuity at the fingertip. The behavioral relevance of increased pRF size in humans therefore remains to be investigated.” (page 23)

“In this respect, the distinction between functional readouts that capture integration versus separation seems relevant (see Figure 1). Specifically, we found that reduced cortical distances between D2 and D3 representations related to worse tactile discrimination between these two fingers (i.e., worse separation in a tactile task) but to better motor control in a task that required haptic object exploration and coordinated finger movements (i.e., improved integration in motor task). These findings highlight the importance of assessing different aspects of signal integration and signal separation to investigate the impact of functional map changes on everyday life. Because this relationship was only seen for the index and middle finger and not for the other fingers, also the factor of spatial extent (i.e., local versus global, see Figure 1) is relevant for comprehensive data analysis.” (page 25)

c. Both Euclidean and Geodesic distance are included as inputs to the factor analysis and are the inputs with the strongest loadings on factor 1. Does this influence the factors?

These two variables are indeed the ones with the highest loadings on factor 1, both in the revised two-factor model and three-factor model. To better understand their contribution to the factor, we should consider the correlations between all different variables loading on this particular factor and their specific variances (psi values). For the Euclidean distance, the specific variance was 0.04 in both models and for the Geodesic distance, it was 0.22 and 0.21 for the two- and three-factor model, respectively. The low values indicate that for both variables, variance is entirely determined by the common factor. Additionally, since these two variables highly correlate with each other (*r*=.85) and both measure cortical distances, it is not surprising that one common factor exists and greatly determines their variance. In our view, this does not constitute a major problem for the other variables included in the model. Variables loading on the same factor correlate in a similar way with both cortical distance measures, while at the same time these measures do not seem to affect the variables loading on the other factor, since they do not significantly correlate with any of them.

d. Why were the various non-significant digit pairs or neighbourhood relations (for confusion, distance, or repr. correlation) not included in the factor analysis? Perhaps one way to show how meaningful the distinction between the factors is would be to show that even non-significant group differences show an appropriate difference in factor loadings.

Although we did consider the possibility of adding more variables into the model, the low number of observations for each variable did not allow us to follow this approach. As we already explained in our answer to comment no. 10, and as is also explained in our answer to comment no. 23, it was essential to reduce the number of degrees of freedom in our data and minimise it to those variables that explained most of the variance. But we agree that this approach is interesting for future studies, particularly those that have the possibility to measure >50 older adults in the 7T-MRI. This is usually a problem for a one-center study, however, due to the strict exclusion criteria for 7T scans that particularly affect older adults.

[Editors' note: further revisions were suggested prior to acceptance, as described below.]

The manuscript has been much improved but there are some remaining issues that need to be addressed, as outlined below:Essential revisions:1. The authors state in the results that the finger maps appeared in topographic alignment in all participants (page 6 and page 11). However, this is hardly appreciable when looking at the figures (Figure 2, 4 and 4S). There is also no description about how this topographic alignment was assessed. We recommend assessing statistically the topographic alignment between digits (different from the between subject dispersion analysis) to support these claims. Topographic alignment is visually depicted in Figure 3E, but this assessment is not described in the Results section.To be specific, on page 6, "The expected topographic alignment of the fingers (thumb [D1], index finger [D2], middle finger [D3], ring finger [D4], and small finger [D5] arranged superior -> inferior in area 3b) was visible in all individuals of both age groups (see Figure 2C)", this seems like an overstatement after visual inspection of Figure 2C. Visual inspection of individual maps (Figure 2, 4 and 4S) suggest the presence of "missing" fingers in some individuals. This might be the case if no voxel responded preferably to a finger.For example P1, P18, P25, P28, P33, P36 seem to have at least 1 missing color in the map. Please specify how the topographic alignment was assessed and how many participants had "missing" fingers in the maps (if any), and report the minimum number of voxels associated with a given finger in the topographic maps. In case a given subjects had a "missing finger", how was missing data treated for statistics?Further, on page 11, "pRF centre locations revealed an organized finger map with D1-D5 arranged from superior to inferior, as expected", the authors probably meant "inferior to superior. As for phase encoding, please specify how the topographic alignment was assessed and whether some participants had "missing" fingers in the maps and how this was treated for statistics. Inspection of figure 4S suggests indeed that some subject had "missing" fingers in their maps.

We apologize for the confusion with respect to the above outlined analyses. On page 6, we refer to the Fourier-based analyses when describing the maps because those were used to estimate surface area, % response amplitude, and f-values, as well as the alignment. This is now clarified in the manuscript (i.e., it now says “topographic Fourier-based maps”). For the Fourier-based analyses, we did not see any missing fingers; each finger was significant for each single individual. This is now also mentioned in the manuscript. For better clarification, each individual map is now shown in an enlarged view in Figure 2—figure supplement 1 and Figure 2—figure supplement 3:

Following the reviewers/editors suggestion, we now moved the description about topographic arrangement to a later part of the manuscript where we describe the group analysis of topographic arrangements based on the blocked-design data (shown in Figure 3E). Here, we now added a visualization of each individual topographic arrangement as Figure 3—figure supplement 1, and we also added an explanation to the text how the data were generated. The exact alignment of each individual’s map can therefore now be inspected.

2. In the GLM used to define finger-specific maps, which are then used for overlap and resting-state analyses, the statistical threshold used is too liberal (p<0.05 uncorrected) which might affect the results of subsequent analyses. Page 36, line 1155: "voxels that survived a significance threshold of p<.05 (uncorrected) and k>3", this is a very liberal threshold to define finger-specific areas, which will likely detect unspecific responses and result in larger overlap between digits. This is problematic as the result "increased overlap in elderly's maps" might be attributed to difference in unspecific responses. In addition, these maps were used for resting-state data extraction, which is also problematic, because an increased overlap would predict an increased resting-state functional connectivity. It may be better to use a more conservative threshold (e.g. 0.001 uncorrected) and to exclude any overlapping areas in the resting-state analysis.

We agree that the dice coefficient analyses are influenced by the chosen statistical threshold. In most papers that we inspected prior to study design, a different statistical threshold and analysis technique was chosen to report dice coefficient results. Therefore, before reporting the final results, we explored the data by calculating finger- and age-specific overlaps using different statistical thresholds. When inspecting the results, it is evident that at the statistical threshold where dice coefficients are above 0.1 (i.e., where neighbouring fingers start to overlap), older adults show greater overlap than younger adults. Therefore, the chosen threshold did not bias our finding that olders show higher dice coefficients compared to younger adults. These additional analyses are now incorporated into the manuscript as Figure 4—figure supplement 3.

With respect to the resting state analyses, changing the statistical threshold will not solve the problem of overlapping voxels, given neighbouring voxels are overlapping even at very high thresholds (see Figure 4—figure supplement 3). For the correlation analyses, however, overlapping voxels were excluded from the data. This information is now added to the methods section.

3. Considering that the authors select the variables 'that were shown differ between age groups' as inputs to the factor analysis, and that they now report a significant main effect of age on dice coefficients (and not just a trend), these should be added to the factor analysis. Related to that, please consider adding the dice result in the summary of your results at the beginning of the discussion.

We have now added mean dice coefficients to the factor analysis as suggested by the reviewers/editors. Some key scientific findings such as a shift in topographic maps in amputees that had been reported using the euclidean distance could not be replicated by using geodesic distance measures (i.e., Makin et al., 2013 Nat Commun). It has therefore been argued that only geodesic distance reflects the true “cortical distance” where euclidean distance also reflects aspects of wm connectivity and superficial wm fibers. In fact, we were surprised by the high correlation coefficient that we found in our study between the two measures. Therefore, we think that this information is important and should be reported. The cut-off criterion for the accepted loadings in our models remained the same as before (|0.4|). The new model reveals similar results as the previous model, where N3 representational similarity has again a separate factor loading (factor 3), and the O’Connor dexterity test, D2-D3 confusion, and cortical distances load on factor 1 (local effects), as before. Factor 2 is now composed of the Grooved Pegboard, the Purdue Pegboard, N3 confusion, pRF size, dice coefficients (global effects). The data and text is now updated in the manuscript (i.e., Figure 7).

The result of greater dice coefficients in older adults has now also been added to the summary of the results at the beginning of the discussion.

4. Response to previous comment 7. The scale is saturating the colours in the RDMs so it is difficult to inspect. Could the authors report the same stats as reported in the main text (ANOVA neighbour*age and ANOVA finger pairs*age) to see if results hold?

We follow the reviewer’s suggestion and now plot the colors using the same colormap in a smaller range between -0.5 and 0.5. In addition, we now report the same statistics as reported in the main text. With multivariate noise normalization, as before, the interaction between age and finger-pair is not significant (F(3,102)=0.26, *p*=.85) but now also the interaction effect between age and neighbour is no longer significant (F(4)=1.14, *p*=.34). This is not unexpected, since in our case, multivariate noise normalization strongly reduced the overall reliability (mean digit reliability before: 0.47 (younger adults), 0.45 (older adults), after: 0.24 (younger adults, 0.24 (older adults)), which has been reported elsewhere previously (Charest, Kriegeskorte and Kay, 2018, Neuroimage, Figure 4a; and Ritchie et al., 2020, Annual Meeting of the Vision Science Society, https://doi.org/10.1167/jov.20.11.515).

[Editors' note: further revisions were suggested prior to acceptance, as described below.]

The manuscript has been substantially improved but there are a few remaining issues that need to be addressed, as outlined below. We anticipate that these remaining issues can be assessed by the editors, without sending the manuscript out to the reviewers again.1. From the authors' response to point 1 (p. 3 of the response to reviewers):"With respect to the missing fingers, there were fingers 'missing' in the pRF centre location maps for 8 out of 36 subjects, as shown in Figure 4S. This is equal for the group of younger and older adults (n=4 in each group). The ANOVA with the factors age and finger calculated on pRF sizes was calculated with the missing cases excluded from the data, as described on page 11. This is justified because overall, the missing values take up 8.9% of the data where biases are expected when more than 10% of the data are missing (Dong, Y., and Peng, C. Y. J. (2013). Principled missing data methods for researchers. SpringerPlus, 2(1), 1-17, Bennett, D. A. (2001). How can I deal with missing data in my study? Aust N Z J Public Health, 25(5), 464-469.)."However, this information does not seem to be included in the manuscript. Page 10 describes the relevant ANOVA, but there is no mention of missing fingers or values. Please make sure to include this.

We have now included the corresponding description on missing fingers on Page 11 in the manuscript.

2. Please include the results of the analysis with multivariate noise normalization (response to reviewers point 4) in the manuscript. This could be either in the text or as a supplementary figure.

We now included the figure as supplementary figure into the manuscript (Figure 5—figure supplement 2), where we report the exact statistical results in the figure caption.

3. eLife's policy is that article titles should clearly indicate the biological system under investigation; please consider changing your title accordingly. "Topographic maps" is ambiguous; you might consider "somatotopic maps" or "topographic maps in somatosensory cortex".

Thanks for noticing this, the title has now been changed to:

“The organizational principles of de-differentiated topographic maps in somatosensory cortex”